# BayesTune: Bayesian Sparse Deep Model Fine-tuning

**Minyoung Kim**[1]
[1]Samsung AI Center Cambridge, UK
mikim21@gmail.com

**Timothy Hospedales**[1,2]
[2]University of Edinburgh, UK
t.hospedales@ed.ac.uk

## Abstract

Deep learning practice is increasingly driven by powerful foundation models (FM), pre-trained at scale and then fine-tuned for specific tasks of interest. A key property of this workflow is the efficacy of performing sparse or parameter-efficient fine-tuning, meaning that by updating only a tiny fraction of the whole FM parameters on a downstream task can lead to surprisingly good performance, often even superior to a full model update. However, it is not clear what is the optimal and principled way to select which parameters to update. Although a growing number of sparse fine-tuning ideas have been proposed, they are mostly not satisfactory, relying on hand-crafted heuristics or heavy approximation. In this paper we propose a novel Bayesian sparse fine-tuning algorithm: we place a (sparse) Laplace prior for each parameter of the FM, with the mean equal to the initial value and the scale parameter having a hyper-prior that encourages small scale. Roughly speaking, the posterior means of the scale parameters indicate how important it is to update the corresponding parameter away from its initial value when solving the downstream task. Given the sparse prior, most scale parameters are small a posteriori, and the few large-valued scale parameters identify those FM parameters that crucially need to be updated away from their initial values. Based on this, we can threshold the scale parameters to decide which parameters to update or freeze, leading to a principled sparse fine-tuning strategy. To efficiently infer the posterior distribution of the scale parameters, we adopt the Langevin MCMC sampler, requiring only two times the complexity of the vanilla SGD. Tested on popular NLP benchmarks as well as the VTAB vision tasks, our approach shows significant improvement over the state-of-the-arts (e.g., 1% point higher than the best SOTA when fine-tuning RoBERTa for GLUE and SuperGLUE benchmarks).

## 1 Introduction

Practical deep learning increasingly relies on a workflow of fine-tuning pre-trained foundation models for the specific task of interest to be solved. The foundation models (FM)s are often pre-trained at scale using large architectures and self-supervised objectives on large scale unlabelled data, allowing them to encode a great deal of general prior knowledge. Fine-tuning then specialises these general purpose FMs to the specific task of interest. The key challenge then becomes how to conduct fine-tuning so as to balance the conflicting goals of adapting to the task at hand; while retaining the knowledge obtained during foundational pre-training and avoiding overfitting on the (typically smaller) downstream training set. This is addressed from a variety of perspectives from careful learning rate scheduling to regularised fine-tuning [26, 12]. However, the mainstream approach in practice is largely based on some instantiation of sparse fine-tuning.

Sparse fine-tuning, also known as parameter efficient fine-tuning (PEFT) [23, 43, 14, 24, 22, 15, 16, 18, 45, 13, 10], manages the adaptation/forgetting trade-off by selectively updating a specific subset of parameters in the FM while keeping others frozen. These approaches also benefit from good parameter scalability with respect to the number of downstream tasks to solve since each downstream

37th Conference on Neural Information Processing Systems (NeurIPS 2023).

task only requires storing a small number of parameters compared to the full FM. A large number of sparse/PEFT approaches have now been proposed, based on either additive or mask-based principles. Additive approaches [16, 15] inject a hand-chosen set of additional small modules to be learned, while keeping the original FM entirely frozen; while mask based approaches [43, 18] manually select a sparse subset of original FM parameters to consider as learnable – such as the prompt parameters in transformers [18]. Despite substantial engineering effort, prior sparse fine-tuning work mainly relies on human intuition and ad-hoc heuristics to specify additive modules and sparse fine-tuning masks. This raises the question *how can we automatically select which parameters to insert or update in a principled and effective way?*

In this paper, we address this issue by proposing an automated sparse fine-tuning method. Our BayesTune framework automatically discovers the parameters to update, with minimal human intervention. In contrast to some early concurrent attempts that rely on heuristics and/or heavy approximations [45, 13, 10], BayesTune is highly principled, underpinned theoretically by Bayesian marginalization of parameter uncertainty. Specifically, we place a sparse Laplace prior on the FM parameters, and the posterior inference of the sparse scale parameters reveals which parameters are most important to update from their initial values. BayesTune is efficiently implemented with Langevin MCMC, which only requires twice the cost of vanilla SGD. We demonstrate BayesTune with both RoBERTa transformer on (Super)GLUE benchmark suite and Vision Transformer on VTAB benchmark suite. BayesTune can achieve a specified sparsity level, and provides a reasonable way to identify a good sparsity/performance trade-off if a sparsity constraint is not pre-defined. Finally, we show that BayesTune can work with both mask-based search spaces to identify existing FM parameters to update by fine-tuning, as well as additive search spaces to identify which new modules to add where for additive adaptation.

## 2 Problem Setup and Motivation

First we introduce some key notations as follows:

$$\theta = (\textbf{Variable}) \text{ Parameters (i.e., weights and biases) of the FM}$$

$$\overline{\theta} = (\textbf{Constant}) \text{ Parameters (i.e., weights and biases) of the } \textbf{pre-trained} \text{ FM}$$

$$D = \text{Training data for the downstream task}$$

We denote by $\theta_i$ ($\in \mathbb{R}$) the $i$-th element of $\theta$ by considering $\theta$ as a whole vector of concatenated parameters, i.e., $\theta = [\theta_1, \ldots, \theta_d]^\top$ where $d$ is the number of parameters (so, $\theta_i$ is scalar). Before we start, we clarify the goal of the sparse fine-tuning formally as follows.

**Desiderata.** We aim to find $\theta$ which performs well on the downstream task, and at the same time remains close to $\overline{\theta}$ sparsely in $L_0$ sense (i.e., $||\theta - \overline{\theta}||_0 = \sum_{i=1}^{d} \mathbb{I}(\theta_i \neq \overline{\theta}_i)$ is small). Roughly speaking, $\theta_i = \overline{\theta}_i$ for many $i$'s. Oftentimes we define and set the sparsity level $p \in [0, 1]$, meaning that the proportion of the FM parameters *updatable* from the pre-trained values is no greater than $(100 \times p)\%$ (i.e., $||\theta - \overline{\theta}||_0 \leq \lfloor d \cdot p \rfloor$).

**About the sparsity level $p$.** We remark here that the sparsity level $p$ is often given by the users, for instance, in case we have a strict pre-specified memory constraint (e.g., embedded platforms), where the allowable extra space for saving the updated new parameters from $\overline{\theta}$ for the downstream task is only $(100 \times p)\%$ of the original FM storage. In other situations, it would be ideal if the sparse fine-tuning algorithm can estimate optimal $p$ that trades off the downstream accuracy against memory overhead (e.g., find the smallest $p$ that enables a certain level of accuracy guarantee). Unfortunately most existing sparse fine-tuning methods are not capable of this feature, and rather resort to time-consuming cross-validation-type search. Although this latter case opens up a new interesting research problem to pursue further, our proposed approach offers reasonable and principled criteria for estimating the optimal $p$ (See Sec. 3 for details).

**How about layer-wise selection?** We also note that our approach is not tied to the layer-wise parameter treatment, namely selecting a few layers to be updated with the rest layers frozen, which is a popular strategy adopted in many sparse fine-tuning methods. The layer-wise treatment is based on the conjecture that whether the parameters important to the downstream task are deviated from the pre-trained ones or not, is determined in a layer-wise all-or-nothing manner. Although this is a reasonable assumption, our approach does not rely on the assumption, and is able to discover optimal sparse updates automatically by inspecting individual parameters over all layers.

Back to our notations, the downstream task must be associated with the relevant loss function $l(\theta; z)$ with $z \in D$ for fine-tuning. Here the task could be either supervised (i.e., $z = (x, y)$ for a pair of input $x$ and its target label $y$) or unsupervised ($z = x$ with input $x$ alone). In our model the loss function is turned into a likelihood model by the conventional trick, $p(z|\theta) \propto \exp(-l(\theta; z))$. Although our approach can deal with both supervised and unsupervised cases seamlessly, we predominantly focus on the supervised classfication case, that is, we define the data likelihood $p(y|x, \theta) \propto \exp(-l(\theta; x, y))$ for $(x, y) \in D$, where $l(\theta; x, y)$ is typically the cross-entropy loss.

## 3 (Proposed) Bayesian Sparse Fine-tuning Model

We propose a (hierarchical) Bayesian model to tackle the downstream prediction task of interest. Specifically, we treat the FM parameters $\theta$ as *random variables* and the downstream training data $D$ as *evidence*. To encourage parameter sparsity, namely most parameters $\theta_i$ remain at the pre-trained $\overline{\theta}_i$, we impose the Laplace prior $p(\theta|\lambda)$,

$$p(\theta|\lambda) = \prod_{i=1}^{d} p(\theta_i|\lambda_i) = \prod_{i=1}^{d} \mathcal{L}(\theta_i; \overline{\theta}_i, \lambda_i) = \prod_{i=1}^{d} \frac{1}{2\lambda_i} \exp(-|\theta_i - \overline{\theta}_i|/\lambda_i) \tag{1}$$

where $\mathcal{L}(x; \mu, b) = \frac{1}{2b} \exp(-|x - \mu|/b)$ is the (univariate) Laplace distribution with mean $\mu$ and scale $b$. Thus we fix the prior means as the pre-trained values $\overline{\theta}$, and only model the scale parameters which are newly introduced random variates denoted by $\lambda$ ($> 0$). Note that every single parameter $\theta_i$ (scalar) is associated with its own prior scale $\lambda_i$ (thus $d$ scale variables in total). Another thing to note is that the final layer (also known as the readout head) of the FM is often completely replaced by a random one for the downstream task. Since this final layer needs to be learned from the scratch with the downstream data, we do not place the Laplace prior on the parameters of the readout head.

Through the scale variables $\lambda$ of the Laplace distributions, we can express our (prior) preference to the degree of parameter deviation from the pre-trained values: small $\lambda_i$ leads to a peaky Laplace around its mean, penalizing even small deviation of $\theta_i$ from $\overline{\theta}_i$; on the other hand, large $\lambda_i$ makes it flat, being less sensitive to deviation, allowing $\theta_i$ to take values freely away from $\overline{\theta}_i$. By carefully choosing $\lambda_i$ values, we can balance effectively between *overfitting* (too much deviation) and *underfitting* (too little deviation). Instead of choosing $\lambda_i$'s manually, our idea is to learn them automatically from data, through the principled Bayesian inference: given the evidence (i.e., the downstream data $D$) we infer the most probable values of $\lambda_i$'s that best explain the evidence, while being small enough to lead to sparse updates overall.

To this end, we regard $\lambda$ as random variables and impose hyper-prior on $\lambda$ (hierarchical Bayes). As we prefer sparse updates in the end, the hyper-prior needs to put higher mass/density on small $\lambda$ values. We adopt the Gamma distribution for this purpose, more specifically,

$$p(\lambda|\alpha, \beta) = \prod_{i=1}^{d} p(\lambda_i|\alpha, \beta) = \prod_{i=1}^{d} \mathcal{G}(\lambda_i; \alpha, \beta) \tag{2}$$

where $\mathcal{G}(x; \alpha, \beta) \propto x^{\alpha-1} \exp(-\beta x)$ for $\alpha, \beta > 0$. Note that although we can introduce individual $\lambda_i$-specific parameters (i.e., $(\alpha_i, \beta_i)$), we instead use a single $(\alpha, \beta)$ shared over all $\lambda_i$'s for simplicity. Since the mode of Gamma is 0 if $\alpha < 1$ and the variance is $\alpha/\beta^2$, we use small $\alpha < 1$ and large $\beta$ to enforce a priori small $\lambda$ values. In our experiments we choose[1] $(\alpha = 0.01, \beta = 100)$ for both the NLP and VTAB vision benchmarks.

Our full model can be written as the following joint distribution:

$$p(D, \theta, \lambda|\alpha, \beta) = \overbrace{p(\lambda|\alpha, \beta)}^{= \prod_i \mathcal{G}(\lambda_i; \alpha, \beta)} \times \underbrace{p(\theta|\lambda)}_{= \prod_i \mathcal{L}(\theta_i; \overline{\theta}_i, \lambda_i)} \times \overbrace{p(D|\theta)}^{= \prod_{(x,y) \in D} p(y|x, \theta)} \tag{3}$$

With this model, our ultimate goal is to infer the posterior distribution of the network weights and scales $p(\theta, \lambda|D)$, namely

$$p(\theta, \lambda|D, \alpha, \beta) = \frac{p(D, \theta, \lambda|\alpha, \beta)}{\iint p(D, \theta, \lambda|\alpha, \beta) \, d\theta d\lambda} \tag{4}$$

---

[1]We also tested with models with further hierarchy by placing priors on $\alpha$ and $\beta$, however, there was no significant advantage over the manually chosen ones.

We adopt the stochastic-gradient MCMC approach [40, 7, 4] to obtain samples from the posterior, especially the Langevin dynamic method (SGLD) [40], which amounts to running the following recurrence to collect posterior samples (after some burn-in steps):

$$[\theta, \lambda] \;\leftarrow\; [\theta, \lambda] + \frac{\eta}{2} \nabla \left( \log p(\lambda|\alpha, \beta) + \log p(\theta|\lambda) + \frac{\hat{N}}{|B|} \log p(B|\theta) \right) + \gamma \epsilon \sqrt{\eta} \qquad (5)$$

where $B \,(\subset D)$ is a minibatch, $\eta$ is small step size, and $\epsilon \sim \mathcal{N}(0, I)$. In standard SGLD, $\hat{N} = |D|$ (training data size) and $\gamma = 1$. However, we allow them to take values different from these default ones: $\hat{N}$ is the *effective* training data size to account for data augmentation[2] (e.g., if there are 5 different augmentation strategies and each allows 10 different variations, the effective data size can be $\hat{N} = 10^5 \times |D|$); the other hyperparameter $\gamma$ can be used to discount the noise effect[3], which is validated in the range $10^{-4:0}$ in our experiments. See Sec. 5.1 for more details. Note that in the parentheses subject to derivatives, the first two terms admit closed-form gradients while the gradient of the last term can be computed by the conventional SGD backprop. Thus each step in (5) is efficient, requiring at most only two times the complexity of the vanilla SGD step.

After a burn-in period, we can maintain those samples $(\theta, \lambda)$ to approximate $p(\theta, \lambda|D)$. For instance, the running average of the $\lambda$ samples, denoted by $\hat{\lambda}$, is a good estimate of the mean of the marginal posterior $p(\lambda|D)$. Since we imposed the hyper-prior that encourages small $\lambda$, ideally the majority of the scale parameters tend to remain small *a posteriori*, whereas there will be only a few large-valued $\hat{\lambda}_i$'s corresponding to those FM parameters that crucially need to be updated away from their pre-trained values to explain the data $D$. Roughly speaking, $\hat{\lambda}_i$s indicates how important it is for the corresponding FM parameters $\theta_i$s to be updated away from their pre-trained values $\bar{\theta}_i$ on the downstream data. Importantly, this $\hat{\lambda}$ can be used in the next stage (described in the next paragraph) to *decide* which network weights $\theta_i$ need to be frozen (those with small $\hat{\lambda}_i$s) and which need to be updated from the pre-trained (those with large $\hat{\lambda}_i$s).

**Thresholding scales and the second stage.** Although we adopted the sparsity-inducing Laplace prior, it is not necessarily the case that the majority of posterior $\theta$ values are sharply staying at the pre-trained values $\bar{\theta}$. We may need some thresholding: If $p$ is given, then we can directly take the top $(100 \times p)\%$ of them. Otherwise we examine for cut-off point – Those $i$'s with small posterior means $\hat{\lambda}_i$ are considered as frozen weights (i.e., $\theta_i = \bar{\theta}_i$), while those $i$'s with large $\hat{\lambda}_i$s can be treated as updatable parameters. In practice, we sort and plot the $\hat{\lambda}_i$s to eyeball and find a reasonable cut-off point. One can also use this $\hat{\lambda}$-plot to decide a reasonable sparsity level $p$ for good accuracy-memory trade-off (See Fig. 1 for illustration). Once we have decided which parameters are updated and frozen, we can run vanilla SGD to train the updatable parameters of the FM, which forms our second stage.

Our overall algorithmm dubbed **BayesTune**, is summarized as pseudocodes in Alg. 1.

# 4 Related Work

## 4.1 Comparison to Existing Sparse Fine-tuning Methods

The recent sparse/PEFT fine-tuning approaches broadly fall into the following three categories: 1) Heuristic search criteria, 2) Attaching small extra modules to the FM (e.g., adapter-based [15]), and 3) Directly formulating the sparse fine-tuning problem as an optimization problem.

**1) Heuristic search criteria.**
- **Random selection** – Randomly select $(100 \times p)\%$ parameters to update.

---

[2]Similar discussions of how to quantify dataset size when using Data Augmentation as well as data size inflation, were made in several prior Bayesian deep learning studies, for instance, [33, 34, 17].

[3]The rationale for noise discounting is as follows. Purely finding the posterior mean (i.e., SGLD without noise discount) risks performing poorly if the posterior is truly multi-modal, because it may converge to a low probability parameter. Also, purely searching for the posterior mode (i.e., MAP instead of SGLD) may be sensitive to data noise, because no stochasticity is taken into account properly. So for combining principle and practice it is reasonable to prefer a discounted noise procedure that balances between identifying a particular mode, but gets a mean estimate in the vicinity of that mode.

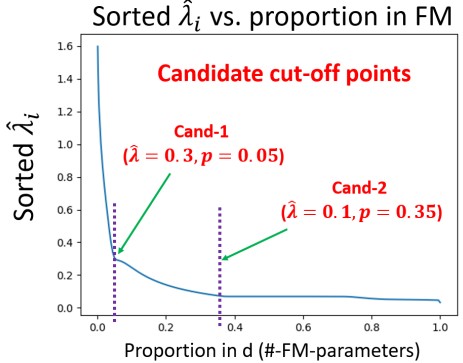
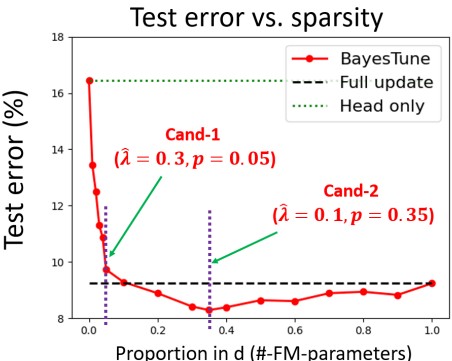

Figure 1: (**Left**) The $\hat{\lambda}$-plot in our BayesTune after stage-1 training. This is with EfficientNet-B0 on the Oxford-Pets dataset. After sorting the posterior mean $\hat{\lambda}_i$ values, we plot several sparsity levels $p$ and the corresponding $\hat{\lambda}$ values. To decide a reasonable sparsity level $p$ for good accuracy-memory trade-off, we illustrate two candidates: Cand-1 with more sparse updates ($p = 0.05$) but many parameters with large $\hat{\lambda}_i$'s (e.g., $0.1 \sim 0.3$) are forced frozen; Cand-2 with less sparse updates ($p = 0.1$) but ensures that those fixed parameters have indeed small $\hat{\lambda}_i$'s (e.g., $< 0.1$). (**Right**) Test errors in stage-2 at different cutoff points in our BayesTune (red curve). Those two candidate points (Cand-1 and Cand-2) are highlighted and superimposed, consistent with the left figure. The more sparse Cand-1 has slightly higher error than the full-update, while less sparse Cand-2 achives the best test performance among all other choices. It exhibits nice accuracy-memory trade-off. Further increasing $p$ (being far more dense) incurs overfitting.

---

**Algorithm 1** BayesTune (two-stage) sparse fine-tuning algorithm.

---

**Input:** Pre-trained model $\overline{\theta}$, downstream data $D$.
**Initialise:** $\theta = \overline{\theta}$, $\lambda_i = 0.0001$ for all $i$.
**I. Stage-1 (Select $(100 \times p)\%$ parameters to update)**
    For $t = 0, 1, 2, \ldots, T_{\max}$:
        0. Sample a minibatch $B = \{(x, y)\}$ from $D$.
        1. Run the SGLD step (5) to get a sample $(\theta, \lambda)$.
        2. If $t > $ burn-in, accumulate $\hat{\lambda} \leftarrow (1 - \zeta)\hat{\lambda} + \zeta\lambda$ where $\zeta = 1/(\text{\#-accumulated-samples}+1)$.
    End-For
    (If $p$ not given) Decide the cut-off point $\lambda_{th}$ and the corresponding sparsity level $p$ using the $\hat{\lambda}$-plot.
    Select those $i$'s with $\hat{\lambda}_i > \lambda_{th}$ as *updatable* parameters.
**II. Stage-2 (Update the selected parameters)**
    Do vanilla SGD with data $D$ while updating only those *updatable* parameters from Stage-1.

---

- **MixOut [23]** – After training the FM model with full update, a random $(100 \times (1 - p))\%$ parameters are reset to the pre-trained values.

- **BitFit [43]** – Update only the bias parameters with the rest frozen.

- **MagPruning [14, 24, 22]** – Take the largest $(100 \times p)\%$ in magnitude as updatable parameters.

Compared to our proposed "BayesTune", these heuristic search methods are less principled, unable to explain why the resulting models should perform well on the downstream tasks

**2) Approaches that attach small extra modules to FM.** They do not search for a subset of FM parameters to update, but attach some small extra modules to the FM (e.g., insert a small MLP module after each layer). All of the FM parameters are frozen, and they only train those attached modules. So it can be seen as sparse fine-tuning considering that the size of the new modules is relatively small compared to that of the FM.

- **Adapter [15]** – A small bottleneck MLP is inserted at the end of each layer of the FM.

- **LoRA [16]** – A low-rank projection weight matrix is inserted before the query/key/value embeddings in each layer.

- **Prompt learning (VPT) [18]** – A set of learnable tokens is inserted in the input of a Transformer attention block in each layer.

- **Neural Prompt Search [45]** – A NAS-based approach (e.g., learning a supernet followed by evolutionary search for a downstream tasks specific subnet) that aims to search for the optimal combination of the above three methods.

Compared to our "BayesTune", these adapter-based models involve the difficult problem of where to place the extra modules and what would be the optimal module dimensions/sizes, which is essentially a very difficult combinatorial search problem, often requiring heuristic search methods like the evolutionary search.

**3) Directly optimizing the sparse optimization problem.** One can formulate the sparse fine-tuning as an optimization problem, for instance,

$$\min_{\theta} \ \mathbb{E}_{(x,y)\sim D}[l(\theta; x, y)] \ \text{ s.t. } \ ||\theta - \overline{\theta}||_0 \leq \lfloor d \cdot p \rfloor. \tag{6}$$

Unfortunately this problem becomes a discrete optimization due to the $L_0$ cardinality constraints, thus difficult to solve in general. Some recent approaches aimed to relax the problem to continuous optimization, but relying on heavy approximation.

- **DiffPruning [13]** – They relax the discrete problem into a continuous one by introducing Bernoulli variables indicating update or freeze, while the reparametrized sampling is further approximated by the Gumbel-softmax treatment.
- **ChildPruning [41, 31]** – They iteratively train the full model parameters and calculate the projected mask to find the child network.
- **Second-order approximation method (SAM) [10]** – They adopt several steps of second-order Taylor approximation of the loss function, leading to a very succinct parameter selection strategy: after computing the gradients of $\theta$ on the data $D$, take the $(100 \times p)\%$ parameters with the largest gradient magnitudes.

Compared to our "BayesTune", these direct optimization strategies rely on strong assumptions and heavy approximation to relax the difficult discrete optimization.

## 4.2 Relation to Existing (Hierarchical) Bayesian Sparse Model Learning

(Hierarchical) Bayesian models are well known in statistical machine learning, and there were some prior works that adopt the Bayesian approaches for sparse deep learning [21, 32, 35, 29]. However, as far as we know, our approach has several key differences from these previous works.

First, most prior works are all about sparse training, instead of *sparse fine-tuning*. Thus they focus on zeroing out many parameters, instead of retaining the pre-trained weights. To the best of our knowledge, our BayesTune is the first to tackle the sparse fine-tuning at the Foundation Model scale using the hierarchical Bayesian framework.

More importantly, the neural networks used in those previous studies are rather small/toy scale (mostly focusing on MLPs and LeNet-sized architectures up to ResNet-18 at the largest) while our method can obtain the state-of-the-art results on large-scale foundation models (ViT, RoBERTa). For example the largest model considered in those prior works, ResNet-18 contains $\sim 11$M parameters vs. RoBERTa's $\sim 123$M parameters, a $10\times$ scale difference. The reason is that they adopt methods that entail extra memory cost like variational inference (VI), which impedes applicability to big networks. We have also done some experiments that compare the computational resources required by VI and SGLD: On ViT networks, the training time is increased by 1.7 times if we replace SGLD by VI; the GPU memory footprint is increased by 2.1 times.

## 5 Experiments

We test our BayesTune on two popular benchmark datasets from NLP and vision for the downstream fine-tuning tasks: (**language**) fine-tuning the pre-trained RoBERTa-base model [28] on the GLUE [37] and SuperGLUE [38] tasks; (**vision**) fine-tuning the ImageNet-22K [6] pre-trained ViT-B/16 model [9] on VTAB-1K [44] image classification/prediction tasks. The details of the experimental settings are discussed in the subsequent sections, Sec. 5.2 and Sec. 5.3.

### 5.1 Implementation of BayesTune

For the SGLD sampling in Stage-1 of our BayesTune algorithm (re: Alg. 1), we have three different training regimes: the *warm-up* phase that only runs vanilla SGD steps without considering the scale

| Method | CoLA | STS-B | MRPC | RTE | CB | COPA | WSC | AVG |
|---|---|---|---|---|---|---|---|---|
| Full update | $58.36^{1.74}$ | $89.80^{0.52}$ | $89.55^{0.81}_{[2]}$ | $76.03^{2.14}$ | $88.93^{2.37[3]}_{[3]}$ | $67.70^{4.41}$ | $53.10^{6.18}$ | $74.78^{2.60}$ |
| Random | $58.35^{1.05[3]}$ | $89.81^{\mathbf{0.11}[1]}$ | $88.73^{0.80}$ | $72.71^{3.23}$ | $90.54^{3.39}_{[2]}$ | $68.80^{2.64}$ | $52.88^{5.97}$ | $74.55^{2.46}$ |
| MixOut | $58.66^{1.96}$ | $90.15^{0.17}_{[4]}$ | $88.69^{0.60}_{[4]}$ | $77.55^{1.64[2]}_{[2]}$ | $86.51^{4.13}$ | $71.30^{4.84}$ | $52.98^{6.78}$ | $75.12^{2.88}_{[4]}$ |
| Bitfit | $56.67^{1.45}$ | $90.12^{0.14}_{[3]}$ | $87.35^{0.58}_{[3]}$ | $72.74^{2.47}$ | $86.96^{3.20}$ | $71.20^{3.79}$ | $55.10^{5.39}$ | $74.31^{2.43}$ |
| MagPruning | $56.57^{2.47}$ | $90.30^{0.14}_{[3]}$ | $88.09^{0.79}$ | $73.53^{1.84}_{[4]}$ | $81.25^{3.50}$ | $71.50^{2.46[2]}_{[4]}$ | $55.67^{2.73[2]}$ | $73.85^{1.99[3]}$ |
| Adapter | $\mathbf{62.11}^{1.22[4]}_{[1]}$ | $90.05^{0.13}_{[2]}$ | $89.29^{0.60}_{[4]}$ | $76.93^{2.05}_{[4]}$ | $87.32^{4.62}$ | $69.50^{2.54}_{[4]}$ | $57.02^{5.27}_{[3]}$ | $76.03^{2.35}_{[3]}$ |
| LoRA | $60.88^{1.48}_{[3]}$ | $87.19^{0.51}$ | $89.53^{0.62}_{[3]}$ | $76.97^{1.92}_{[3]}$ | $84.64^{3.76}$ | $69.70^{2.83}$ | $56.84^{4.52}_{[4]}$ | $75.11^{2.24}_{[4]}$ |
| DiffPruning | $58.53^{1.49}$ | $89.59^{0.34}$ | $78.79^{6.09}$ | $69.93^{7.87}$ | $86.25^{2.65}_{[4]}$ | $72.10^{2.91}_{[3]}$ | $53.37^{3.60}_{[4]}$ | $72.65^{3.57}$ |
| ChildPruning | $60.00^{1.29}$ | $89.97^{1.51}$ | $87.19^{3.86}$ | $75.76^{4.38}$ | $86.61^{3.22}$ | $69.40^{4.00}$ | $55.59^{3.81}$ | $74.93^{3.15}$ |
| SAM | $60.89^{0.96[2]}_{[2]}$ | $\mathbf{90.59}^{0.14[3]}_{[1]}$ | $88.84^{\mathbf{0.49}}_{[1]}$ | $76.79^{1.72}_{[3]}$ | $88.93^{1.75[2]}_{[3]}$ | $74.30^{\mathbf{2.45}[1]}_{[2]}$ | $59.52^{3.08[3]}_{[2]}$ | $77.12^{1.51[2]}_{[2]}$ |
| **BayesTune** | $60.85^{\mathbf{0.47}[1]}_{[4]}$ | $90.40^{0.14[3]}_{[2]}$ | $\mathbf{90.61}^{0.56[2]}_{[1]}$ | $\mathbf{77.87}^{\mathbf{0.64}[1]}_{[1]}$ | $\mathbf{91.25}^{\mathbf{1.25}[1]}_{[1]}$ | $\mathbf{75.00}^{2.49[3]}_{[1]}$ | $\mathbf{60.87}^{\mathbf{2.62}[1]}_{[1]}$ | $\mathbf{78.12}^{\mathbf{1.17}[1]}_{[1]}$ |

Table 1: Results on NLP benchmarks. For each dataset/task (column), the average accuracy and the standard deviation (in superscript) shown over 10 runs with different random seeds are reported. The ranks (up to the fourth) among the competing methods are also shown in the brackets and in red. The figures of the competing methods are excerpted from [10].

variables $\lambda$ and random noise; followed by the *burn-in* phase where the SGLD steps are performed as (5) but no $(\theta,\lambda)$ samples are collected; followed by the *normal* phase in which we do collect samples. Following the conventional practice, we also perform the *thinning* steps [27] (i.e., collecting samples at a certain frequency) to mitigate undesired temporal correlation effects.

Another important implementation tip is the use of *reweighed cost terms* in Langevin dynamics – As shown in (5), the cost function is involved with the training data size $N = |D|$. Even though $N$ is known, the data augmentation would considerably increase the *effective* data size by several orders of magnitudes (e.g., if there are 5 different augmentation strategies and each allows 10 different variations, the effective data size can be $10^5 \times N$). To account for it, we regard the effective data size as a hyperparameter denoted by $\hat{N}$ that can be chosen from validation[4]. Replacing $|D|$ by $\hat{N}$ and taking it out of the parentheses in (5), we have the gradient (normalized by $\hat{N}$) multiplied by the step size $(\eta/2)\hat{N}$. First we choose $\eta$ to match the learning rate in the neural net optimizer (e.g., Adam[5] [20]), that is, $\eta = lr \times 2/\hat{N}$. Accordingly the noise term becomes $\epsilon \cdot (2/(\hat{N} \times lr))^{1/2}$. We also find it effective oftentimes to discount the noise effect: we introduce the noise discount factor $\gamma$ (another hyperparameter) by which the noise term is multiplied. We validate $\gamma$ in the range $10^{-4:0}$ (e.g., $\gamma = 1$ corresponds to the default no-discount case).

Throughout all experiments we use the Gamma prior parameters ($\alpha = 0.01, \beta = 100$), the scale variables $\lambda_i$'s are all initialized to 0.0001, and the learning rate for $\lambda$ is 0.01 without scheduling. Other task-specific implementation details can be found in the subsequent sections. *The Python/PyTorch code to reproduce the results is available at* `https://github.com/SamsungLabs/BayesTune`[6].

## 5.2 NLP Benchmarks

We consider fine-tuning the pre-trained RoBERTa-base model [28], the large-scale foundation language model comprised of 125 million parameters, on several downstream tasks in GLUE [37] and SuperGLUE [38] benchmarks. We follow the experimental settings from [10], in which the original development sets serve as test sets, and the validation sets are formed by holding out random 10% of

---

[4]For instance, in the VTAB-1K, although $N = 1000$ or 800, we let $\hat{N}$ have a range $10^{7:12}$.

[5]We used the Adam optimiser for updating the model parameters, thus there may exist some internal gradient adaptation and momentum effect under the hood in our SGLD steps. This effect may be related to that of the adaptive drift and momentum in SGLD that were analysed in some previous works, e.g., [19]. However, we believe that our SGLD update scheme with Adam would not result in significantly different solution compared to the original SGLD formulation at least in practice.

[6]Alternatively, `https://github.com/minyoungkim21/BayesTune`

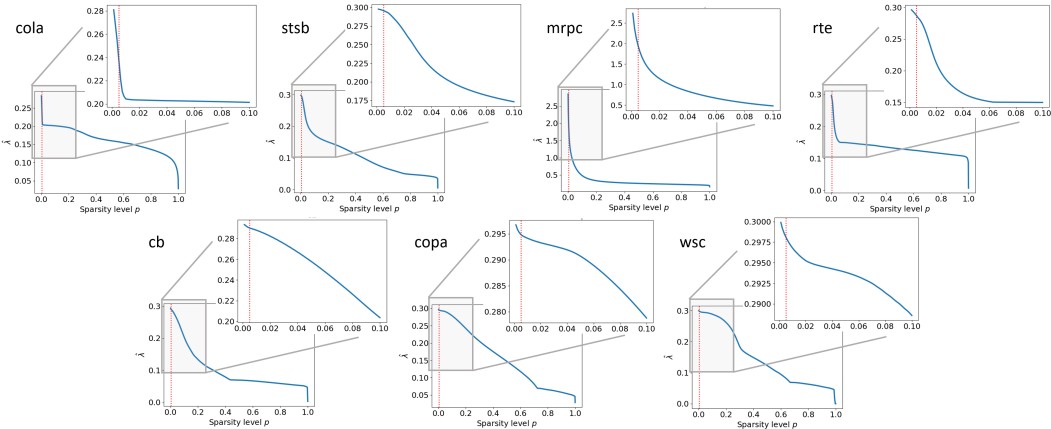

Figure 2: (NLP benchmarks) Posterior mean $\hat{\lambda}$ vs. sparsity level $p$. The default sparsity level $p = 0.005$ is depicted as red dotted vertical lines. For each dataset, we plot the original $p \in [0, 1]$ range (lower-left), which is magnified for the smaller range $p \in [0, 0.1]$ (upper-right).

the training sets. Instead of having a fixed number of training epochs, the development sets are used for an early stopping with the no-improvement tolerance of 40 epochs. We implement our BayesTune on top of the codebase of [10] which is based on the `jiant` framework (`https://jiant.info/`). Similarly as [10], we consider 7 tasks: Corpus of Linguistic Acceptability (**CoLA**) [39], Semantic Textual Similarity Benchmark (**STS-B**) [2], Microsoft Research Paraphrase Corpus (**MRPC**) [8], Recognizing Textual Entailment (**RTE**) [5, 11, 1], Commitment Bank (**CB**) [30], Choice of Plausible Alternatives (**COPA**) [36], and Winograd Schema Challenge (**WSC**) [25].

Our BayesTune is compared against strong baselines and the state-of-the-arts, including: **Random**, **Mixout** [23], **BitFit** [43], **MagPruning** [24, 22], **Adapter** [15], **LoRA** [16], **DiffPruning**, **Child-Pruning** [41, 31], and **SAM** [10], as discussed in Sec. 4. For all methods, we use the sparsity level $p = 0.005$ as suggested in [13]. For the competing methods, we follow the hyperparameter settings from [10]; In BayeTune, we use $10K$ warm-up steps, $2K$ burn-in steps, and thinning at every $100$ steps for all tasks. The batch size is 16, and the learning rate for the model parameters is $10^{-4}$ for Stage-1 and $10^{-3}$ for Stage-2. For the (dataset-dependent) evaluation metrics, we adopt the protocols in [37, 38]. We run all models on a single NVIDIA V100 GPU with 32GB memory. The results are summarized in Table 1 where means and standard deviations over 10 random seeds are reported.

First, it is quite surprising that the random selection strategies (**Random** and **Mixout**) are as effective as many sparse fine-tuning methods. This may be attributed to the massive over-parametrization of the deep foundation model. However, more careful and sophisticated selection is important as those three methods, our **BayesTune**, **SAM**, and **Adapter**, outperform the random strategies by large margin. Overall our **BayesTune** performs the best among the competing methods, about $1\%$ point higher on average than the runner-up **SAM** [10]. We achieve the first place on 5 out of 7 tasks, and also attain the smallest standard deviations, implying that the resulting sparse fine-tuned models are more robust and reliable. This result signifies the effectiveness of our sparse Laplace scale-based parameter selection strategy that is underpinned by the principled Bayesian posterior inference with efficient[7] stochastic gradient Langevin sampling.

In Fig. 2, we also show the posterior mean scale $\hat{\lambda}$ vs. sparsity level $p$. The cut-off threshold points corresponding to the sparsity level $p = 0.005$ are around $\lambda_{th} \in [0.25, 0.30]$ for all tasks except for **MRPC**. This cut-off scale range might be considered as reasonable choice as most of the parameters would have Laplace scales smaller than this range, exhibiting no pronounced indication of crucial deviation from the pre-trained values, although this is only conjecture based on our empirical results. Further rigorous theoretical analysis on the relation between the scale threshold and the generalization performance is beyond the scope of the current paper, and is left as future work.

### 5.3 Vision Benchmarks

Next we test our BayesTune on the VTAB-1K [44], an image classification/prediction task adaptation benchmark suite comprised of 19 different image datasets. It contains images/tasks that exhibit

---

[7]The SGLD steps (5) require at most only two times the complexity of the vanilla SGD steps.

| Method | #param (M) | Cifar100 | Caltech101 | DTD | Flower102 | Pets | SVHN | Sun397 | Camelyon | EuroSAT | Resisc45 | Retinopathy | Clevr-Count | Clevr-Dist | DMLab | KITTI | dSpr-Loc | dSpr-Ori | sNORB-Azim | sNORB-Ele | Avg Rank (↓) | # Rank 1 (↑) |
|---|---|---|---|---|---|---|---|---|---|---|---|---|---|---|---|---|---|---|---|---|---|---|
| Full update | 85.8 | 68.9 | 87.7 | 64.3 | 87.2 | 86.9 | 87.4 | 38.8 | 79.7 | 95.7 | 84.2 | 73.9 | 56.3 | 58.6 | 41.7 | 65.5 | 57.5 | 46.7 | 25.7 | 29.1 | | |
| Linear | 0.04 | 64.4 | 85.0 | 63.2 | 97.0 | 86.3 | 36.6 | 51.0 | 78.5 | 87.5 | 68.5 | 74.0 | 34.3 | 30.6 | 33.2 | 55.4 | 12.5 | 20.0 | 9.6 | 19.2 | | |
| VPT [18] | 0.64 | **78.8** | 90.8 | 65.8 | 98.0 | 88.3 | 78.1 | 49.6 | 81.8 | **96.1** | 83.4 | 68.4 | 68.5 | 60.0 | 46.5 | 72.8 | 73.6 | 47.9 | **32.9** | 37.8 | 4.16 | 3 |
| Adapter [15] | 0.16 | 69.2 | 90.1 | 68.0 | 98.8 | 89.9 | 82.8 | **54.3** | 84.0 | 94.9 | 81.9 | 75.5 | 80.9 | 65.3 | 48.6 | 78.3 | 74.8 | 48.5 | 29.9 | 41.6 | 3.68 | 1 |
| LoRA [16] | 0.29 | 67.1 | 91.4 | 69.4 | 98.8 | 90.4 | 85.3 | 54.0 | **84.9** | 95.3 | **84.4** | 73.6 | **82.9** | **69.2** | 49.8 | 78.5 | 75.7 | 47.1 | 31.0 | 44.0 | 2.68 | 4 |
| NOAH [45] | 0.43 | 69.6 | **92.7** | **70.2** | **99.1** | 90.4 | 86.1 | 53.7 | 84.4 | 95.4 | 83.9 | 75.8 | 82.8 | 68.9 | **49.9** | 81.7 | 81.8 | 48.3 | 32.8 | **44.2** | **1.95** | 5 |
| **BayesTune** | Avg 0.37 | 68.9 (.07) | 92.6 (.37) | 69.5 (.04) | **99.1** (.37) | **90.8** (.15) | **88.1** (.67) | 50.0 (.04) | 84.6 (.60) | 95.8 (.60) | 82.8 (.37) | **76.0** (.07) | 82.6 (.30) | 67.4 (.22) | 49.6 (.52) | **82.3** (.60) | 81.9 (.60) | **49.9** (.30) | 22.6 (.52) | 39.3 (.67) | 2.37 | **7** |

Table 2: VTAB-1K results. The accuracies at the optimal sparsity levels are reported for our BayesTune. For BayesTune, the optimal number of the updated parameters is dataset-dependent, and these optimal numbers are depicted in the parentheses. The figures of the competing methods are exerpted from [18, 15, 16, 45].

highly diverse aspects/conditions such as: different image acquisition (by standard cameras or special-purpose ones for remote sensing or medical imaging), different objects/concepts (generic, fine-grained, or abstract), and tasks (object recognition, counting, or pose estimation). Each dataset in VTAB-1K consists of $1K$ training examples, and we use the splits officially provided (train $80\%$ and validation $20\%$).

We aim to fine-tune the ImageNet-22K [6] pre-trained ViT-B/16 model [9] on each dataset. We implement our BayesTune using the codebase from [45] while employing most of their hyperparameters without changes (e.g., the number of training epochs 100 and batch size 64). In our BayesTune, we use 50 warm-up epochs, 10 burn-in epochs, and thinning at every 5 batch steps for all datasets. The learning rate for the model parameters is $10^{-3}$ and weight decay $10^{-3}$ for both stages.

We first attempted the proposed method with our original setting, i.e., place the $\lambda$ variables to the backbone (ViT) parameters, and let the Bayesian inference figure out the most critical parameters to update, namely those with large $\lambda_i$'s. Unfortunately the result was significantly worse than the adapter-based methods: VPT [18], Adapter [15], LoRA [16], and NOAH [45], for most datasets. As reported in several works in the literature, adapter-based models perform significantly better than the full backbone update for this benchmark (e.g., Table 1 in [45]). Along this line, we conjecture that this trend also holds for other sparse fine-tuning methods, namely the superiority of the adapter-based approaches to sparse backbone update strategies, particularly for this VTAB benchmark.

To fairly take the advantage of extra attachable modules as the adapter-based approaches, we attach extra modules to ViT for all three popular strategies: visual prompts in VPT [18], adapter modules in Adapter [15], and low-rank matrices in LoRA [16]. This design idea is similar to (and motivated from) the neural prompt search in NOAH [45], but they formed a grid search problem to determine which layers to attach the adapter modules and how many module dimensions to attach – solved by the neural architecture search (NAS) technique with the super-net training, specifically adopting Autoformer [3], followed by running Evolutionary algorithms for optimal architecture search.

On the other hand, we add 10-dim full parameters for each module/layer, and they are all initialized to 0. We let the posterior inference in our BayesTune automatically figure out which of these extra modules parameters to be updated away from 0 and which to be frozen at 0. We used the zero-initialization because having all-0 extra modules can be considered to be identical to the original pre-trained ViT *without extra modules*, thus serving as *pre-trained* extra modules. We only place $\lambda$ variables to the extra modules (ViT parameters frozen)[8]. In summary, our setup is similar to NOAH [45] in that we take all three adapter strategies into account, but the main difference is: NOAH does a difficult super-net training and Evolutionary search to determine the optimal module-wise dimensions. On the other hand we do sparse selection of the module parameters to be updated from 0 across all 10-dim full modules.

---

[8]We initially attempted to place $\lambda$ to ViT parameters as well, but unfortunately it did not perform well. The reason being probably is that most of the selected parameters for update come from the ViT backbone, essentially reducing to the original sparse backbone update strategy that failed.

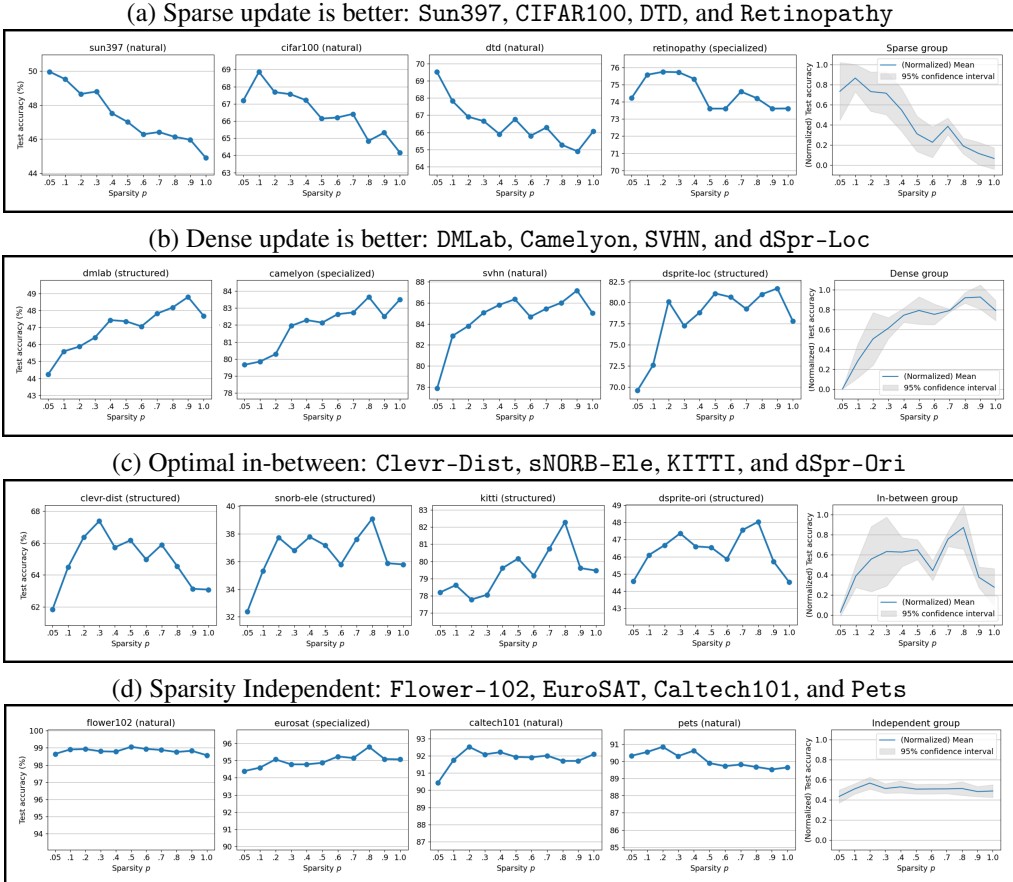

Figure 3: VTAB datasets grouped by the sparse/dense adapter update effects. E.g., the datasets in group-(a) exhibit higher accuracy when the adapter modules are updated more sparsely than densely. In groub-(c), neither extreme sparse nor dense updates are optimal, but the sparsity levels in between perform the best. In group-(d), we have no clear correlation between update sparsity and test accuracy. Individual plots show test accuracies vs. sparsity levels, where the rightmost plot in each group shows the mean and 95% confidence interval of the members within the group in normalized accuracy scale.

The results are summarized in Table 2. We report the performance of the best sparsity levels found from $p \in \{0.05, 0.1, 0.2, \ldots, 0.9, 1.0\}$. Although the final accuracies of our BayesTune are not always the best, we achieve Rank 1 on 7 out of 19 datasets and the second place on average over all datasets. The performance of BayesTune is comparable to other adapter-based methods, within the same range as those. That is, our main message is that *the adapter-based enhanced architecture fine-tuning can be done almost comparably well by the proposed Bayesian method, instead of relying on the heuristic search such as the Evolutionary Search or costly human expert design effort.* Compared to NOAH, our BayesTune results in more sparse adapter updates on average (0.38M vs. 0.43M). In Fig. 3 we also visualize the clustering of VTAB-1K datasets according to the sparse/dense adapter update effects: we assign each dataset into either of *sparse*, *dense*, and *in-between* clusters according to the optimal sparse update level $p$: small, large, medium, respectively. Such a grouping of the VTAB datasets based on fine-tuning sparsity characteristics has not been explored in the community, and we can easily obtain this finding using our BayesTune.

## 6 Conclusion

We introduced BayesTune, a framework for automated fine-tuning that provides a principled yet efficient approach to identifying which parameters to update in sparse fine-tuning by posterior inference of parameter-wise scale in a hierarchical Bayesian model. We demonstrated BayesTune to be effective for parameter efficient fine-tuning in both NLP and vision tasks, using both mask-based and additive search spaces.

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

# Appendix

## A  Chosen Hyperparameters

We grid-search hyperparameters on validation, where the two key hyperparameters are: the effective training data size $\hat{N}$ and the noise discount factor $\gamma$ (re: Sec. 5.1). The candidate sets are formed as: $\hat{N} \in \{10^8, 10^9, \cdots, 10^{12}\}$, $\gamma \in \{10^{-4}, 10^{-2}, 10^0\}$ for NLP, and $\hat{N} \in \{10^6, 10^7, \cdots, 10^{12}\}$, $\gamma \in \{10^{-4}, 10^{-3}, \cdots, 10^0\}$ for VTAB. The chosen hyperparameters are as follows $(\hat{N}, \gamma)$: (NLP) cola $= (11, 10^{-4})$, stsb $= (12, 10^{-4})$, mrpc $= (12, 10^0)$, rte $= (8, 10^{-4})$, cb $= (10, 10^{-4})$, copa $= (8, 10^{-2})$, wsc $= (10, 10^{-4})$; (VTAB) cifar100 $= (7, 10^{-1})$, caltech101 $= (9, 10^{-2})$, dtd $= (12, 10^0)$, flower102 $= (12, 10^{-2})$, pets $= (12, 10^0)$, svhn $= (10, 10^0)$, sun397 $= (7, 10^{-1})$, camelyon $= (6, 10^0)$, eurosat $= (7, 10^{-1})$, resisc45 $= (12, 10^{-2})$, retinopathy $= (7, 10^{-2})$, clevr-count $= (7, 10^{-3})$, clevr-dist $= (7, 10^{-3})$, dmlab $= (8, 10^0)$, kitti $= (7, 10^0)$, dsprite-loc $= (12, 10^{-4})$, dsprite-ori $= (12, 10^{-3})$, snorb-azim $= (7, 10^{-2})$, snorb-ele $= (6, 10^{-1})$.

## B  More Analysis

**(NLP) Test accuracies at other sparsity levels.** Although $p = 0.005$ is recognized as the optimal sparsity level overall for the GLUE and SuperGLUE tasks, we evaluate the test performance of our BayesTune for different sparsity levels: $p \in \{0.01, 0.05, 0.1, 0.2, 0.3, 0.5\}$. The average test accuracies are shown in Fig. 4. We see that overall there is less significant change in test performance so long as the sparsity level $p$ is small enough, and the resulting sparse updates selected by our BayesTune lead to equally good performance as those with the default value. However, increasing $p$ further (e.g., $p = 0.5$) considerably degrades the performance, which signifies the importance of sparse fine-tuning to avoid potential overfitting.

**(VTAB) Scale posterior mean $\hat{\lambda}$ vs. sparsity level $p$.** We visualize the plots that relate the sorted scale posterior means $\hat{\lambda}$ to the sparsity levels $p$ in Fig. 5. The plots are aligned with the the test accuracy plots analyzed in the main paper. As the plots are grouped along the optimal sparsity values, we see certain trends: for the *sparse group* (`sun397` and `cifar100`), the scale $\hat{\lambda}$ values are overall small scaled (in the range of $[0, 0.2]$) with sharp drops at small $\hat{\lambda}$; for the *dense group*: (`camelyon` and `dmlab`), $\hat{\lambda}$ scale is even larger (in the range of $[0, 0.5]$) with relatively smooth decaying at small values; lastly for the *in-between group* (`clever-dist`, `dspr-ori`, `kitti`, and `snorb-ele`), we have much narrower $\hat{\lambda}$ ranges in between 0 and 0.1 except for `kitti`.

## C  Some Ablation Study

**MAP vs. SGLD.** The Maximum-A-Posterior (MAP) aims to find a mode of the posterior distribution $p(\lambda|D)$, and might be considered as an alternative solution to our SGLD posterior samples. In particular, MAP might be attained by turning off the noise term in (5), that is, by setting $\gamma = 0$. However, MAP can be more sensitive to data noise than the mean of the posterior. We have some empirical comparison between MAP and SGLD on the NLP tasks in Table 3, and it shows that this distinction does lead to empirical benefit. Another benefit of SGLD

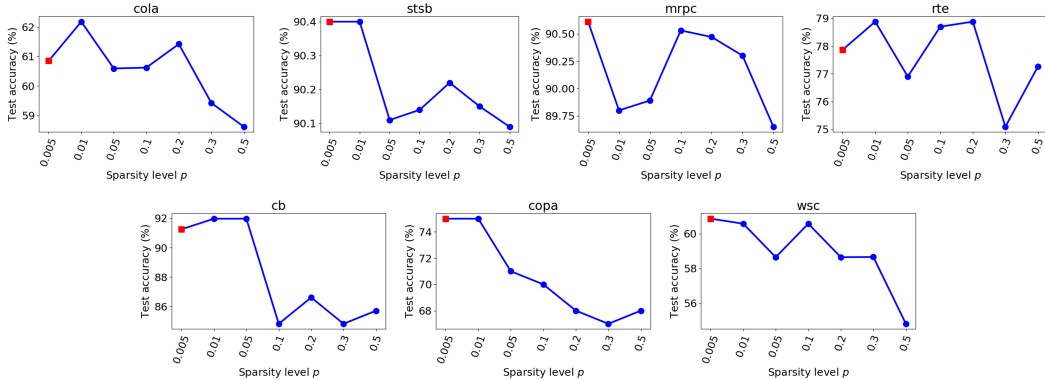

Figure 4: (NLP benchmarks) Test accuracies at sparsity levels other than the default $p = 0.005$. We evaluate the BayesTune sparse update models with $p \in \{0.01, 0.05, 0.1, 0.2, 0.3, 0.5\}$, where the default ones $p = 0.005$ are shown as red square markers.

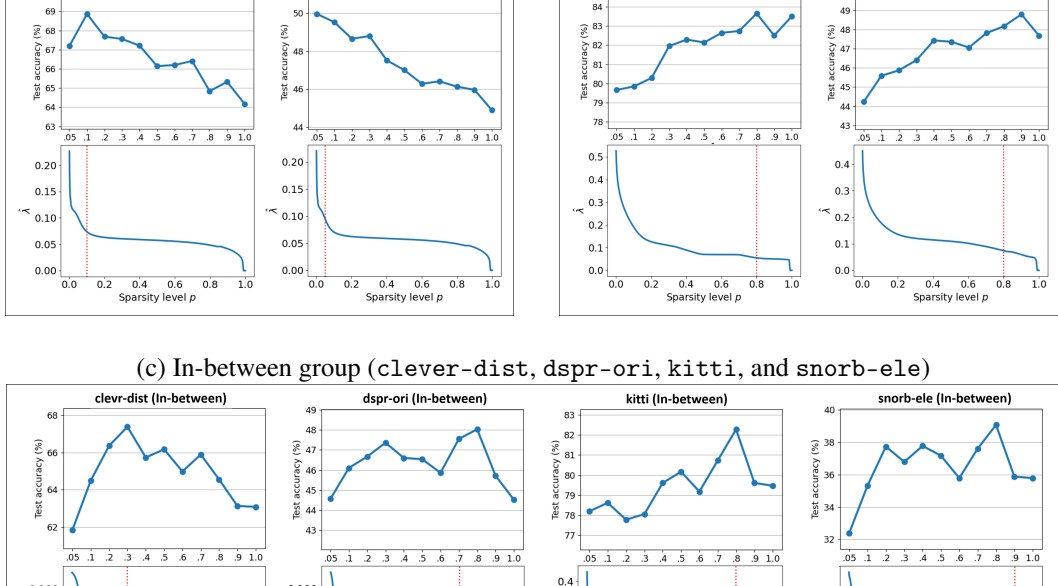

Figure 5: (VTAB benchmarks) The plots of the sorted scale posterior means $\hat{\lambda}$ vs. sparsity levels $p$, each of which is aligned with the the corresponding test accuracy plot. The plots are grouped along the optimal sparsity values where each group exhibits similar trends.

| Method | CoLA | STS-B | MRPC | RTE | CB | COPA | WSC | AVG |
|---|---|---|---|---|---|---|---|---|
| MAP | 58.55 | 90.13 | 90.54 | 76.72 | 86.71 | 71.09 | 60.34 | 76.30 |
| SGLD (Ours) | 60.85 | 90.40 | 90.61 | 77.87 | 91.25 | 75.00 | 60.87 | 78.12 |

Table 3: MAP vs. SGLD results on NLP benchmarks. Averaged over 10 random runs.

is that we can also exploit the variance of the posterior in parameter selection (e.g., for two parameters with similar posterior mean values, we prefer to select the one with smaller posterior variance). In future algorithms, we can also exploit this idea of variance-based weight pruning.

**Comparison to L1 regulariser.** The recent L1-regularised sparse model learning (L1-SP) [42] used the L1 regularisation term to encourage the updated weights to stay close to the original pre-trained weights. Thus this model might have a similar effect as our BayesTune. The L1-SP can provide an alternative regularisation-based approach to fine-tuning, however, is designed without the principled Bayesian modeling solution, which is the main difference from the proposed BayesTune. We have conducted some extra comparison experiments between the two approaches. Note that L1-SP does not necessarily lead to exactly sparse solutions. Therefore we offer a two-stage extension of L1-SP. in the first stage, we run the L1-SP training, and in the second stage those weights to be updated are selected based on the relative L1 distances from the pre-trained weights (i,e., taking those

| Method | CoLA | STS-B | MRPC | RTE | CB | COPA | WSC | AVG |
|---|---|---|---|---|---|---|---|---|
| L1-SP (stage I) | 50.50 | 88.07 | 84.54 | 50.00 | 62.44 | 60.40 | 52.88 | 64.12 |
| L1-SP (stage II, $(100 \times p)\%$) | 54.59 | 88.11 | 89.25 | 68.85 | 81.55 | 70.75 | 55.38 | 72.64 |
| BayesTune (stage II, $(100 \times p)\%$) | 60.85 | 90.40 | 90.61 | 77.87 | 91.25 | 75.00 | 60.87 | 78.12 |

Table 4: L1-SP [42] vs. BayesTune results on NLP benchmarks. Averaged over 10 random runs.

$(100 \times p)\%$ weights with the largest relative changes from the pre-trained weights). The results on the NLP tasks are shown in Table 4.

For the L1 penalty balancing constant hyperparameter, we choose optimal values by grid search from $\{10^{-3}, 10^{-4}, 10^{-5}, 10^{-6}\}$. We can see that L1-SP considerably lags behind our BayesTune, which is mainly attributed to its failure to capture uncertainty in L1 regularisation, thus being potentially sensitive to noise in data (a similar reason as the MAP estimate). Moreover, only penalising the parameters deviation from the pre-trained weights as in Stage 1 significantly underperforms the sparse cut-off strategy in Stage 2, signifying that sparse update is critical.

## D  Layer-wise and Module-wise Sparsity Patterns of BayesTuned Networks

**Sparsity patterns of RoBERTa-base on NLP tasks.** We visualize the module-wise and layer-wise sparsity patterns of the BayesTuned RoBERTa-base networks on 7 NLP tasks in Fig. 6–19. First, for the layer-wise sparsity pattern: (Except for `mrpc`) The proportions of the selected updatable parameters are more or less uniformly distributed across the 12 layers of the Transformer, while the first word embedding layer and the last classification layer are significantly less and more selected, respectively. This is intuitively appealing as the task-specific features may tend to be determined at the higher, more global levels in texts/sentences, to account for longer-range dependency. Next, looking at the module-wise sparsity patterns, the proportions are highly non-uniform, layer-specific, and also task/dataset-dependent. For instance, the bias modules in some layers are very densely selected, while they are very sparsely selected in other layers. This shows clear discrepancy to the heuristic strategies like BitFit [43] in which the bias modules are selected 100% for all layers.

**Sparsity patterns of ViT-B/16 on VTAB vision tasks.** The module-wise and layer-wise sparsity patterns of the BayesTuned ViT-B/16 networks on VTAB benchmark datasets are shown in Fig. 20–38. We also superimpose the optimal $p$ values (dataset dependent). The resulting patterns are quite similar to the NLP case: Except for a few cases, the lowest level visual prompt layers are selected far less, sometimes ignored, compared to the later layers. The last linear classification head, although not shown here in the sparsity diagrams, is selected 100%. Overall the layer-wise selection patterns are nearly uniform while the module-wise selection patterns are highly non-uniform and dataset dependent.

cola (**Module-wise** sparsity pattern)

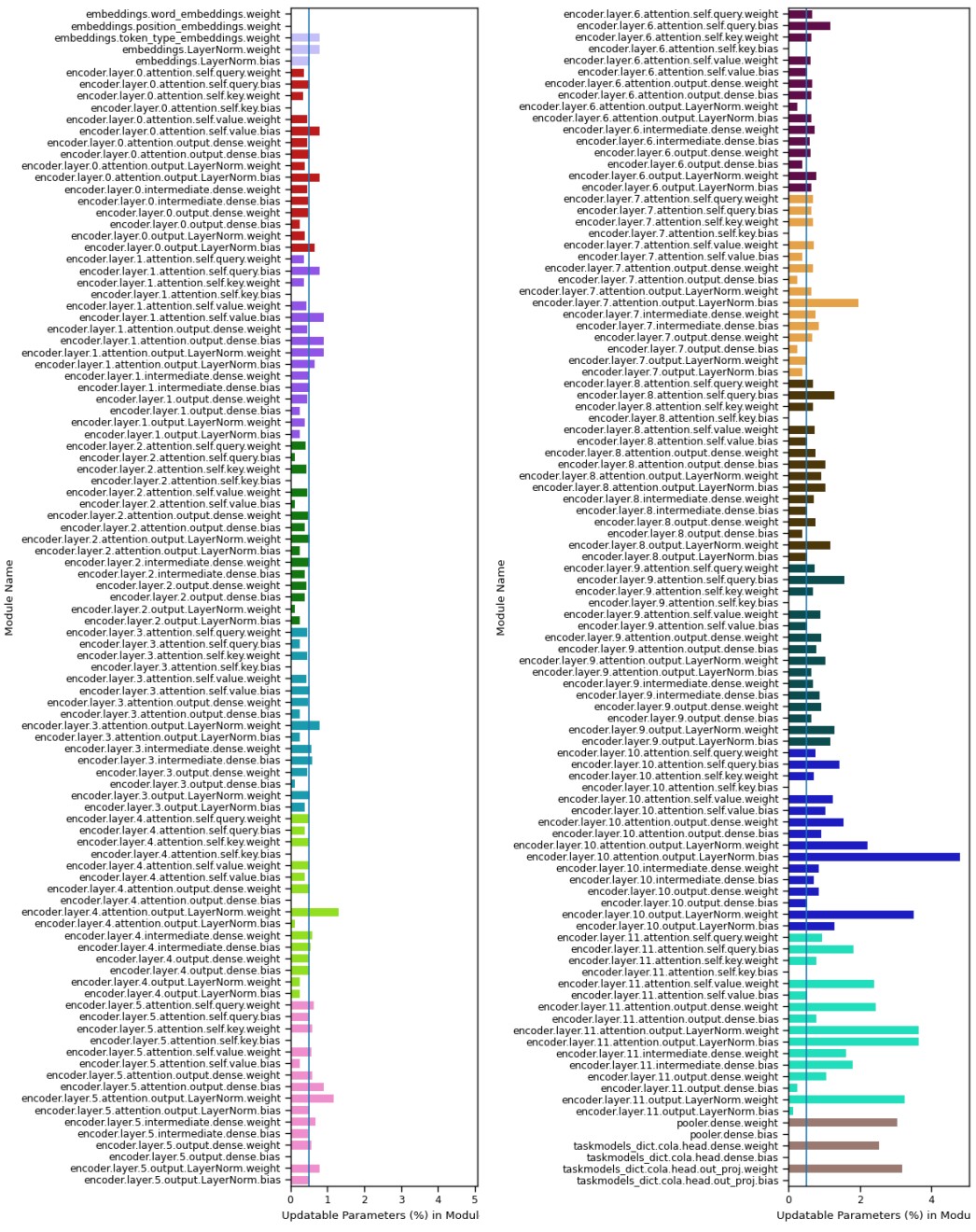

Figure 6: Sparsity pattern of the modules in RoBERTa-base on cola. Module-wise updatable parameters (%). (Left) The first half of the network and (Right) the second half. The default sparsity level $p^* = 0.5\%$ is shown as vertical line.

cola (**Layer-wise** sparsity pattern)

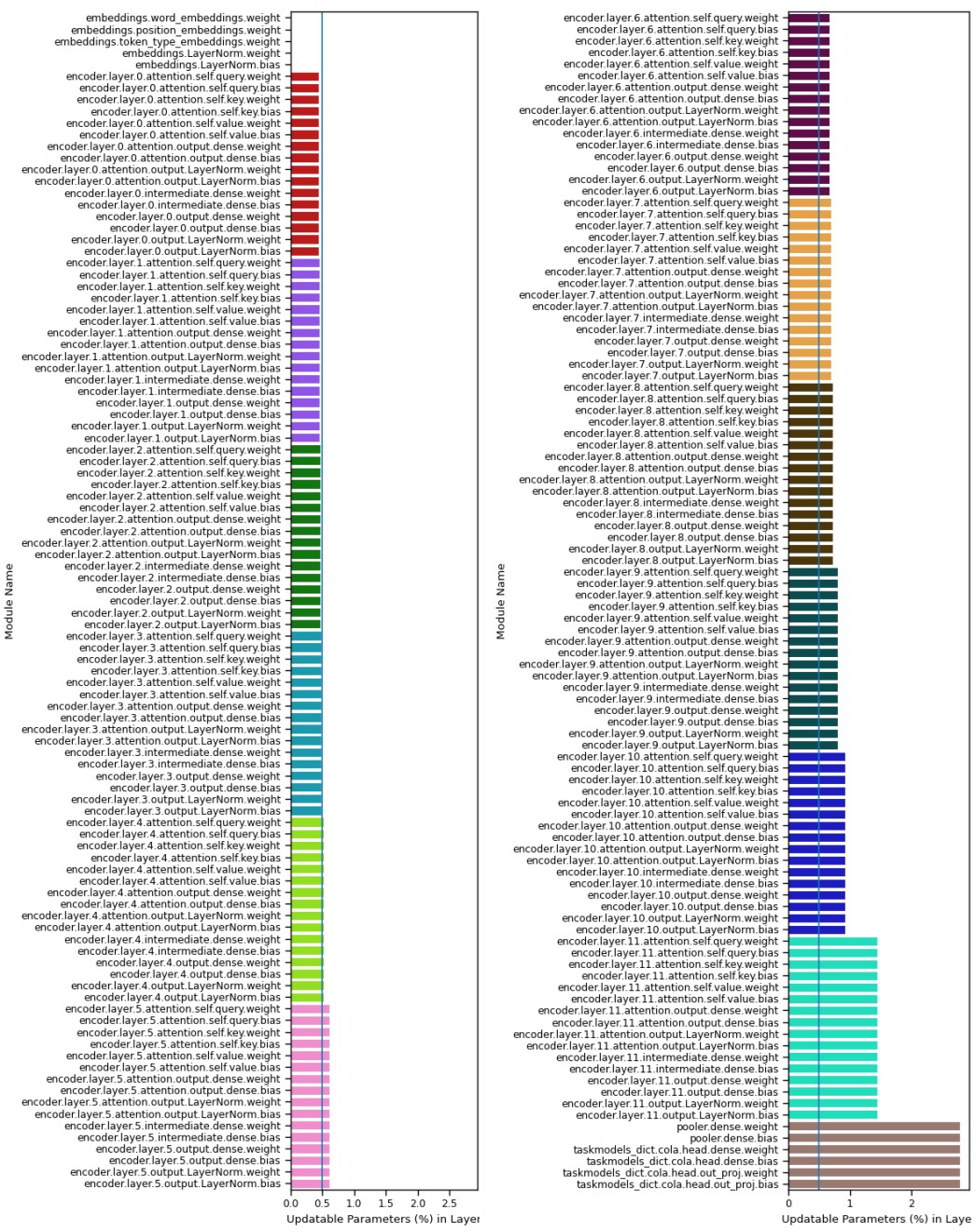

Figure 7: Sparsity pattern of the layers in RoBERTa-base on cola. Layer-wise updatable parameters (%). (Left) The first half of the network and (Right) the second half. The default sparsity level $p^* = 0.5\%$ is shown as vertical line.

stsb (**Module-wise** sparsity pattern)

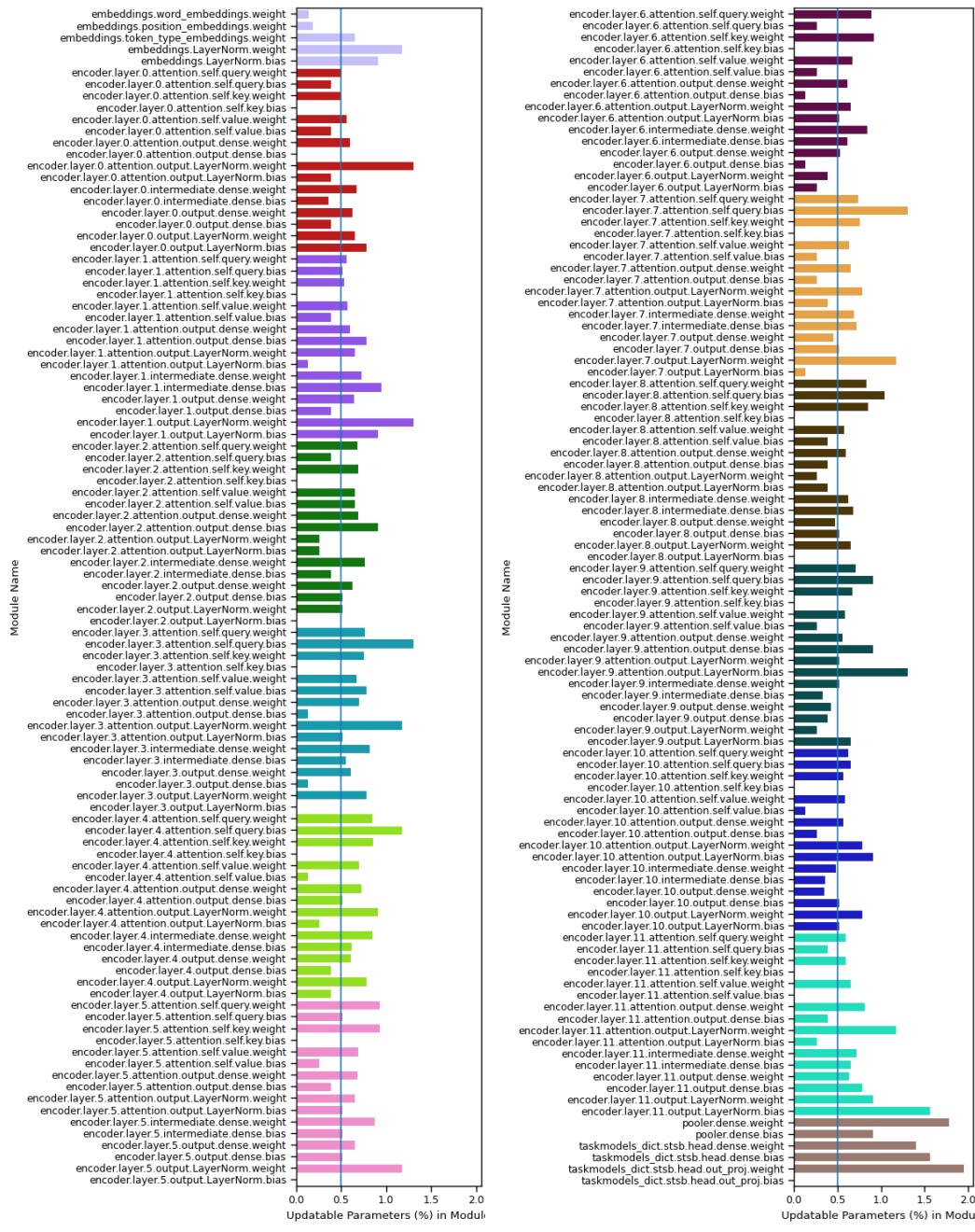

Figure 8: Sparsity pattern of the modules in RoBERTa-base on stsb. Module-wise updatable parameters (%). (Left) The first half of the network and (Right) the second half. The default sparsity level $p^* = 0.5\%$ is shown as vertical line.

# stsb (**Layer-wise** sparsity pattern)

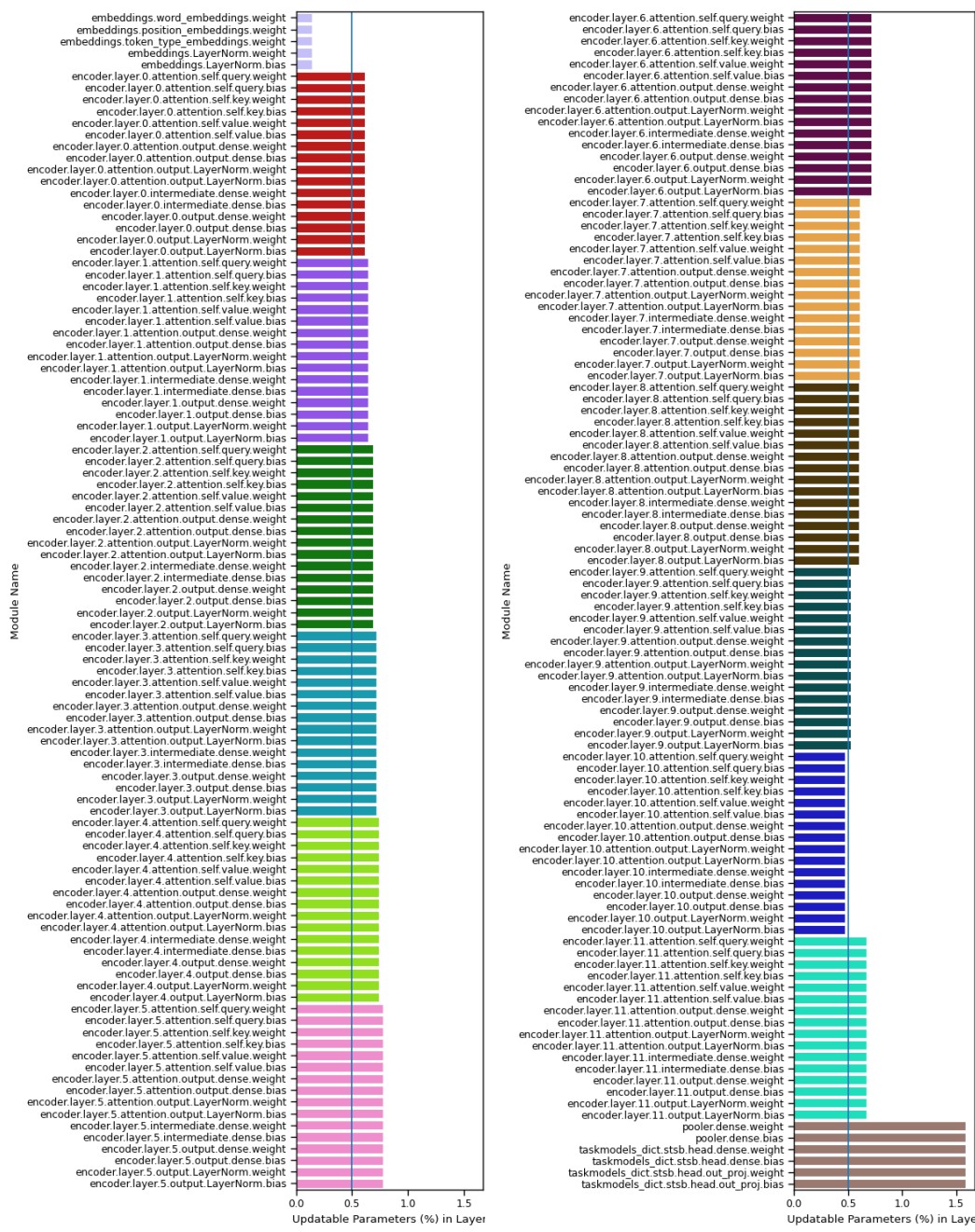

Figure 9: Sparsity pattern of the layers in RoBERTa-base on stsb. Layer-wise updatable parameters (%). (Left) The first half of the network and (Right) the second half. The default sparsity level $p^* = 0.5\%$ is shown as vertical line.

# mrpc (**Module-wise** sparsity pattern)

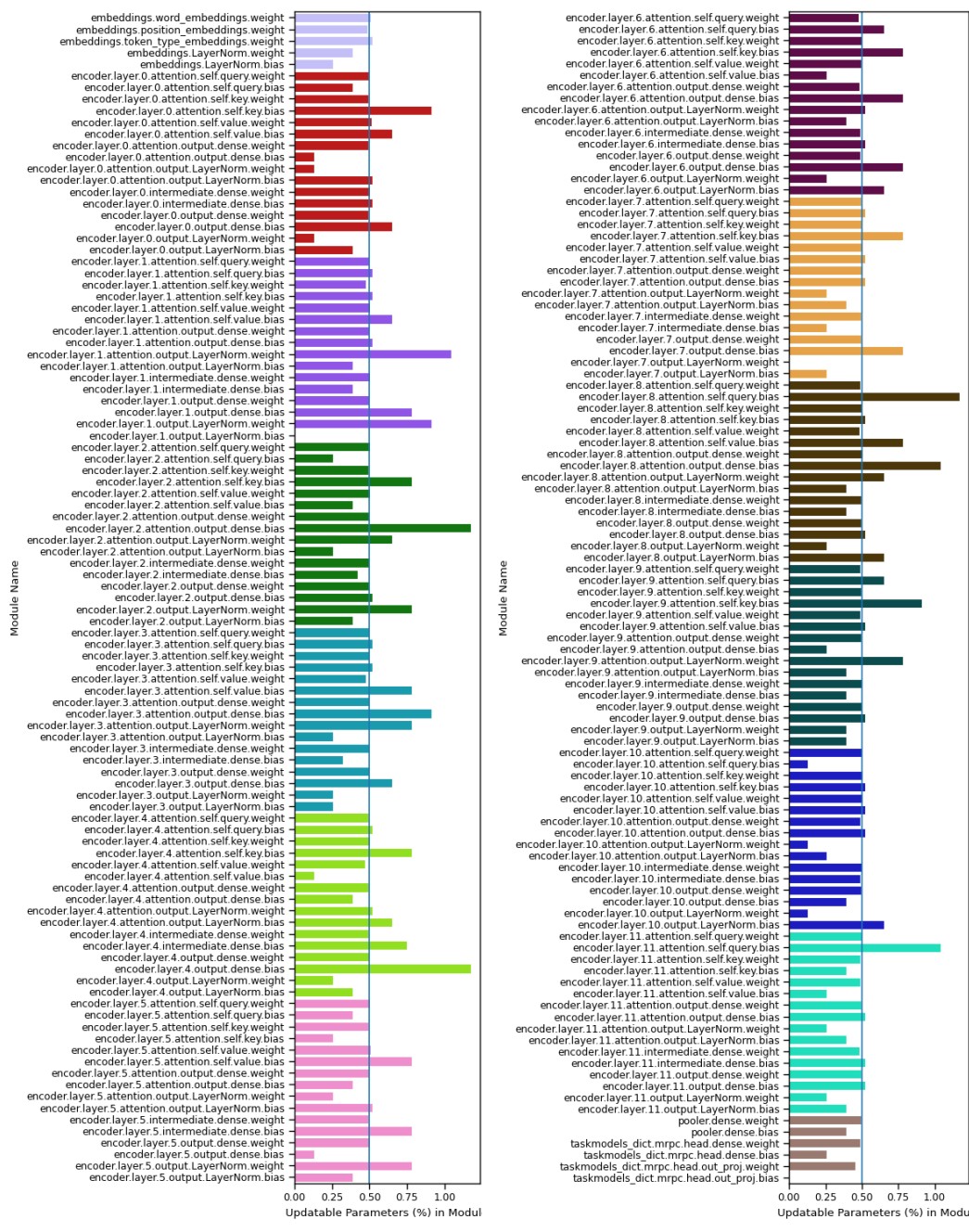

Figure 10: Sparsity pattern of the modules in RoBERTa-base on mrpc. Module-wise updatable parameters (%). (Left) The first half of the network and (Right) the second half. The default sparsity level $p^* = 0.5\%$ is shown as vertical line.

mrpc (**Layer-wise** sparsity pattern)

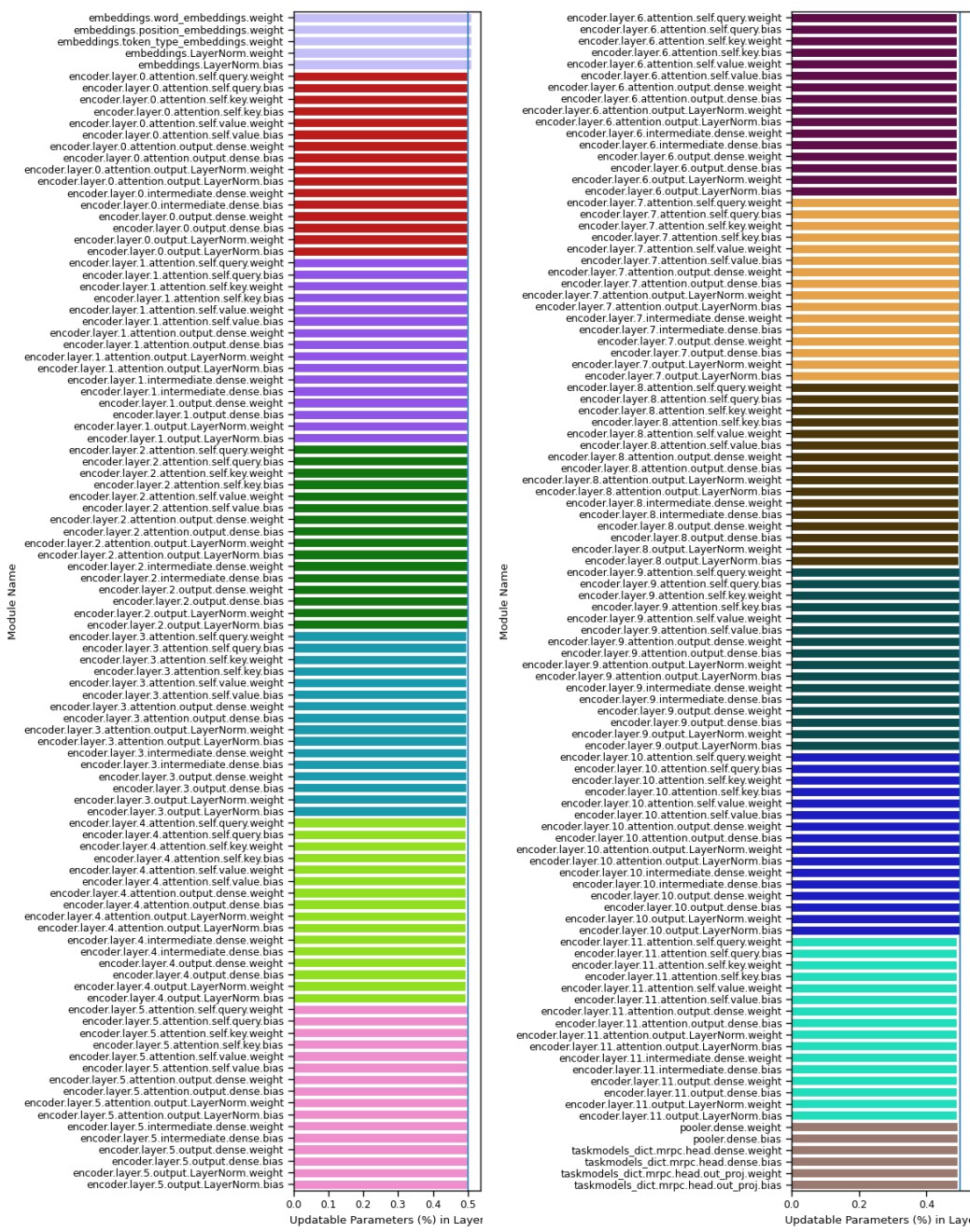

Figure 11: Sparsity pattern of the layers in RoBERTa-base on mrpc. Layer-wise updatable parameters (%). (Left) The first half of the network and (Right) the second half. The default sparsity level $p^* = 0.5\%$ is shown as vertical line.

# rte (**Module-wise** sparsity pattern)

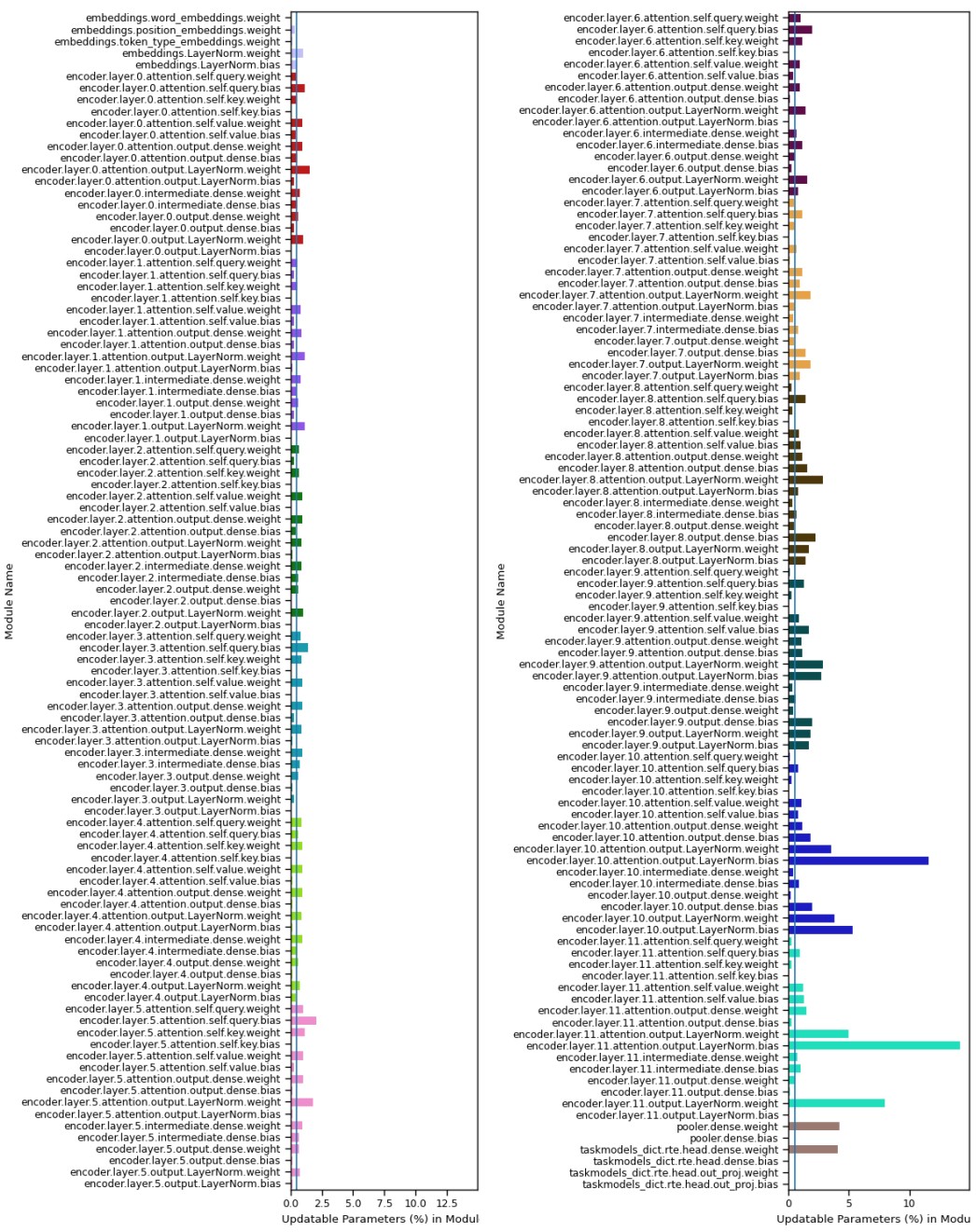

Figure 12: Sparsity pattern of the modules in RoBERTa-base on `rte`. Module-wise updatable parameters (%). (Left) The first half of the network and (Right) the second half. The default sparsity level $p^* = 0.5\%$ is shown as vertical line.

# rte (**Layer-wise** sparsity pattern)

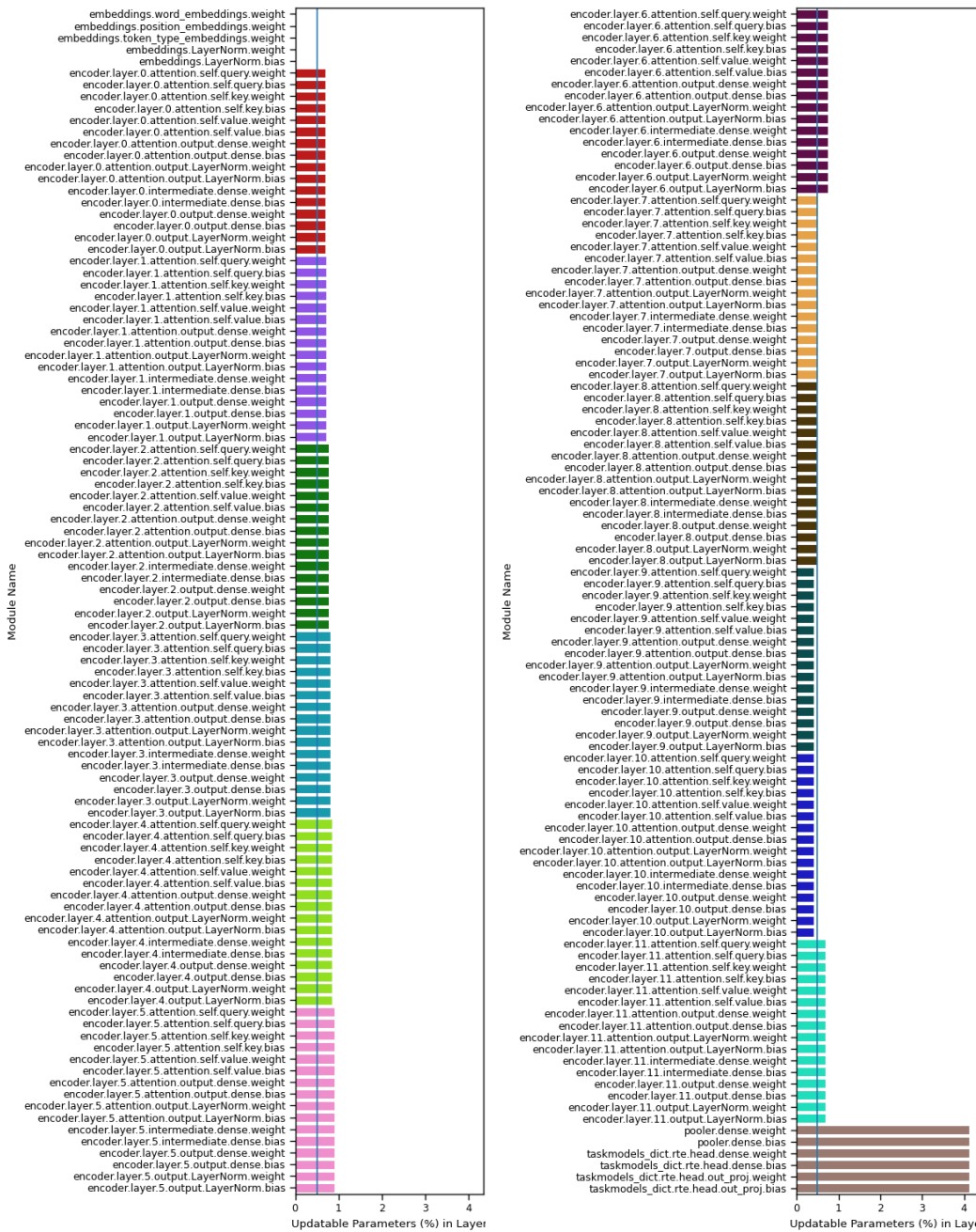

Figure 13: Sparsity pattern of the layers in RoBERTa-base on `rte`. Layer-wise updatable parameters (%). (Left) The first half of the network and (Right) the second half. The default sparsity level $p^* = 0.5\%$ is shown as vertical line.

# cb (**Module-wise** sparsity pattern)

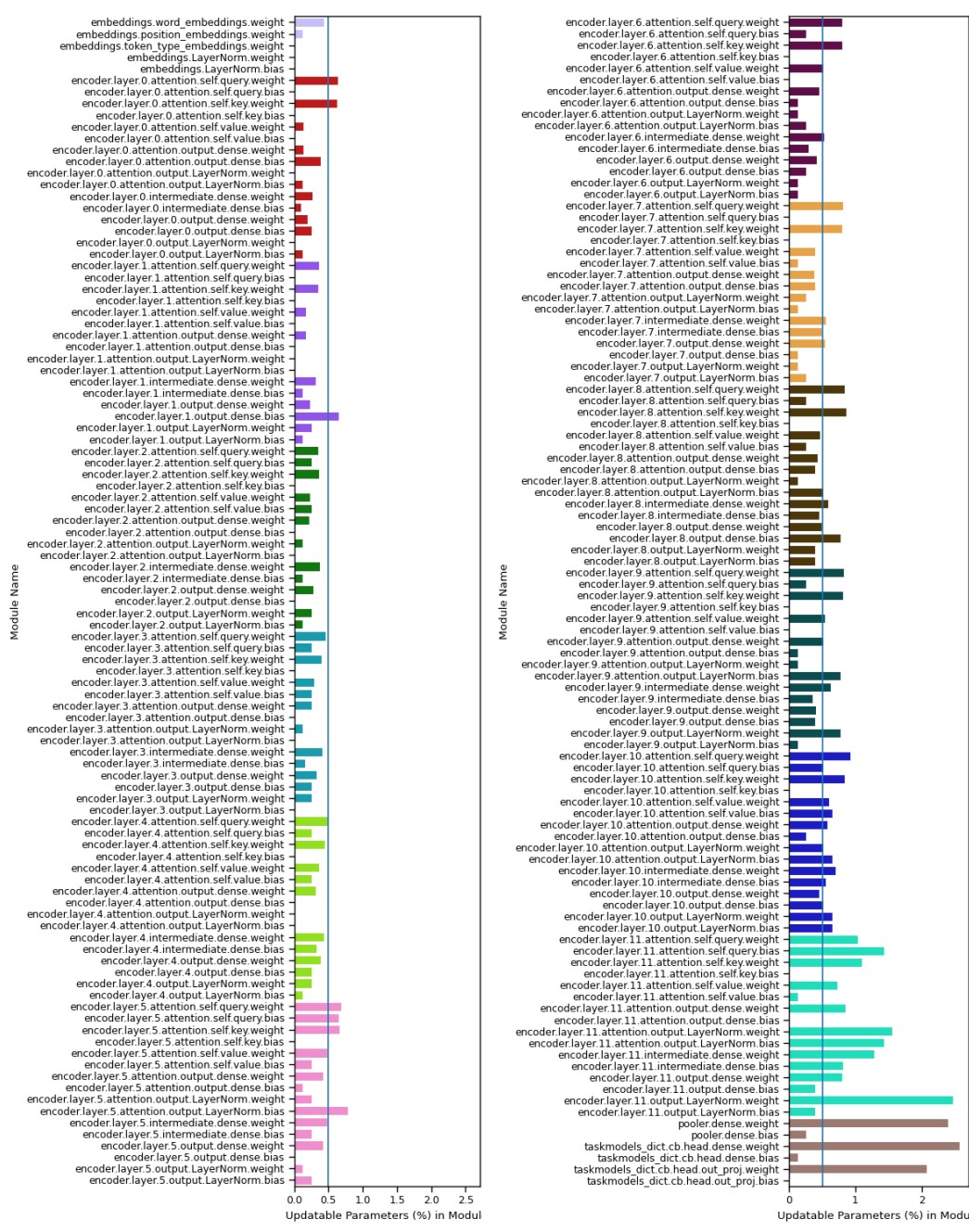

Figure 14: Sparsity pattern of the modules in RoBERTa-base on cb. Module-wise updatable parameters (%). (Left) The first half of the network and (Right) the second half. The default sparsity level $p^* = 0.5\%$ is shown as vertical line.

cb (**Layer-wise** sparsity pattern)

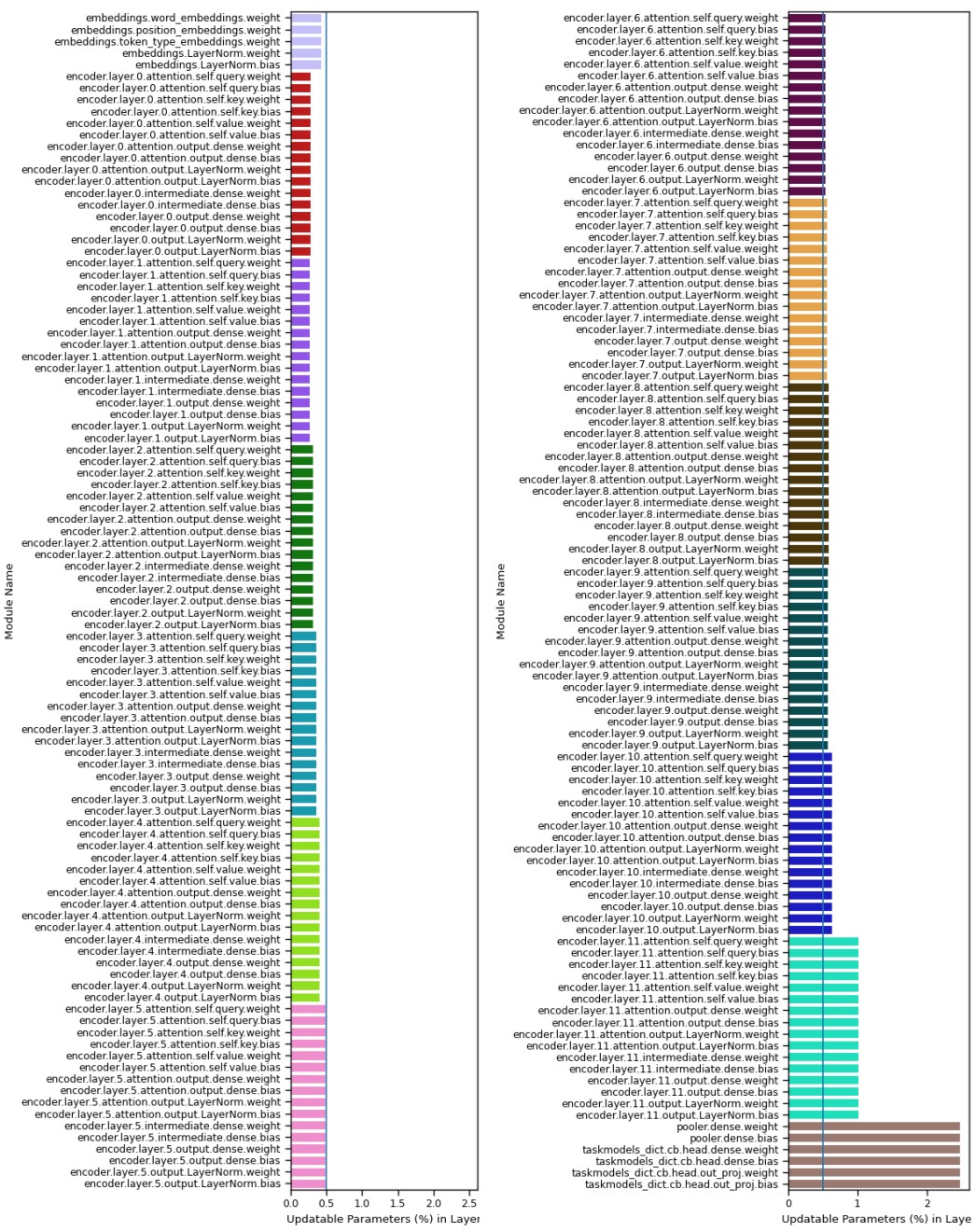

Figure 15: Sparsity pattern of the layers in RoBERTa-base on cb. Layer-wise updatable parameters (%). (Left) The first half of the network and (Right) the second half. The default sparsity level $p^* = 0.5\%$ is shown as vertical line.

## copa (**Module-wise** sparsity pattern)

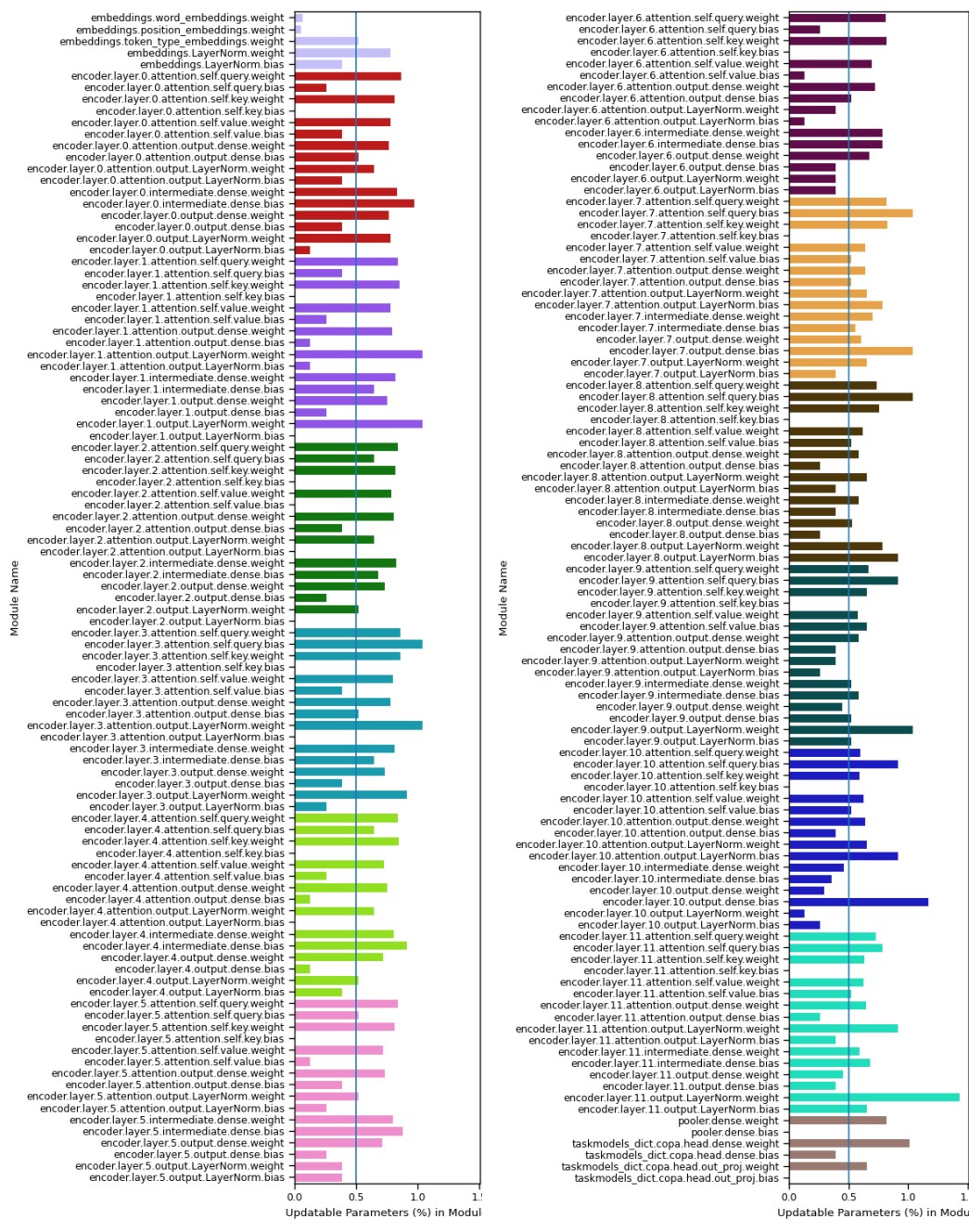

Figure 16: Sparsity pattern of the modules in RoBERTa-base on copa. Module-wise updatable parameters (%). (Left) The first half of the network and (Right) the second half. The default sparsity level $p^* = 0.5\%$ is shown as vertical line.

copa (**Layer-wise** sparsity pattern)

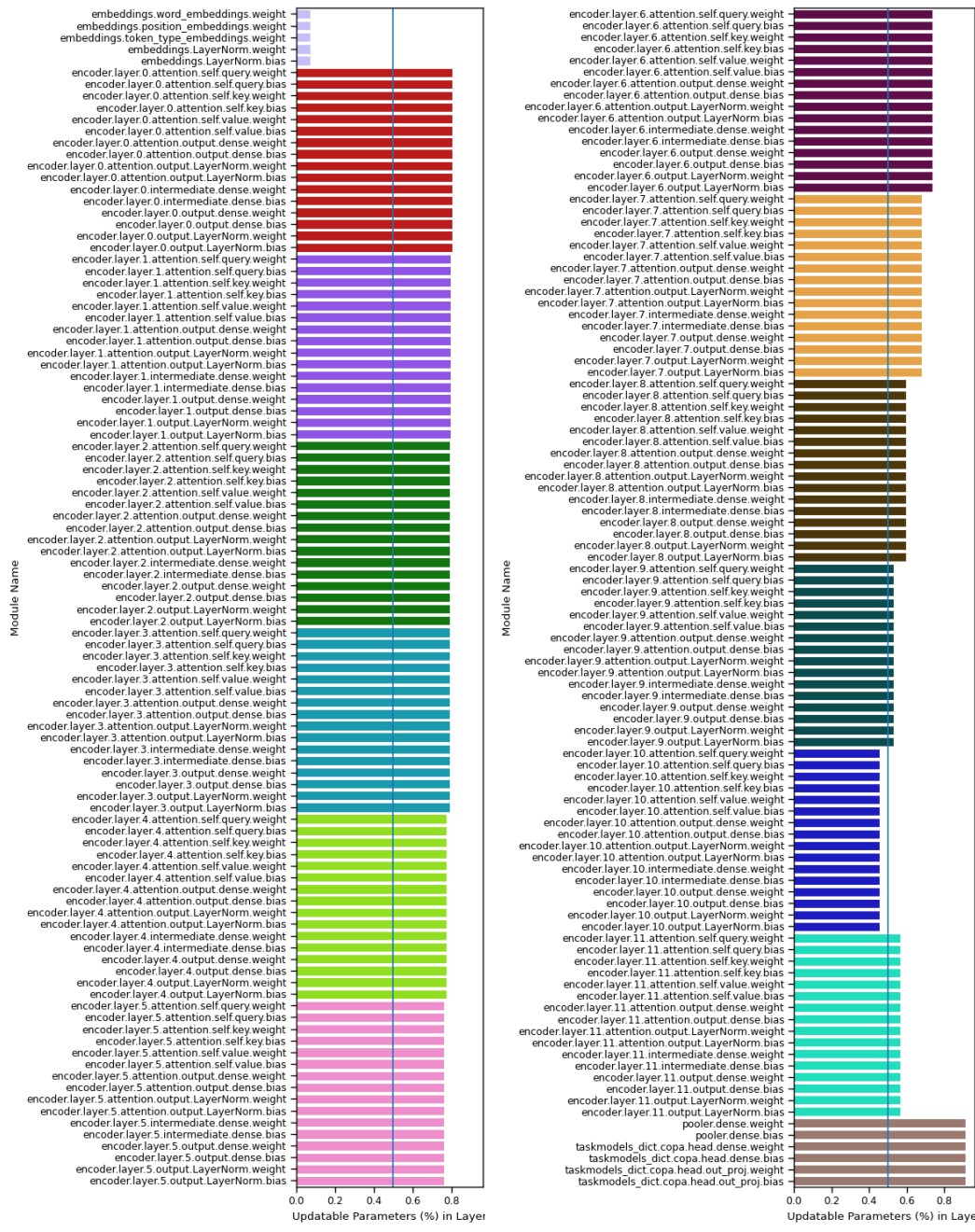

Figure 17: Sparsity pattern of the layers in RoBERTa-base on copa. Layer-wise updatable parameters (%). (Left) The first half of the network and (Right) the second half. The default sparsity level $p^* = 0.5\%$ is shown as vertical line.

# wsc (**Module-wise** sparsity pattern)

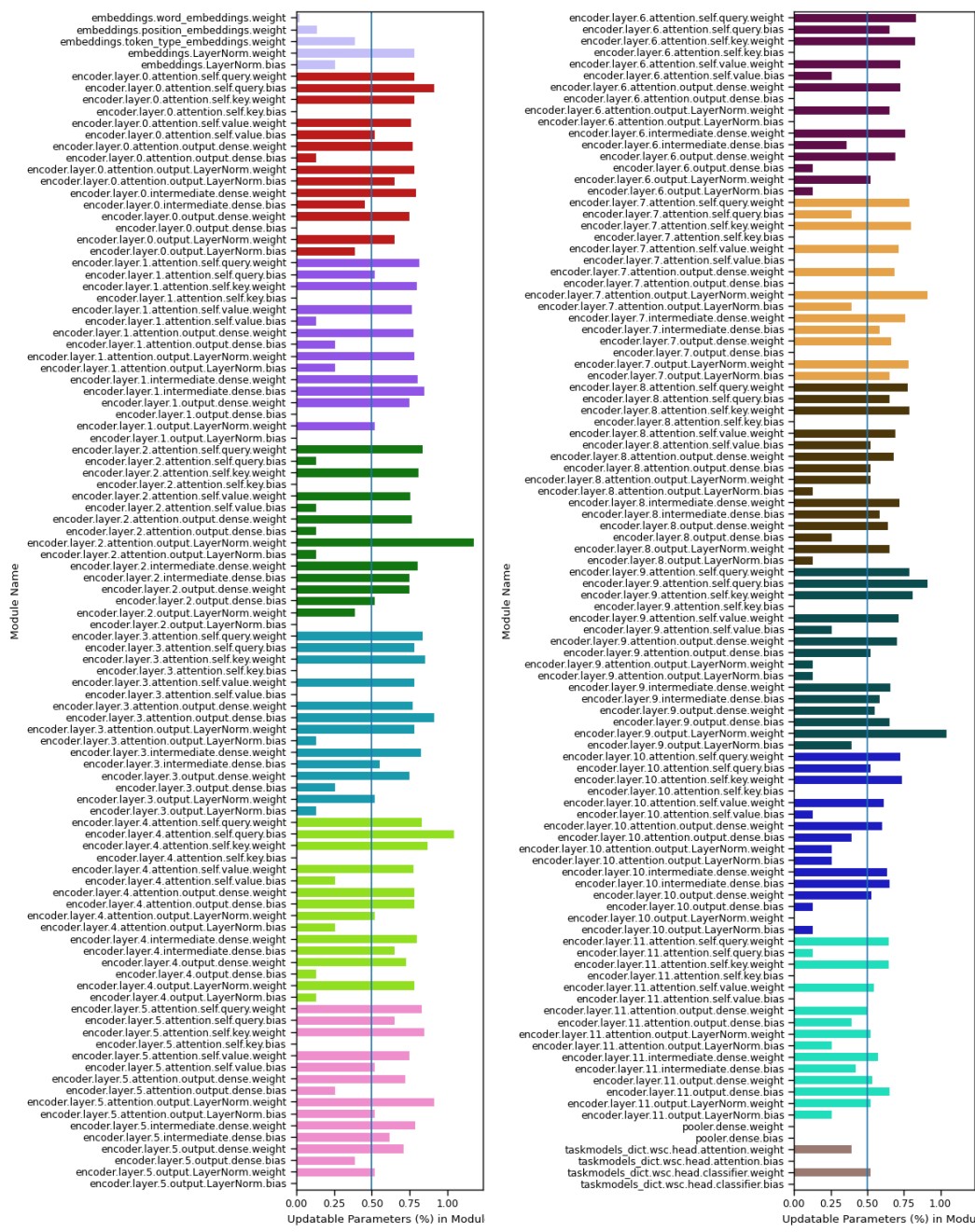

Figure 18: Sparsity pattern of the modules in RoBERTa-base on wsc. Module-wise updatable parameters (%). (Left) The first half of the network and (Right) the second half. The default sparsity level $p^* = 0.5\%$ is shown as vertical line.

# wsc (**Layer-wise** sparsity pattern)

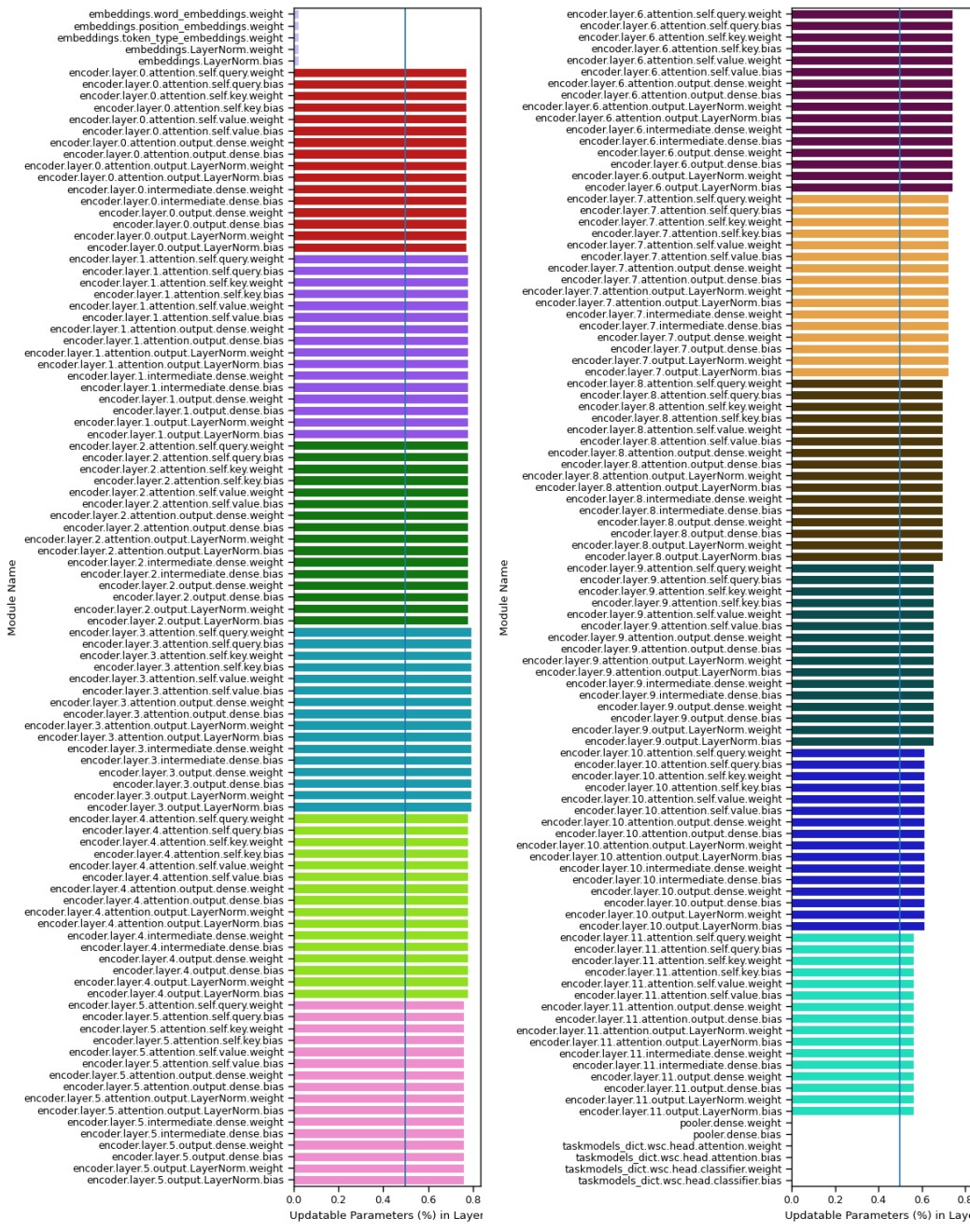

Figure 19: Sparsity pattern of the layers in RoBERTa-base on wsc. Layer-wise updatable parameters (%). (Left) The first half of the network and (Right) the second half. The default sparsity level $p^* = 0.5\%$ is shown as vertical line.

# cifar100

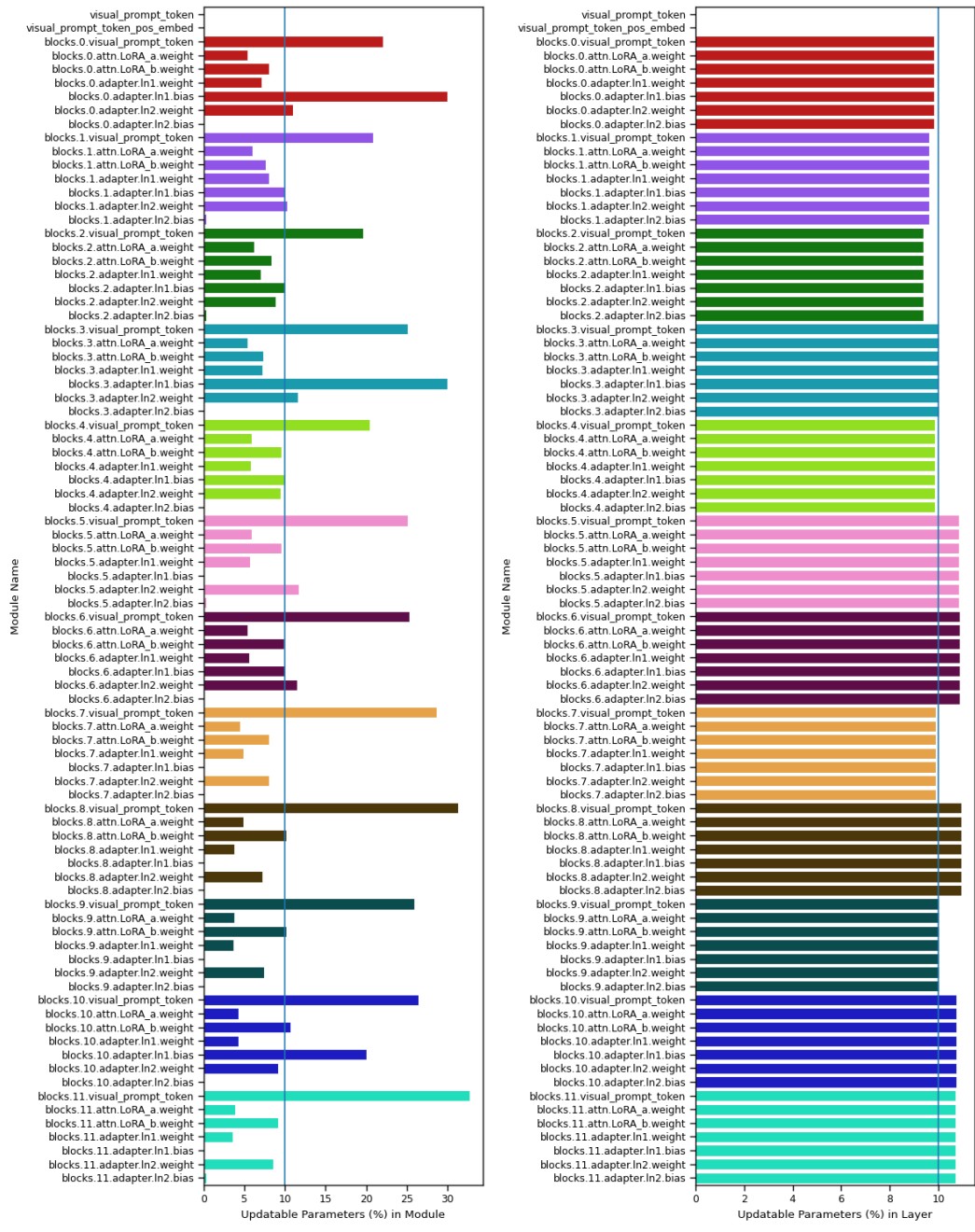

Figure 20: Sparsity pattern of attached modules to ViT-B/16 on cifar100. (Left) Module-wise updatable parameters (%) and (Right) Layer-wise updatable parameters (%). The BayesTune's optimal sparsity level $p^* = 10\%$ is shown as vertical line.

# caltech101

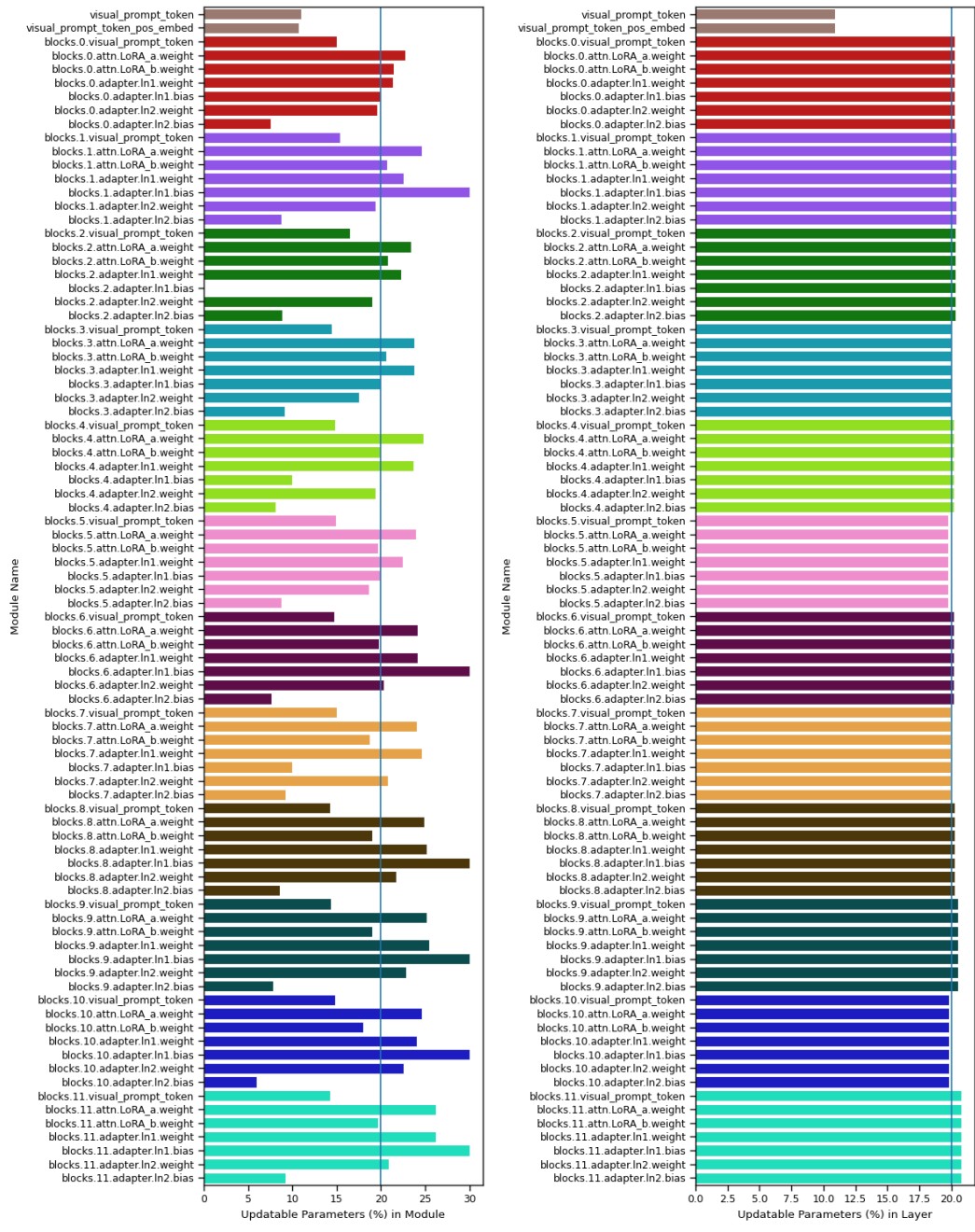

Figure 21: Sparsity pattern of attached modules to ViT-B/16 on caltech101. (Left) Module-wise updatable parameters (%) and (Right) Layer-wise updatable parameters (%). The BayesTune's optimal sparsity level $p^* = 20\%$ is shown as vertical line.

dtd

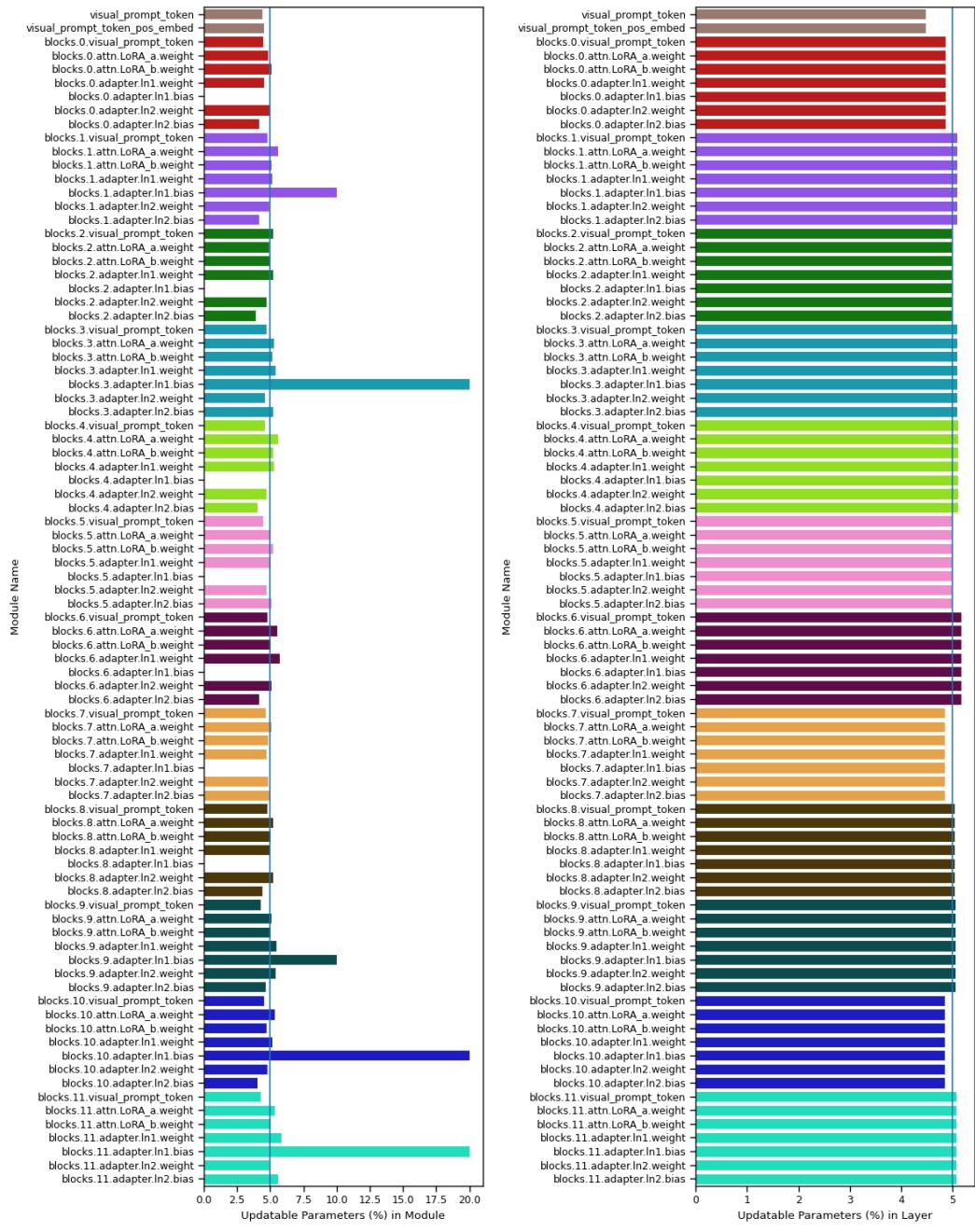

Figure 22: Sparsity pattern of attached modules to ViT-B/16 on dtd. (Left) Module-wise updatable parameters (%) and (Right) Layer-wise updatable parameters (%). The BayesTune's optimal sparsity level $p^* = 5\%$ is shown as vertical line.

# flower102

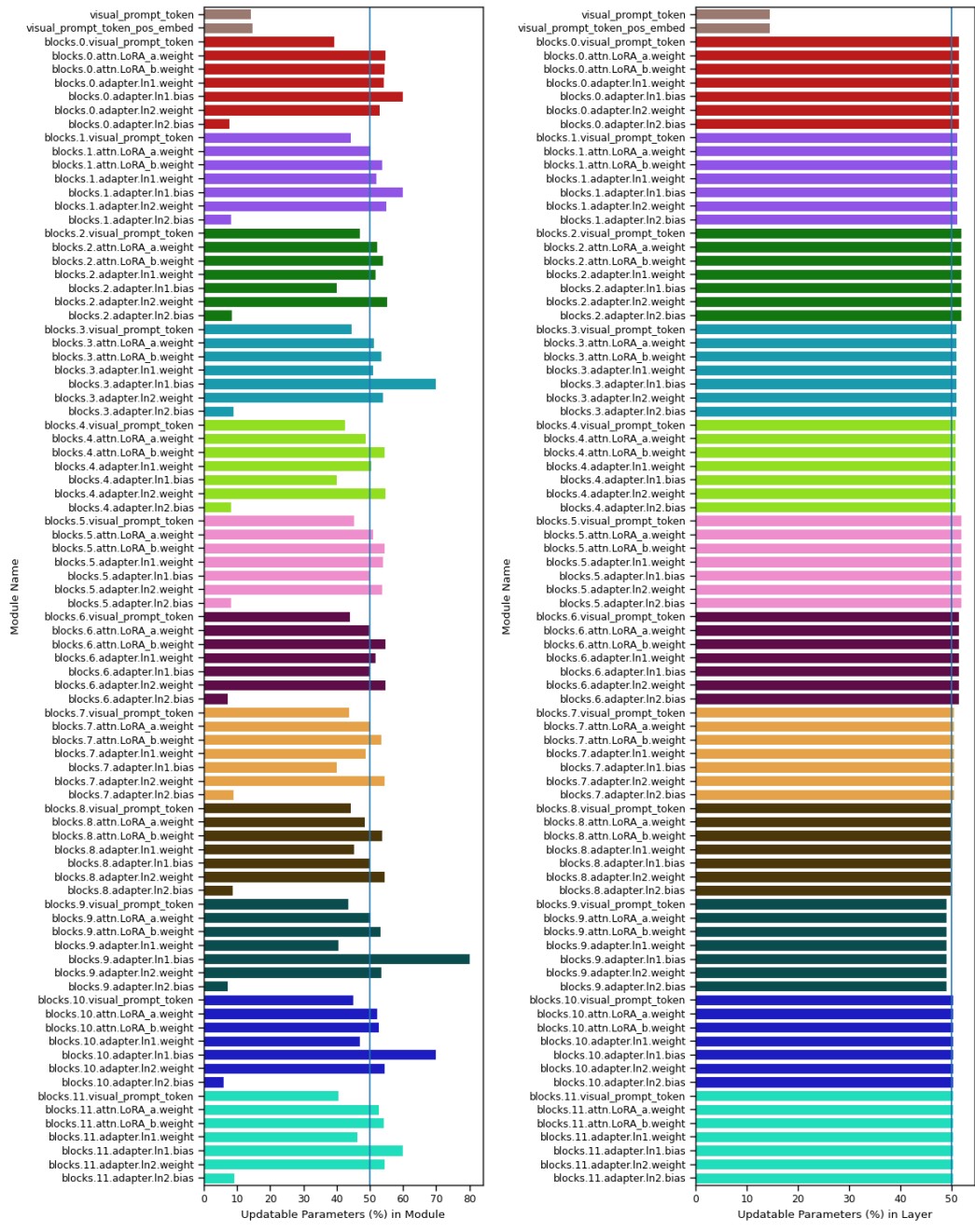

Figure 23: Sparsity pattern of attached modules to ViT-B/16 on flower102. (Left) Module-wise updatable parameters (%) and (Right) Layer-wise updatable parameters (%). The BayesTune's optimal sparsity level $p^* = 50\%$ is shown as vertical line.

pets

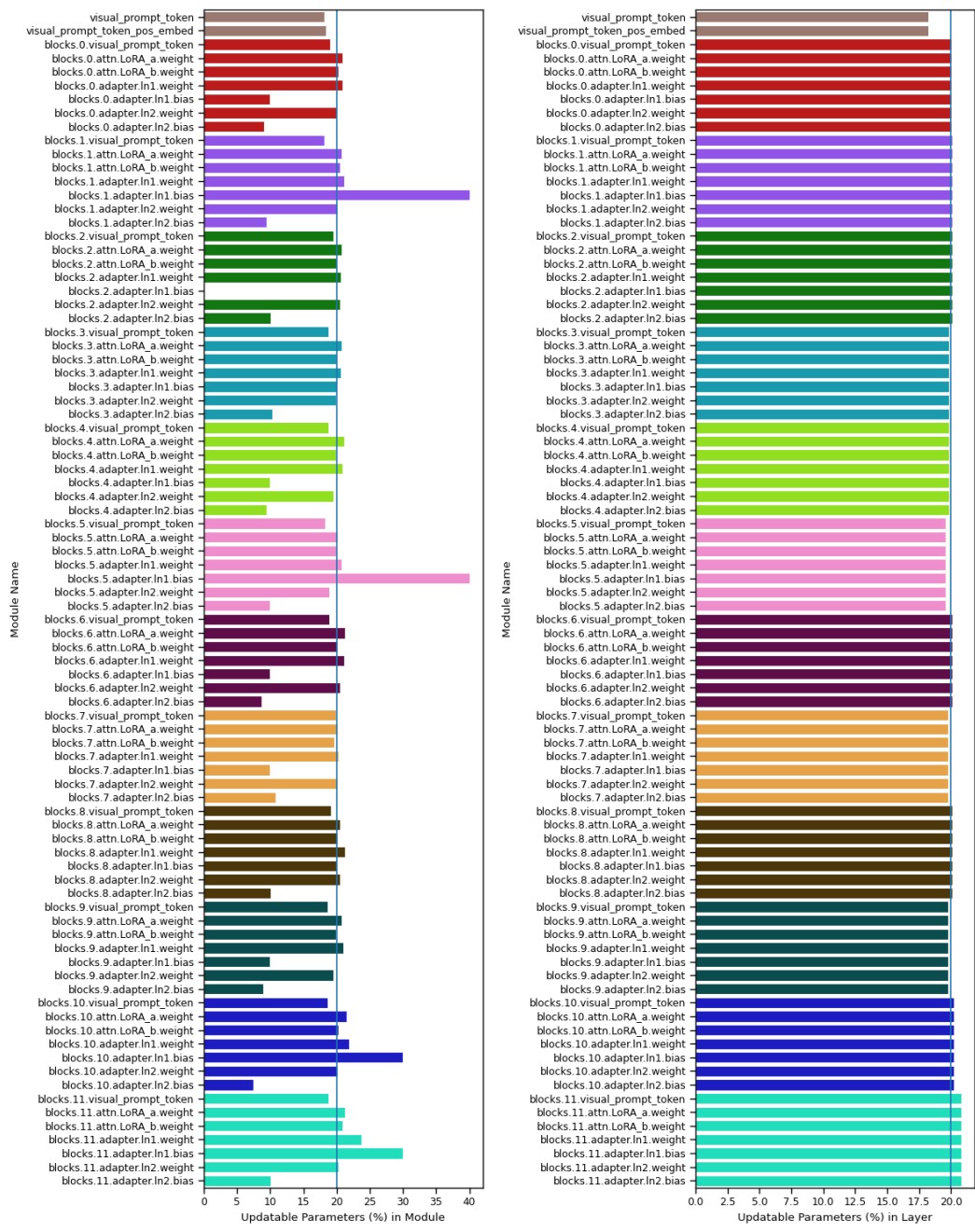

Figure 24: Sparsity pattern of attached modules to ViT-B/16 on pets. (Left) Module-wise updatable parameters (%) and (Right) Layer-wise updatable parameters (%). The BayesTune's optimal sparsity level $p^* = 20\%$ is shown as vertical line.

svhn

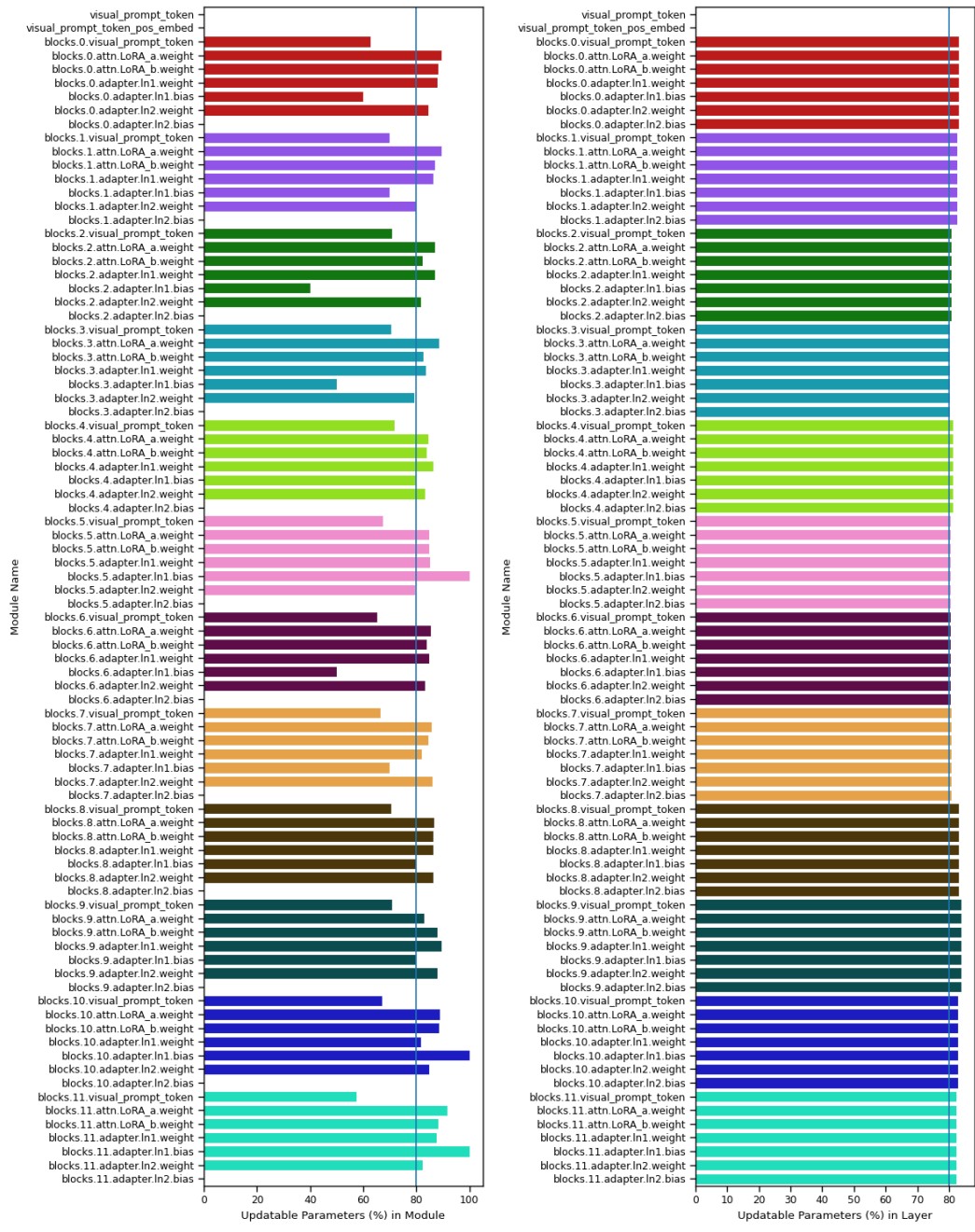

Figure 25: Sparsity pattern of attached modules to ViT-B/16 on svhn. (Left) Module-wise updatable parameters (%) and (Right) Layer-wise updatable parameters (%). The BayesTune's optimal sparsity level $p^* = 80\%$ is shown as vertical line.

## sun397

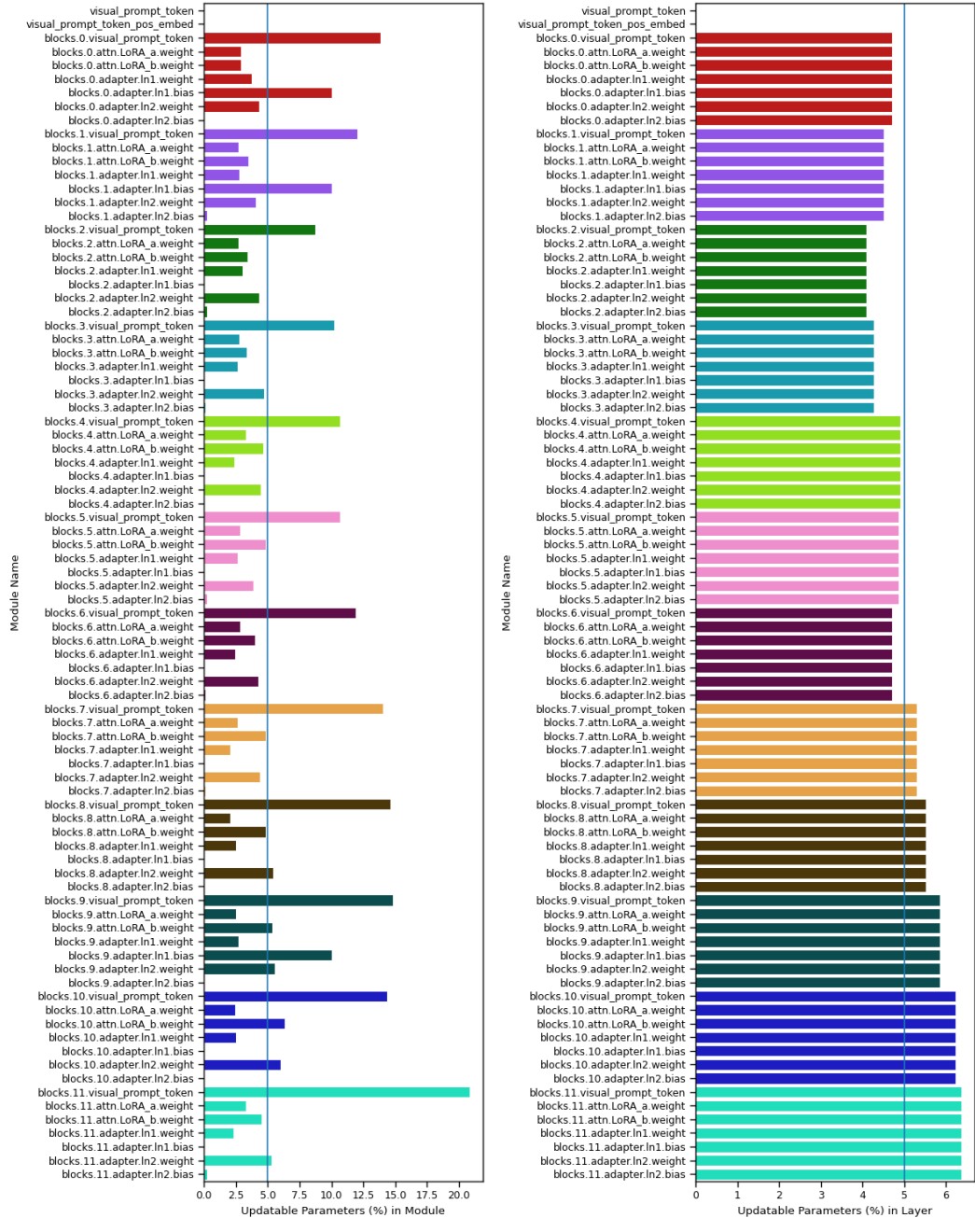

Figure 26: Sparsity pattern of attached modules to ViT-B/16 on sun397. (Left) Module-wise updatable parameters (%) and (Right) Layer-wise updatable parameters (%). The BayesTune's optimal sparsity level $p^* = 5\%$ is shown as vertical line.

camelyon

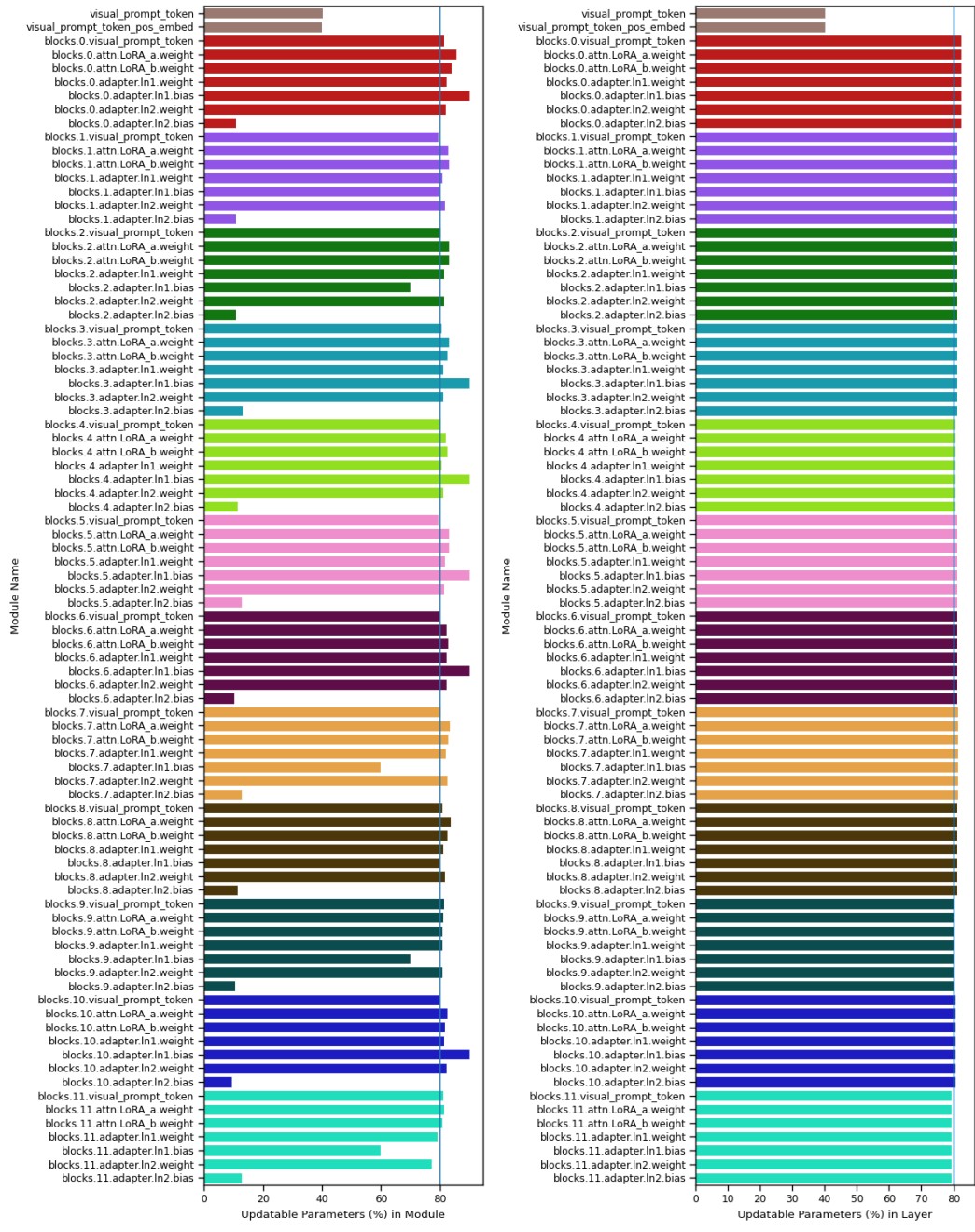

Figure 27: Sparsity pattern of attached modules to ViT-B/16 on camelyon. (Left) Module-wise updatable parameters (%) and (Right) Layer-wise updatable parameters (%). The BayesTune's optimal sparsity level $p^* = 80\%$ is shown as vertical line.

eurosat

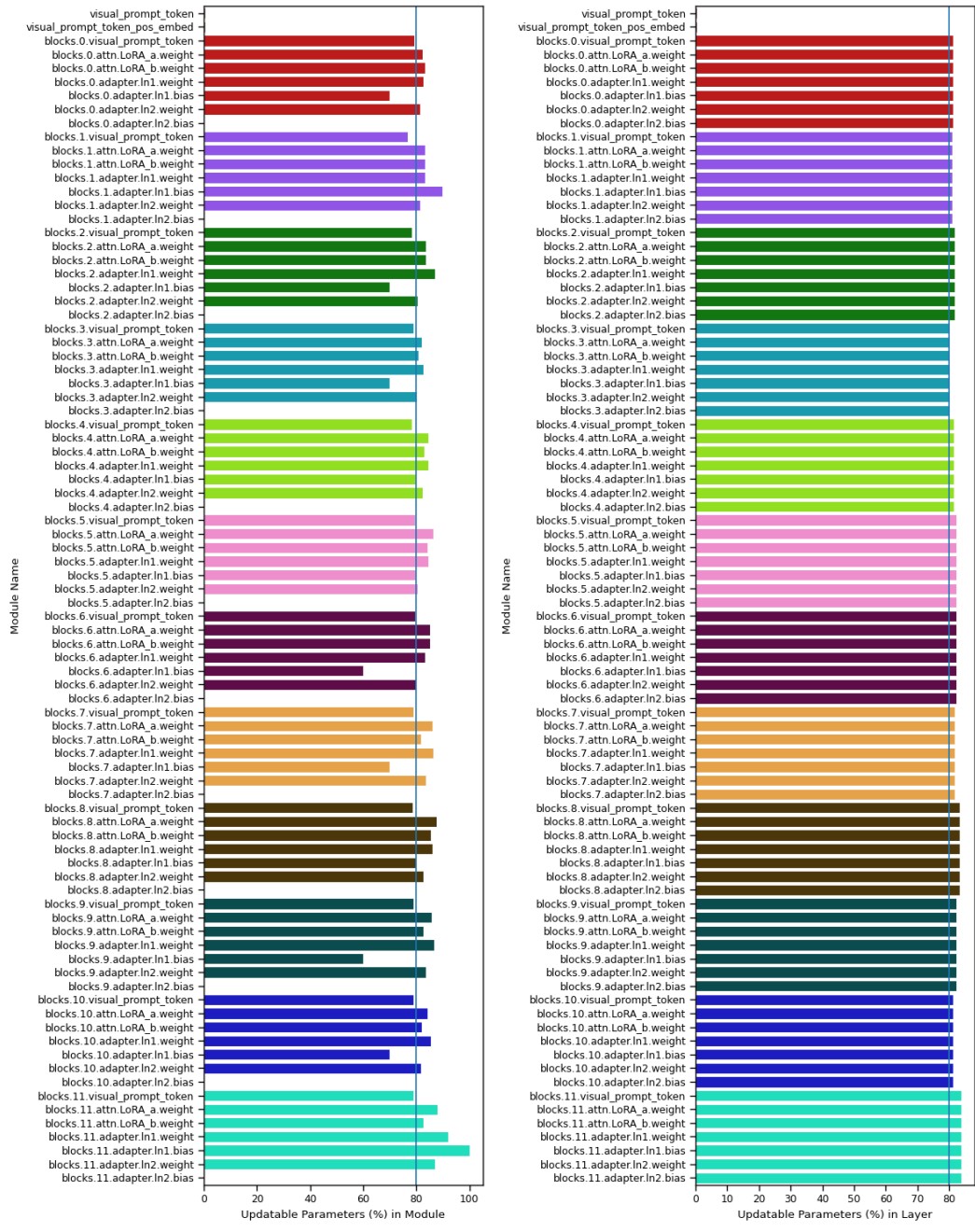

Figure 28: Sparsity pattern of attached modules to ViT-B/16 on eurosat. (Left) Module-wise updatable parameters (%) and (Right) Layer-wise updatable parameters (%). The BayesTune's optimal sparsity level $p^* = 80\%$ is shown as vertical line.

resisc45

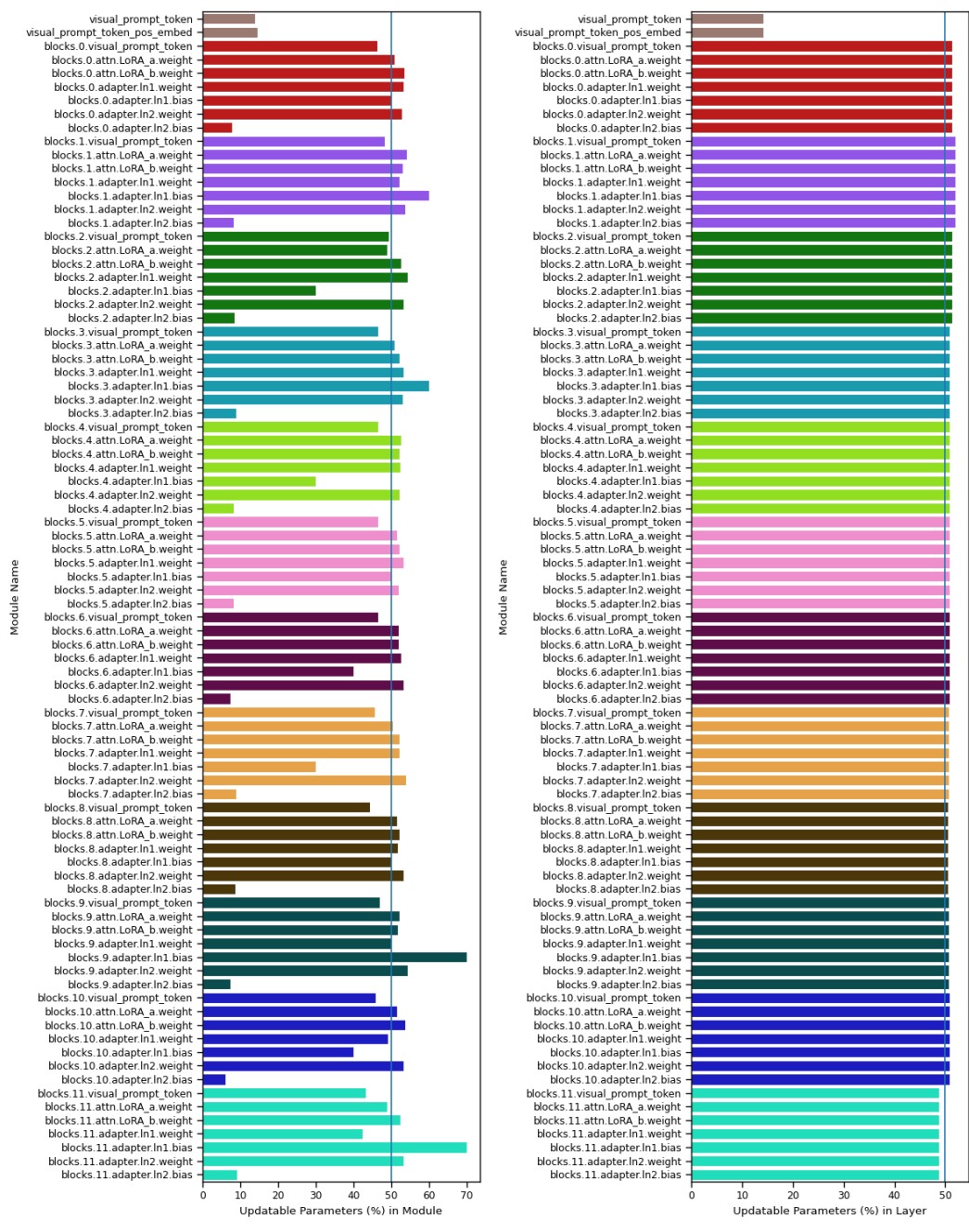

Figure 29: Sparsity pattern of attached modules to ViT-B/16 on resisc45. (Left) Module-wise updatable parameters (%) and (Right) Layer-wise updatable parameters (%). The BayesTune's optimal sparsity level $p^* = 50\%$ is shown as vertical line.

# retinopathy

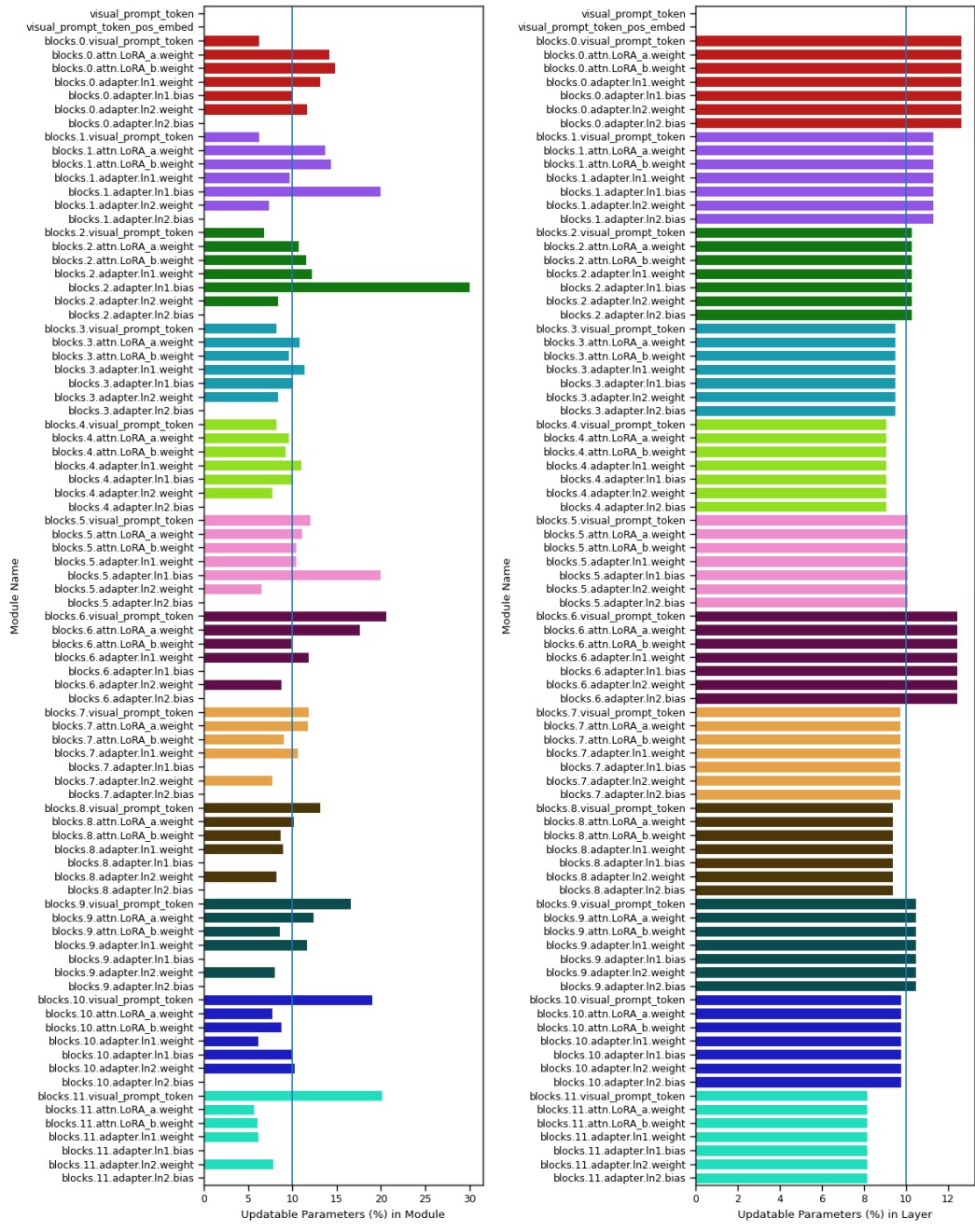

Figure 30: Sparsity pattern of attached modules to ViT-B/16 on `retinopathy`. (Left) Module-wise updatable parameters (%) and (Right) Layer-wise updatable parameters (%). The BayesTune's optimal sparsity level $p^* = 10\%$ is shown as vertical line.

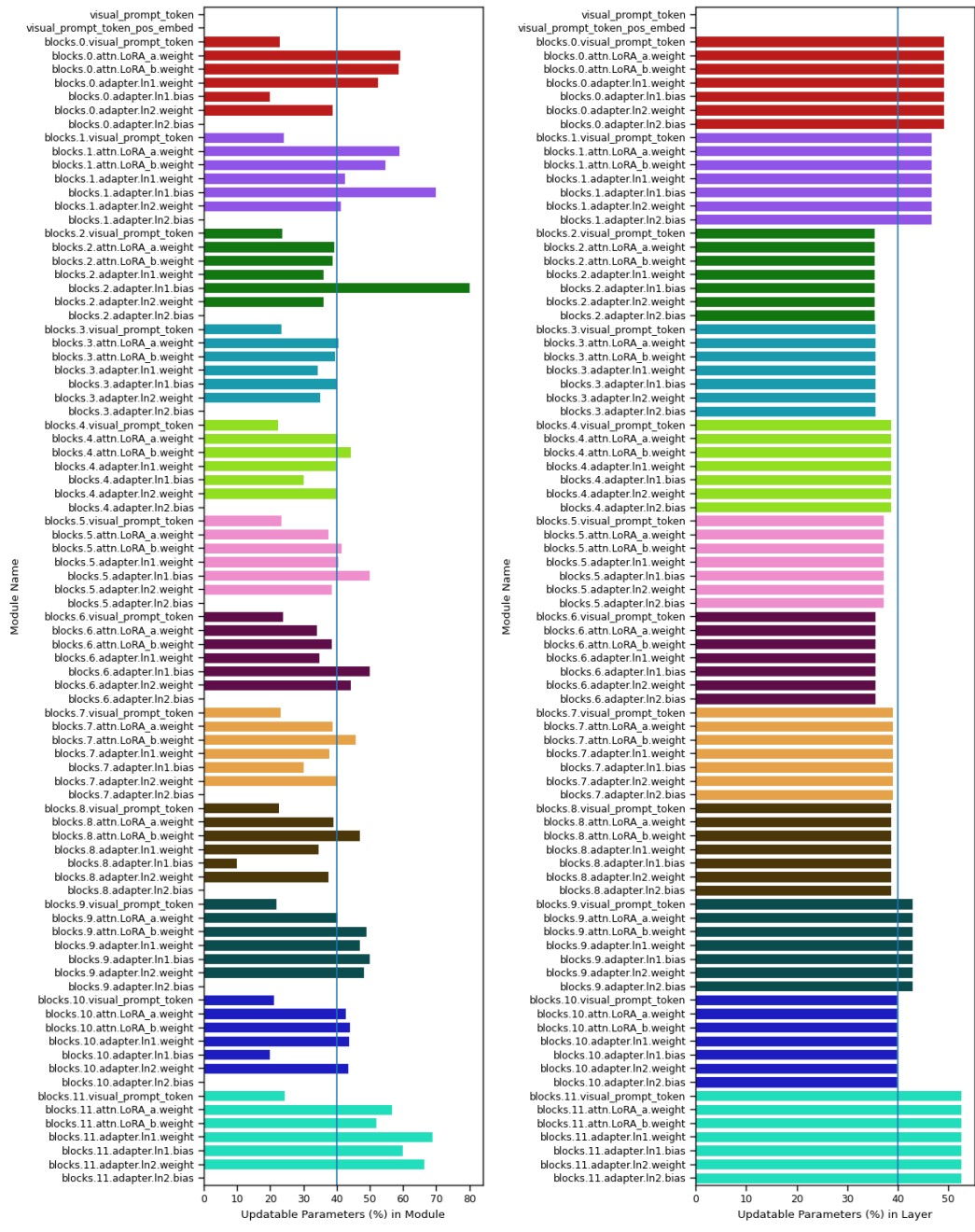

Figure 31: Sparsity pattern of attached modules to ViT-B/16 on clevr-count. (Left) Module-wise updatable parameters (%) and (Right) Layer-wise updatable parameters (%). The BayesTune's optimal sparsity level $p^* = 40\%$ is shown as vertical line.

# clevr-dist

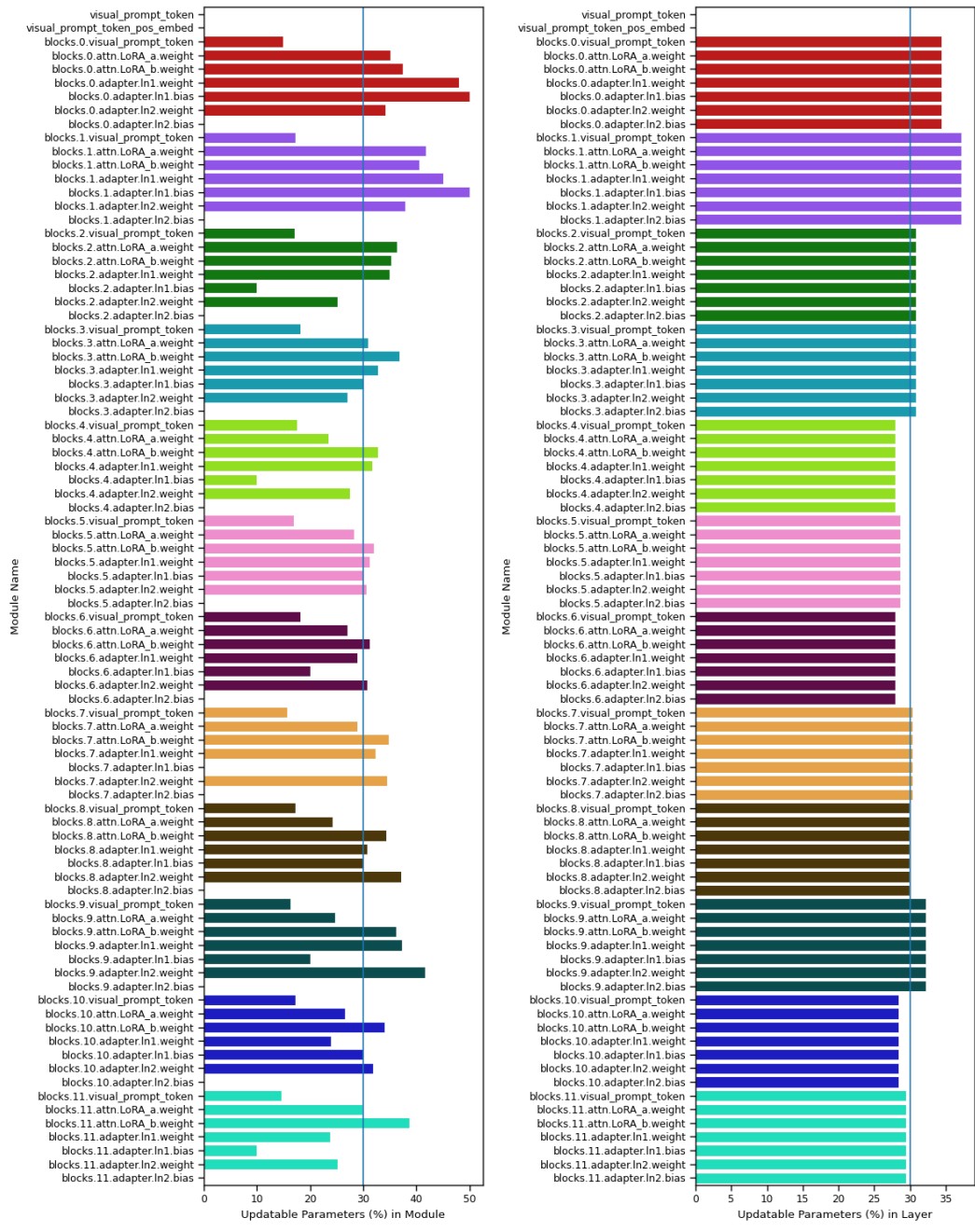

Figure 32: Sparsity pattern of attached modules to ViT-B/16 on `clevr-dist`. (Left) Module-wise updatable parameters (%) and (Right) Layer-wise updatable parameters (%). The BayesTune's optimal sparsity level $p^* = 30\%$ is shown as vertical line.

**dmlab**

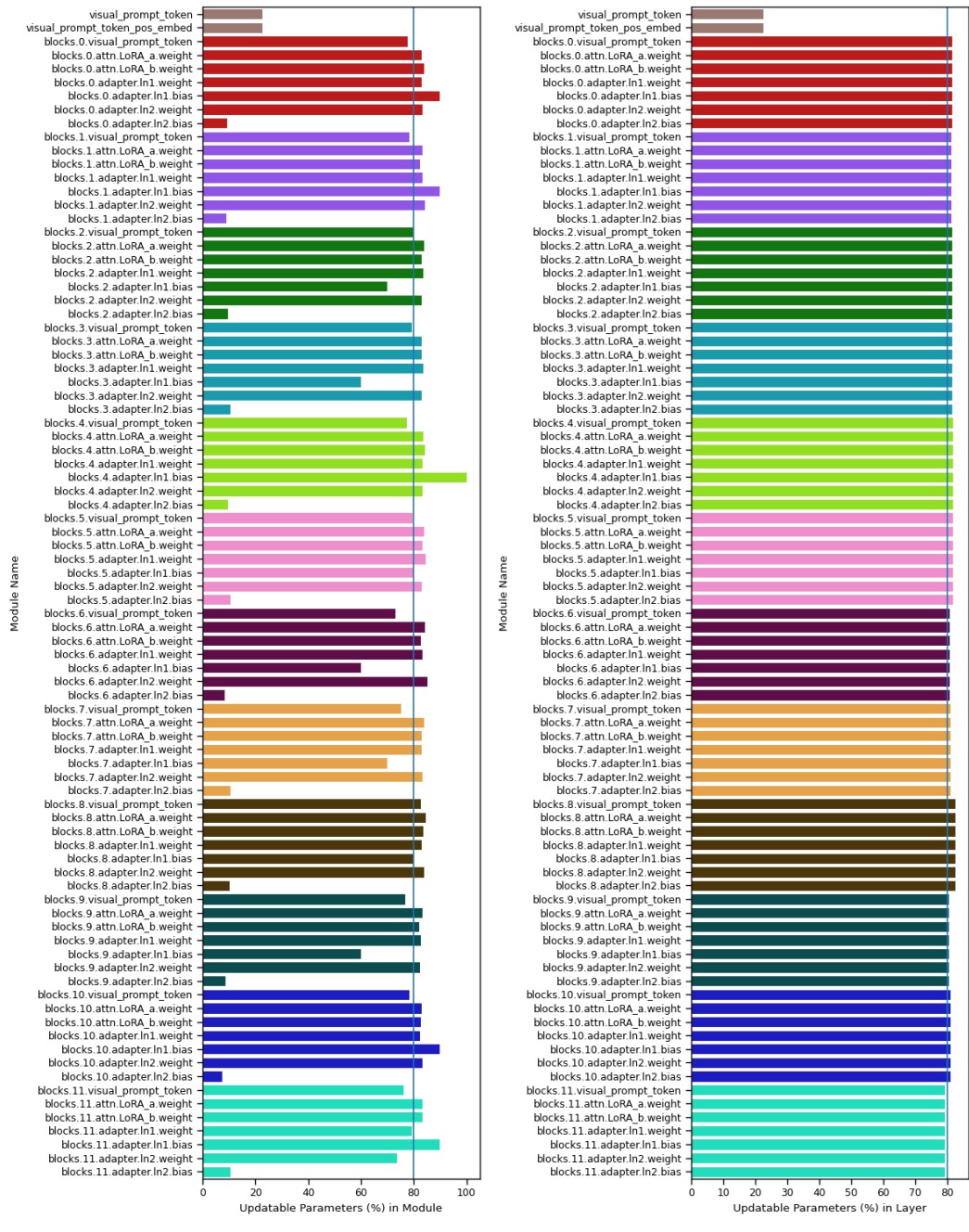

Figure 33: Sparsity pattern of attached modules to ViT-B/16 on `dmlab`. (Left) Module-wise updatable parameters (%) and (Right) Layer-wise updatable parameters (%). The BayesTune's optimal sparsity level $p^* = 80\%$ is shown as vertical line.

**kitti**

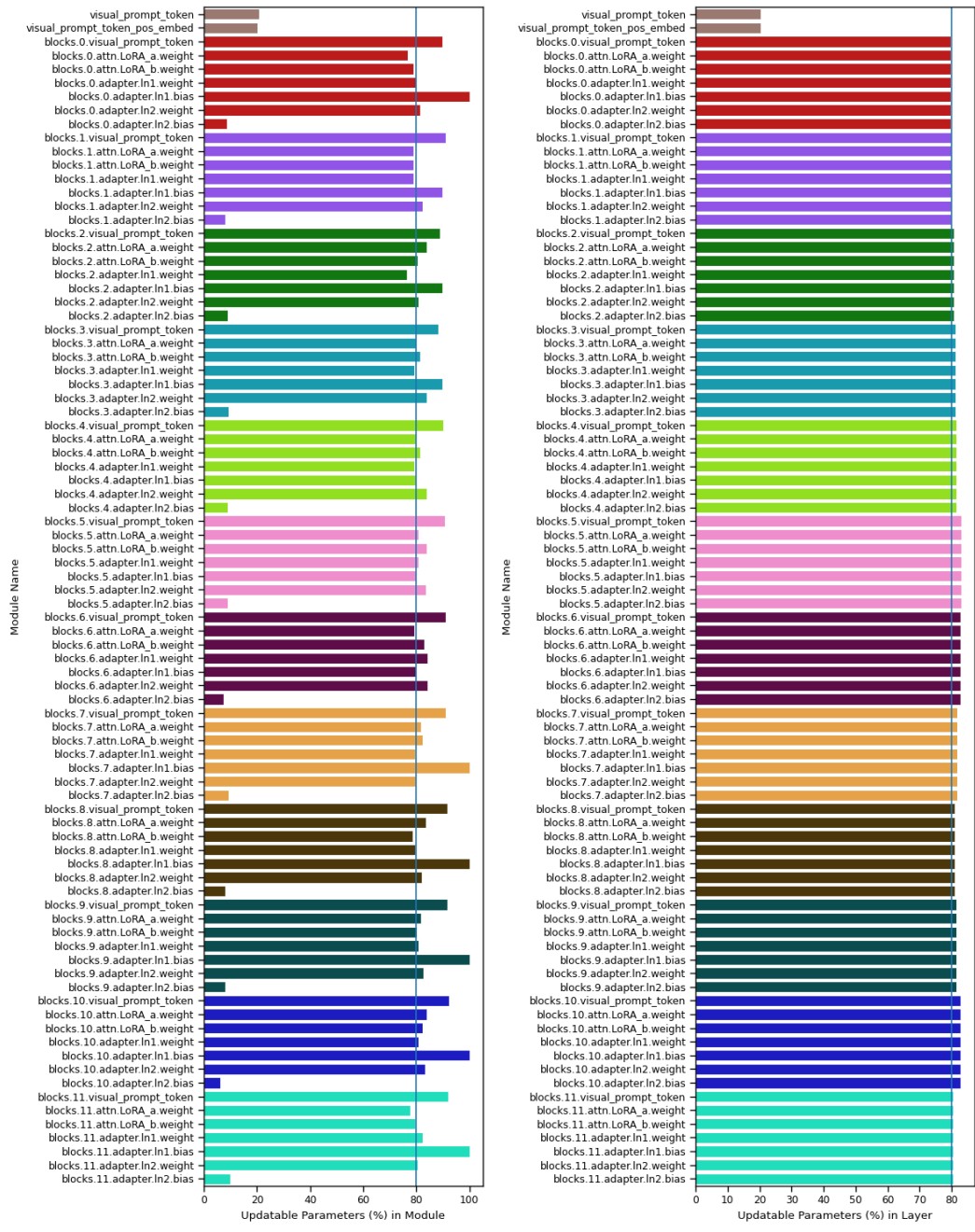

Figure 34: Sparsity pattern of attached modules to ViT-B/16 on kitti. (Left) Module-wise updatable parameters (%) and (Right) Layer-wise updatable parameters (%). The BayesTune's optimal sparsity level $p^* = 80\%$ is shown as vertical line.

# dsprite-loc

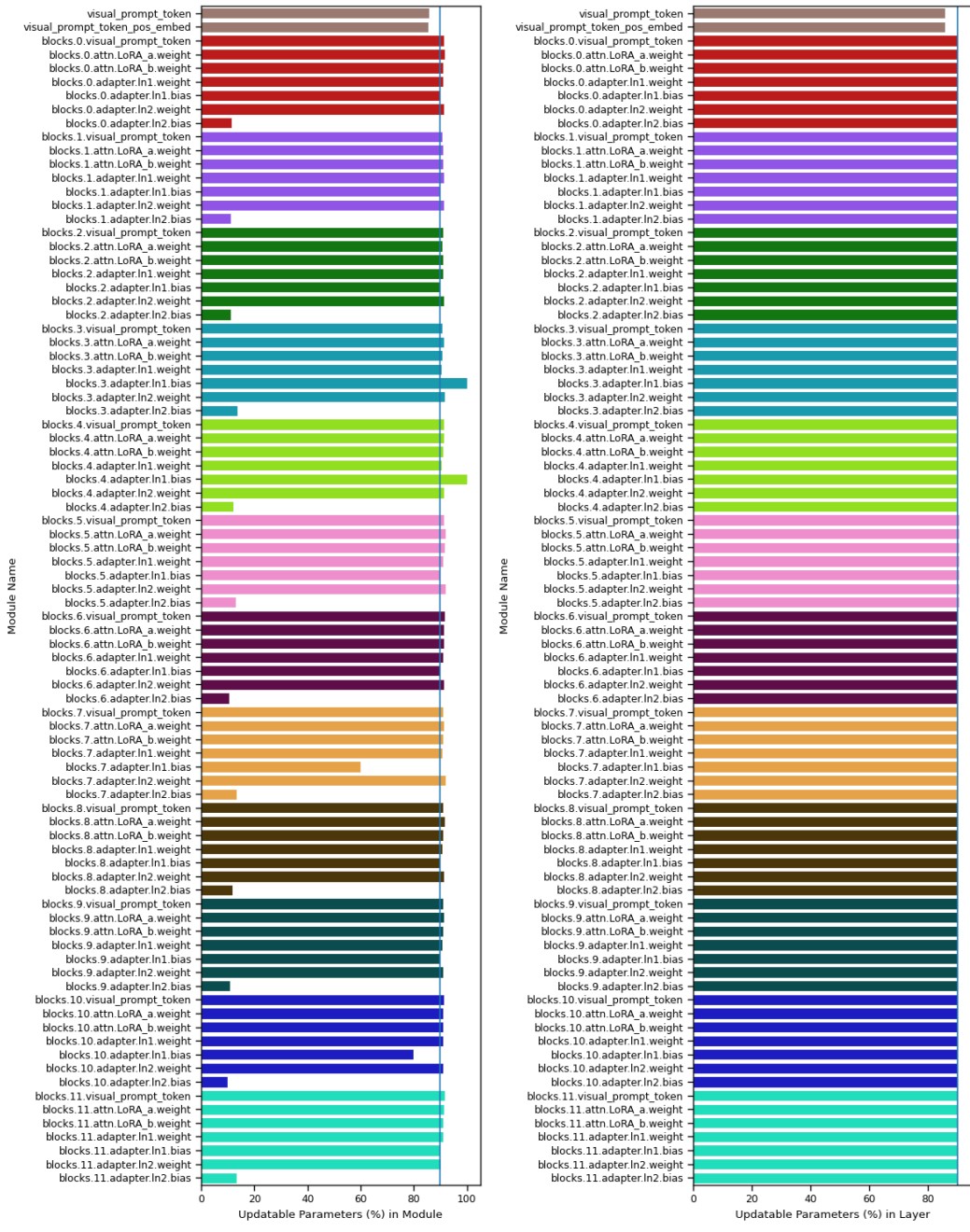

Figure 35: Sparsity pattern of attached modules to ViT-B/16 on `dsprite-loc`. (Left) Module-wise updatable parameters (%) and (Right) Layer-wise updatable parameters (%). The BayesTune's optimal sparsity level $p^* = 90\%$ is shown as vertical line.

# dsprite-ori

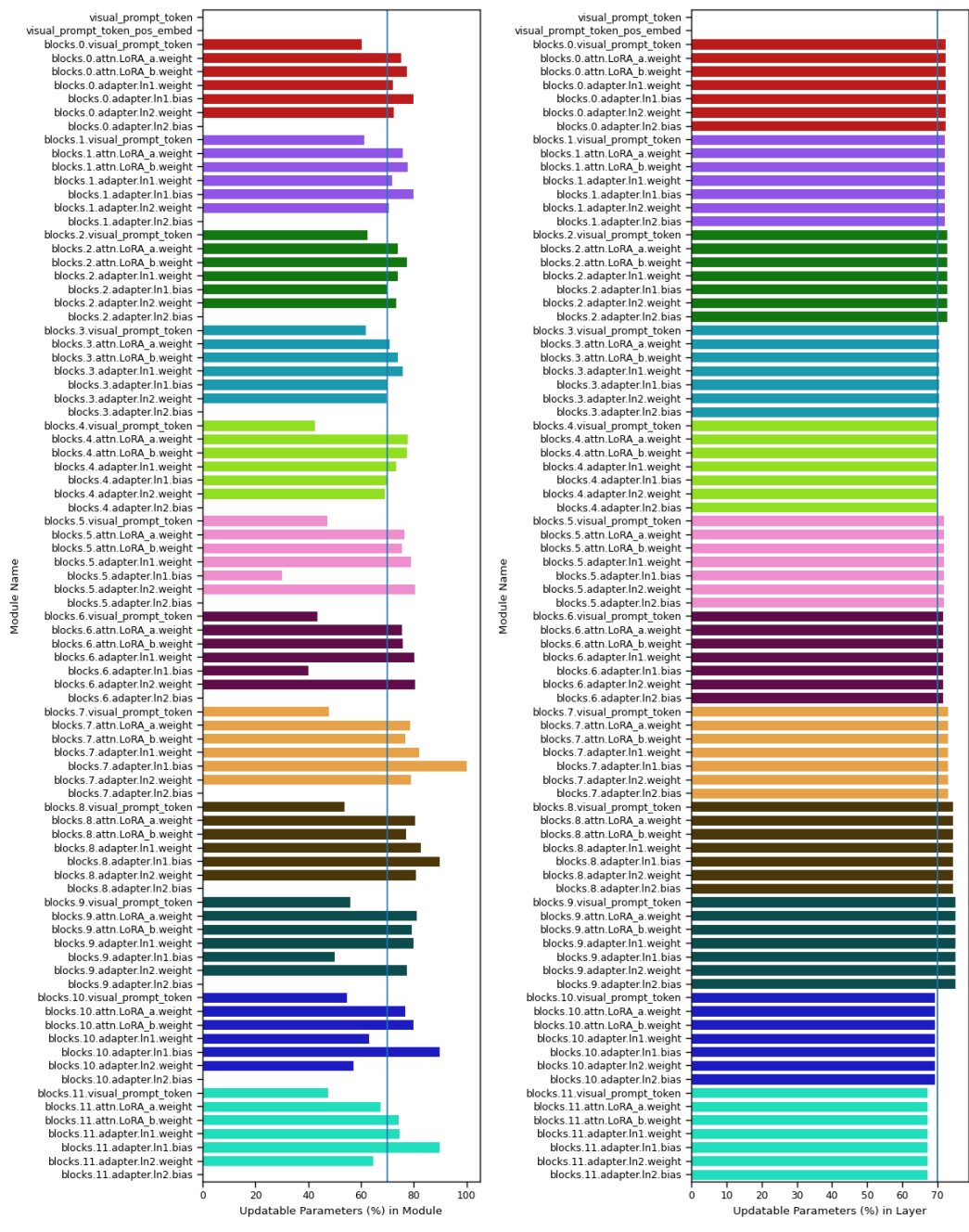

Figure 36: Sparsity pattern of attached modules to ViT-B/16 on `dsprite-ori`. (Left) Module-wise updatable parameters (%) and (Right) Layer-wise updatable parameters (%). The BayesTune's optimal sparsity level $p^* = 70\%$ is shown as vertical line.

**snorb-azim**

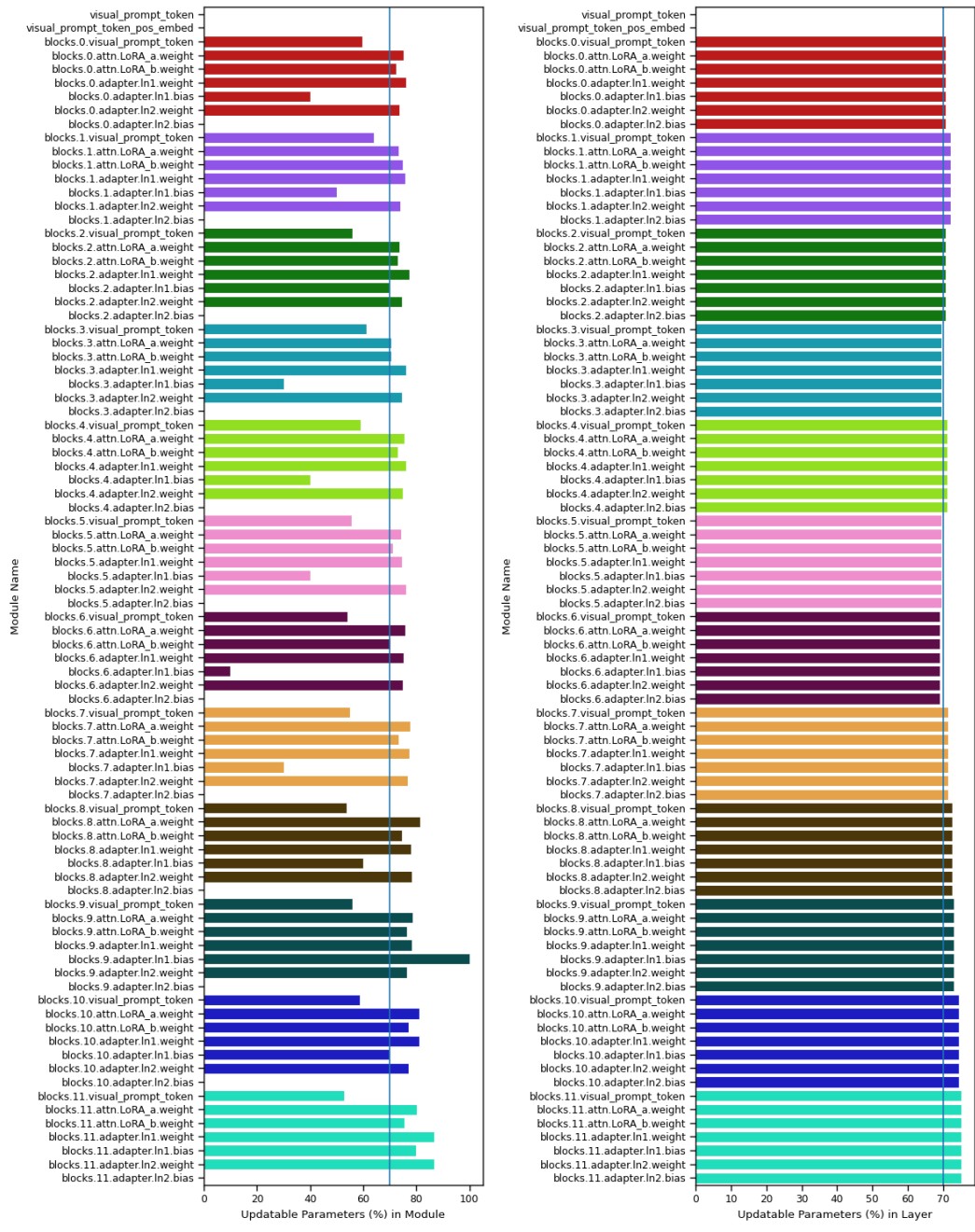

Figure 37: Sparsity pattern of attached modules to ViT-B/16 on `snorb-azim`. (Left) Module-wise updatable parameters (%) and (Right) Layer-wise updatable parameters (%). The BayesTune's optimal sparsity level $p^* = 70\%$ is shown as vertical line.

# snorb-ele

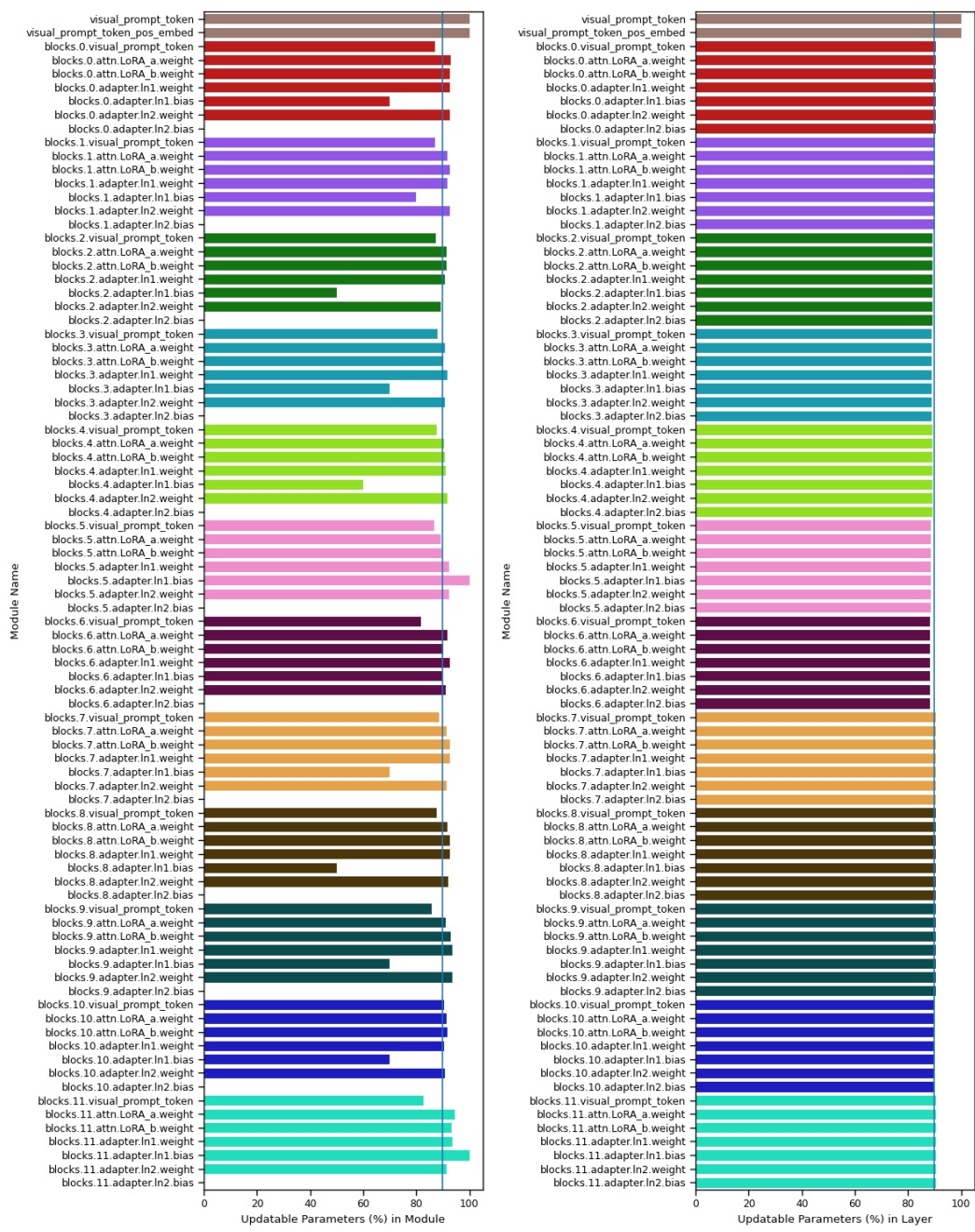

Figure 38: Sparsity pattern of attached modules to ViT-B/16 on `snorb-ele`. (Left) Module-wise updatable parameters (%) and (Right) Layer-wise updatable parameters (%). The BayesTune's optimal sparsity level $p^* = 90\%$ is shown as vertical line.

