# OpenReview forum: "BayesTune: Bayesian Sparse Deep Model Fine-tuning"
_NeurIPS.cc/2023/Conference — NeurIPS 2023 poster_

### Official Review · Reviewer_JDrk · 2023-06-21

**Soundness:** 3 good
**Presentation:** 4 excellent
**Contribution:** 2 fair
**Rating:** 5
**Confidence:** 4

**Summary:**

The authors introduce BayesTune, a method for choosing which are the parameters to fine-tune in a pre-trained model. Their formulation is based on Bayesian inference where they use a Laplace prior over the parameters/network weights. The prior has two variables: mean, which is the value of the pre-trained parameters, and scale which specifies how important is to fine-tune it. These hyperparameters are also controlled by a hyperprior that is fixed among all the experiments. For inferring the posterior distribution of the weights and scales. they adopt Langevin dynamic method. After obtaining the scale value, they compute a cut-off value that determines the parameters to be updated. Experiments performed in Computer Vision and NLP tasks demonstrate that the method is competitive and outperform common techniques.

**Strengths:**

* The problem of efficient fine-tuning is very important nowadays given the availability of large pre-trained models.
* The method is intuitive and interpretable, as it is based on the Bayesian interpretation of the network weights.
* There is a large set of experiments demonstrating the utility of the method.

**Weaknesses:**

* This model is close to related work such as SP-regularization [1] which uses a regularization term to encourage the updated weights to stay close to the original pre-trained weights. Using an L1-SP regularizer might have a similar effect as the one introduced in this paper. I think it is important the authors mention this and compare BayesTune against it.
* The efficiency of the method is not clear to me. For instance, in an attention layer, updating only some parameters (sparse updates) still demands computation of the matrix multiplications for query, key, and value matrices. The same logic applies when backpropagating. Thus, choosing only some parameters to update might only save memory demands, but not time. If my reasoning is correct, then the authors should specify this in the paper. Also, some measures of time and memory consumption might be useful.
* There is a discrepancy between Algorithm 1 and the method described in the paragraph starting at line 206. Thus, I think Algorithm 1 is incomplete.
* Their method seems competitive for NLP but not so for Computer Vision, based on Table 2.

[1] Xuhong, L. I., Yves Grandvalet, and Franck Davoine. "Explicit inductive bias for transfer learning with convolutional networks." International Conference on Machine Learning. PMLR, 2018.

**Questions:**

* Did you optimize the hyperparameters of the other methods?
* Why table 2 does not have standard deviations?
* Could you elaborate further on how you choose the cut-off point?
* How did you choose the hyperprior values ($\alpha, \beta$)?



**Limitations:**

No limitation was mentioned by the authors.

---

> ### Author Rebuttal · Authors · 2023-08-07
>
> >**1. This model is close to related work such as SP-regularization [1] which uses a regularization term to encourage the updated weights to stay close to the original pre-trained weights. Using an L1-SP regularizer might have a similar effect as the one introduced in this paper. I think it is important the authors mention this and compare BayesTune against it.**
>
> Thanks for the citation and good question. We will discuss this paper in the revision. We agree that SP [1] provides an alternative regularisation-based approach to fine-tuning (but without the principled Bayesian modeling solution), and that if SP [1] is extended to a L1 regulariser it might provide an alternative approach to sparse fine-tuning. Thus we have conducted some extra comparison experiments.
>
> Note that despite using an L1 regulariser, SP does not necessarily lead to exactly sparse solutions. Therefore we offer a two-stage extension of L1-SP. in the first stage, we run the L1-SP training, and in the second stage those weights to be updated are selected based on the relative L1 distances from the pretrained weights (ie, taking those $p$% weights with the largest relative changes from the pretrained weights).
>
> The results on NLP tasks are as follows:
>
> Avg 10 runs | CoLA | STS-B | MRPC | RTE | CB | COPA | WSC | **AVG**
> :---: | :---: | :---: |  :---: |  :---: |  :---: |  :---: | :---: | :---:
> L1-SP (stg 1) | 50.50 | 88.07 | 84.54 | 50.00 | 62.44 | 60.40 | 52.88 | 64.12
> L1-SP (stg-2, $p$%) | 54.59 | 88.11 | 89.25 | 68.85 | 81.55 | 70.75 | 55.38 | 72.64
> Our SGLD ($p$%) | 60.85 | 90.40 | 90.61 | 77.87 | 91.25 | 75.00 | 60.87 | 78.12
>
> For the L1 penalty balancing constant hyperparameter, we choose optimal values by grid search from {$10^{-3},10^{-4},10^{-5},10^{-6}$}. We can see that L1-SP considerably lags behind our SGLD, which is mainly attributed to its failure to capture uncertainty in L1 regularisation, thus being potentially sensitive to noise in data (a similar reason as the MAP estimate). Moreover, only penalising the parameters deviation from the pre-trained weights as in Stage 1 significantly underperforms the sparse cut-off strategy in Stage 2, signifying that sparse update is critical.
>
>
> >**2. The efficiency of the method is not clear to me. For instance, in an attention layer, updating only some parameters (sparse updates) still demands computation of the matrix multiplications for query, key, and value matrices. The same logic applies when backpropagating. Thus, choosing only some parameters to update might only save memory demands, but not time. If my reasoning is correct, then the authors should specify this in the paper. Also, some measures of time and memory consumption might be useful.**
>
> Thank you for the insightful comment. Yes, we agree on the overall backpropagation timing overhead in attention layers. We emphasise that our selective sparse fine-tuning competitors are also edge-wise (eg. SAM, DiffPrune, MagPrune), and thus our method is the same as theirs in this regard. We will clarify this in our revised paper.
>
> If saving latency/computation is important, our algorithm can easily be modified to provide this by sharing $\lambda$ across blocks or layers. EG: A layer-wise sparse $\lambda$ could avoid the need for attention computation in a certain layer. We actually explored this in our preliminary studies, but didn’t ultimately go down this path in the paper. Obtaining improved computation efficiency in this way would typically lead to slightly worse accuracy than the current edge-wise sparsity assumption. But we emphasise that this would be the same for all competitors, if they were correspondingly modified for layer/block-wise sparsity.
>
> >**3. There is a discrepancy between Algorithm 1 and the method described in the paragraph starting at line 206. Thus, I think Algorithm 1 is incomplete.**
>
> We left out the modifications from Alg.1 to make it concise, but we will add them to Alg.1.
>
> >**4. The method seems competitive for NLP but not so for Computer Vision, based on Table 2.**
>
> Yes. As we commented in the text Line 308-310, the main benefit/message is to avoid complex heuristic (evolutionary) search used by the state of the art competitor NOAH.
>
> >**5. Did you optimize the hyperparameters of the other methods?**
>
> We did not do it for ourselves as we excerpted the results from the respective previous papers. But each of the competing methods reports its best result after some hyperparameter tuning.
>
> >**6. Why Table 2 does not have standard deviations?**
>
> Because of the high cost of running the VTAB tasks, it is difficult to have many runs. We rather follow the official train/val/test split. Reporting the point estimates is also common practice for this benchmark.
>
> >**7. Could you elaborate further on how you choose the cut-off point?**
>
> For the NLP, we have a strict user-specified cut-off point $p=0.005$, being fair with other competing methods. For the VTAB, we have different cut-off points across tasks, and the optimal one is chosen as per the performance on the validation sets.
>
> >**8. How did you choose the hyperprior values ($\alpha$, $\beta$)?**
>
> It is just a heuristic choice.

---

> > ### Comment · Reviewer_JDrk · 2023-08-16
> > **Reply to rebuttal**
> >
> > Thanks to the authors for the detailed rebuttal. I consider updating my score from 4 to 5. Most of my doubts are cleared, however, still there are two points I would like to know:
> >
> > * How did you choose the hyperprior values and the cut-off points? How sensible are these hyperparameters across tasks? Some empirical measures regarding this would be beneficial.
> >
> > * How much is the average execution time for fine-tuning a network per task? Again, some empirical values are important.

---

> > > ### Author Response · Authors · 2023-08-17
> > >
> > > Thank you for the follow-up comments and questions. Our responses are as follows:
> > >
> > > > How did you choose the hyperprior values and the cut-off points? How sensible are these hyperparameters across tasks? Some empirical measures regarding this would be beneficial.
> > >
> > > **Hyperprior values:**
> > >
> > > We heuristically chose the hyperprior values $\alpha=0.01,\beta=100$ based on the mean/variance/skewness/kurtosis of the Gamma distribution to make the prior of $\lambda$ sharply decreasing away from 0. As stated in our paper (footnote-1, p.3), we also tested with models with further hierarchy by placing priors on $\alpha$ and $\beta$, however, there was no significant advantage over the manually chosen ones.
> > >
> > > **Cut-off points:**
> > >
> > > As stated in Line 305 (p.9), the cut-off points for the VTAB vision tasks were chosen by the grid search ($p \in [0.05, 0.1, 0.2, …, 1.0]$) on the validation set. For the sensitivity of the performance to the cut-off values $p$, Figure 3 (in p.9) can be referred to.  As shown, there exist significant differences in test accuracies for large changes of $p$, however, the sensitivity is rather minor near the optimal values.
> > >
> > > > How much is the average execution time for fine-tuning a network per task? Again, some empirical values are important.
> > >
> > > We have the running time records for the VTAB, where we ran our model on a single Tesla-V100 GPU. The task-wise averaged per-epoch running times are as follows. Other competing methods (eg, LoRA) have similar running times as our finetuning times in column Stage-2.
> > >
> > >  | (seconds) | Stage-1 | Stage-2 |
> > >  | :---: | :---: | :---: |
> > >  | cifar100 | 7.6 | 6.5 |
> > >  | caltech101 | 7.3 | 7.5
> > >  | dtd | 7.4 | 7.9 |
> > >  | flower102 | 8.0 | 8.1 |
> > >  | pets | 7.9 | 7.6 |
> > >  | svhn | 7.0 | 6.9 |
> > >  | sun397 | 7.5 | 7.5 |
> > >  | camelyon | 7.8 | 7.6 |
> > >  | eurosat | 7.1 | 7.5 |
> > >  | resisc45 | 7.5 | 7.5 |
> > >  | retinopathy | 8.1 | 7.8 |
> > >  | clevr-count | 7.0 | 7.6 |
> > >  | clevr-dist | 7.6 | 7.9 |
> > >  | dmlab | 8.1 | 7.8 |
> > >  | kitti | 8.0 | 8.4 |
> > >  | dsprite-loc | 7.9 | 7.8 |
> > >  | dsprite-ori | 7.5 | 7.7 |
> > >  | snorb-azim | 7.6 | 7.7 |
> > >  | snorb-ele | 7.9 | 7.4 |
> > >  | AVERAGE | 7.6 | 7.6 |

---

### Official Review · Reviewer_ZJiD · 2023-07-06

**Soundness:** 2 fair
**Presentation:** 3 good
**Contribution:** 3 good
**Rating:** 5
**Confidence:** 3

**Summary:**

The paper proposes an approach for selecting a subset of weights in a foundation model to fine-tune on a downstream task. The method consists of a two-stage pipeline, where in the first stage a Laplace prior is placed on each weight with a Gamma hyper-prior on the scale. Samples are obtained via SGLD and only the weights with a mean posterior scale above some threshold are then trained via SGD in the second stage. The method is evaluated on GLUE and SuperGLUE tasks with RoBERTa and on VTAB-1k image prediction tasks with a vision transformer and compares overall favorably to a range of baselines from the literature.

There are quite a few design choices constituting the proposed method and, unfortunately, none of their added complexity is justified via ablation studies. Further, the paper in my view overstates how principled it is quite significantly, so that at this point I would lean towards rejection.

**Strengths:**

* The approach is new as far as I am aware.
* The technical description of the method is clear.
* Performance seems to be good and approaches for better fine-tuning are of high interest to the community.

**Weaknesses:**

* The core problem in my view is that the method consists of quite a few moving parts, but these aren’t justified via ablations. It is not clear at all where the performance improvements come from and whether all parts of the method are needed. E.g. it might be the case that the two-stage procedure with magnitude-based pruning would be enough (at least my understanding of MagPruning based on the description in the paper is that the smallest pre-trained values are pruned). Similarly I wonder if sampling in Stage 1 is necessary or if MAP estimates would be good enough for the scale parameters.
* I don’t really see what makes the proposed method particularly principled as claimed at various points in the paper. I don’t think there is a probabilistic justification for the two-stage procedure and fudging the dataset size and noise scale for SGLD is just a hack.

**Questions:**

* How do the ablation baselines I mention in weakness 1 perform compared to the method as described in the paper?
* Is adapting the noise scale and dataset size really needed for stage 1? I’m aware of the cold posterior effect when sampling for a Bayesian model average, however here we don’t seem to need the predictions but the actual parameter samples? How much does tuning these improve performance over the principled choice?

**Typos/minor**:
* l155: “algorithmm” -> “algorithm”, “pseudocodes” -> “pseudocode”
* The abstract is extremely long without being particularly descriptive (e.g. the two-stage nature of the method isn’t even mentioned explicitly) and reads more like a small introduction. I’d suggest making it significantly more concise.
* For me there is way too much going on in Tab 1. I’d suggest at least dropping the bold-facing and rank indicators for the standard deviations, and making the rank indicators gray rather than red (unless you consider them primary information, in which case they should be bigger. But at the moment they are secondary in terms of size and position, so overall the table is unnecessarily difficult to process).

---

> ### Author Rebuttal · Authors · 2023-08-07
>
> Thank you for the insightful comments and suggestions. Our responses are as follows.
>
> >**1. Regarding ablation studies:**
>
> In response to the reviewer’s request, we have done the ablation studies: 1) Mag-Pruning with two-stage, and 2) MAP instead of SGLD. Please see the table below:
>
> Avg 10 runs | CoLA | STS-B | MRPC | RTE | CB | COPA | WSC | **AVG**
> :---: | :---: | :---: |  :---: |  :---: |  :---: |  :---: | :---: | :---:
> 2-stg MagPrune | 53.51 | 88.33 | 88.96 | 70.83 | 80.36 | 67.60 | 58.07 | 72.52
> MAP   | 58.55 | 90.13 | 90.54 | 76.72 | 86.71 | 71.09 | 60.34 | 76.30
> SGLD (Ours) | 60.85 | 90.40 | 90.61 | 77.87 | 91.25 | 75.00 | 60.87 | 78.12
>
> The results show that two stage mag pruning is not sufficient, and that posterior mean estimation via SGLD provides empirical benefit over MAP optimisation. We will add this ablation study in the revised version. We are grateful to the reviewer for helping improve the paper.
>
> >**2. Doubts about principled method:**
>
> The reviewer raised three components of the algorithm which s/he felt undermined the claims of a principled solution. We explain these as follows:
>
>   * **Two-stage procedure**:
>   Sparse Bayesian learners express a prior that prefers unmodified weights, but to enforce this as a hard constraint that can practically be used for memory saving, it is standard practice to  threshold after posterior probability inference (e.g. in the seminal Bayesian Compression for Deep Learning, NeurIPS’17; and the recent “Masked Bayesian Neural Networks” mentioned by reviewer **F3ir**). Since there is solid precedent for this standard step to bridge Bayesian models with practical implementation, we do not see it as compromising the principle of our method.
>
>   * **Dataset size inflation**:
> In Bayesian deep learning which depends on an explicit measure of dataset size, there has been discussion about how to quantify dataset size when using Data Augmentation. While this is not yet a completely solved question, several prior Bayesian deep learning studies have made similar suggestions on inflating the original training data size to account for augmentation (References below). Thus we believe this is entirely reasonable, and disagree that it is a hack:
>
>     - Disentangling the roles of curation, data-augmentation and the prior in the cold posterior effect, L. Noci et al., NeurIPS 2021.
>     - Practical deep learning with Bayesian principles, K.Osawa, et al., NeurIPS 2019.
>     - What are Bayesian neural network posteriors really like?, P. Izmailov et al., ICML 2021.
>
>   * **Noise scale**:
> Purely finding the posterior mean (ie, SGLD without noise discount) risks performing poorly if the posterior is truly multi-modal, because it may converge to a low probability parameter.
> Also, purely searching for the posterior mode (ie, MAP instead of SGLD) may be sensitive to data noise, because no stochasticity is taken into account properly. So for combining principle and practice it’s reasonable to prefer a discounted noise procedure that balances between identifying a particular mode, but gets a mean estimate in the vicinity of that mode.

---

> > ### Comment · Reviewer_ZJiD · 2023-08-15
> >
> > Thank you for the ablation results, I have decided to raise my score.

---

> > > ### Author Response · Authors · 2023-08-15
> > > **Thank you very much!**
> > >
> > > Thank you very much!

---

### Official Review · Reviewer_GHms · 2023-07-10

**Soundness:** 3 good
**Presentation:** 2 fair
**Contribution:** 3 good
**Rating:** 7
**Confidence:** 3

**Summary:**

A principled approach for selecting a subset of parameters to fine-tune in large foundational models is proposed. The authors rely on Bayesian inference to identify this subset. They begin by placing a Laplace prior over the model weights and a gamma hyperprior over the weight scale. Next, they employ an MCMC method to obtain a posterior distribution. Finally, they rank the weights based on the posterior scale values and proceed to fine-tune the parameters with the highest inferred scale values. Across standard NLP and vision adaption tasks, they demonstrate a strong empirical performance compared to previous state-of-the-art approaches.

**Strengths:**

- I find the proposed methodology technically sound, novel, and appealing. Though it should be kept in mind that I am not particularly familiar with the related work on the finetuning of large models. Similarly, I like methods that are Bayesian, so this might add to my potentially biased evaluation here.
- I like the simplicity of the proposed method. It does not happen often that I understand the method upon the first pass through the paper, but I think that was the case when reading this manuscript.
- The empirical results are strong and I want to commend the authors on extensive evaluation (i.e., they do not stop at the language modality, but additionally consider a vision domain as well).

**Weaknesses:**

A presentation could be somewhat improved to further strengthen the manuscript. Some concrete suggestions:
- I find it a bit strange to start introducing the notation already in the Introduction section. Hence, I would change your current section 1.1 into a separate section 2.
- Similarly, I find it a bit weird to list out the related work as bullet-points they way you do in Section 3. I find it more natural to use separate paragraphs for that (see Section 8 in [1] for an example of that). This way you can also directly talk about how each related approach connects to your proposed model, instead of doing that in a separate paragraph as is currently done (lines 192-198).
- Line 227: I assume you will include a GitHub link here, not put the code in the Supplementary material (whatever that means).
- Grammar and style could be improved at some points to improve readbility. This is easily done these days via tools like Grammarly, ChatGPT...

**Questions:**

- Although the choice of Laplace prior is adequately discussed and justified in Section 2, the explanation of the mean-field assumption in equation (1) could benefit from further elaboration. While I understand that this simplification is made for computational tractability, it would be valuable to provide additional justification for this decision. Additionally, I am curious whether the authors anticipate any additional performance improvements by attempting to model correlations between different model parameters. Similarly, I am interested in their choice of the approximate inference scheme (SGLD). Did the authors consider any other approaches (e.g., variational inference)?

**Limitations:**

Limitations are currently not discussed. Perhaps authors could use the gained space from restructuring the related work section (see above) to add a paragraph or two on the limitations of their approach.



[1] Daxberger, E., Nalisnick, E., Allingham, J.U., Antorán, J. and Hernández-Lobato, J.M., 2021, July. Bayesian deep learning via subnetwork inference. In International Conference on Machine Learning (pp. 2510-2521). PMLR.

---

> ### Author Rebuttal · Authors · 2023-08-07
>
> >**1. (The reviewer provided various detailed comments on paper layout.)
> Line 227: I assume you will include a GitHub link here, not put the code in the Supplementary material (whatever that means).
> Grammar and style could be improved.**
>
> Thank you very much for careful reading and all detailed suggestions that can improve the paper. We will refine the paper as the reviewer suggested. Regarding GitHub link: Because of the blind reviewing, we were not able to include the GitHub link at this point, but we will do it after the review phase.
>
> >**2. Although the choice of Laplace prior is adequately discussed and justified in Section 2, the explanation of the mean-field assumption in equation (1) could benefit from further elaboration. While I understand that this simplification is made for computational tractability, it would be valuable to provide additional justification for this decision.
> Any additional performance improvements by attempting to model correlations between different model parameters.
> About their choice of the approximate inference scheme (SGLD). Did the authors consider any other approaches (e.g., variational inference)?**
>
> The main reason for mean-field assumption (we guess that the reviewer meant *the prior being factorised over individual parameters*), is for simplicity. There might be benefits to correlation modeling/block sparsity, (which is easy to implement in our framework by using a common $\lambda$ shared across a block), but we did not do this initially as it raises the question of how to choose the block structure. Therefore we leave this to future work for now.
>
> Variational inference can be used in principle, but as it incurs additional complexity (both memory and FLOPS) for handling doubled posterior parameters (means and scales), we opted for SGLD. It may also suffer from lower accuracy due to needing an additional assumption on the posterior form (eg., Gaussian), which may introduce an additional source of approximation inaccuracy.

---

> > ### Comment · Reviewer_GHms · 2023-08-18
> >
> > Thank you for your rebuttal, I acknowledge I have read it together with other reviews.
> >
> > > guess that the reviewer meant the prior being factorised over individual parameters
> >
> > Indeed, that's what I meant.
> >
> > > Variational inference can be used in principle, but as it incurs additional complexity (both memory and FLOPS) for handling doubled posterior parameters (means and scales), we opted for SGLD.
> >
> > Hm, interesting, I thought VI is meant to be a cheaper alternative to MCMC approaches.

---

### Official Review · Reviewer_F3ir · 2023-07-10

**Soundness:** 3 good
**Presentation:** 3 good
**Contribution:** 2 fair
**Rating:** 5
**Confidence:** 3

**Summary:**

The paper proposes a new Bayes-based framework for sparse fine-tuning. The proposed method applies a Laplace prior (centered at the pre-trained weights) and a hyper Gamma prior over the scale parameter \lambda of the Laplace prior. The method then performs posterior inference over \lambda to determine whether or not a parameter is "useful": A large value of  \lambda indicates a flat prior, i.e. an informative parameter, and vice versa. The paper then adopts SGLD to perform inference over \lambda and then uses human assistance to determine the cut point for \lambda given a budget. The method is evaluated on language and vision tasks and compared against a wide range of parameter efficient fine-tuning methods. The proposed method overall demonstrates performance superior and acquires a sparser model than baseline methods.

## Post rebuttal update: I raised my score from 4 to 5 seeing the new results provided by the author.

**Strengths:**

- The proposed method is technically sound and well presented.

- The proposed method is evaluated on a wide range of tasks.

- The authors consider a thorough amount of baseline methods.

**Weaknesses:**

The biggest issue of the proposed method, in my opinion, is the novelty. The use hierarchical prior, which induces sparsity, is a widely known idea since the last century, from Radford Neal and David MacKay's early works to recent works such as [1, 2, 3, 4]. Although these works do not consider the fine-tuning setting, they can technically be applied to the fine-tuning of large pre-trained models by simply letting the prior to be centered at the pre-trained weigh rather than zero.

In addition, the configuration of SGLD is not presented very clearly. I would suggest the author use formula to describe the modifications to SGLD described from line 212 to line 223, e.g. a modified version of Eq.5 where a few additional hyper-parameters are added to the likelihood and the injected noise term.

*Minor: Some figures are not in vector format and the fonts are too small.

*Minor: It would be interesting to see the performance of the proposed model on *Large* language model.


[1] Masked Bayesian Neural Networks : Theoretical Guarantee and its Posterior Inference
[2] Dropout as a structured shrinkage prior.
[3] Posterior concentration for sparse deep learning.
[4] Bayesian compression for deep learning

**Questions:**

- What optimizers are used for BayesTune? Is it standard SGD through out the entire algorithm or it is a mix of SGD and Adam.

- Does the baseline approach use same optimizer or they use different optimizers? The choice of optimizer can potentially affect the final performance.

- What happen if you increase the rank of LoRA to let LoRA have the same the number of parameter used by BayesTune?

- What's the advantage of posterior inference over MAP estimation in this setting, i.e. remove the last term in Eq.5?


**Limitations:**

The proposed method introduces extra hyper-parameter for SGLD (besides step-size scheduling): Effective data size and noise discount factor.

---

> ### Author Rebuttal · Authors · 2023-08-07
>
> >**1. The biggest issue is the novelty. They use hierarchical prior, which induces sparsity, is a widely known idea since the last century, from Radford Neal and David MacKay's early works to recent works [1,2,3,4]. Although these works do not consider the fine-tuning setting, they can technically be applied to the fine-tuning of large pre-trained models ...**
>
> Hierarchical Bayes (HB) is indeed well known, and there were some works that adopt HB for sparse deep learning as the reviewer listed. But as far as we know, our approach has three main differences from these previous works:
>
>   - i) Prior works are all about sparse *training*, instead of sparse *fine-tuning*. Thus they focus on zeroing out many parameters, instead of retaining pre-trained weights. The reviewer said “These works can technically be applied to fine-tuning …”, but to the best of our knowledge, no one has done it before.
>
>   - ii) Importantly, the neural networks used in those previous studies are rather small/toy scale (mostly focusing on MLPs and LeNet sized architectures up to RN18 at largest)  while our method can obtain state of the art results on large-scale foundation models (ViT, RoBERTa). For example the largest model considered by the cited papers, RN18 in [1] is ~11M parameters vs RoBERTa’s ~123M parameters, a 10X scale difference.
> The reason is that the Bayesian learning methods the reviewer referred to used inefficient MCMC approaches like Metropolis-Hastings, or methods that entail extra memory cost like variational inference (VI). All of this impedes applicability to big networks. We have just done some experiments that compare the computational resources required by VI and SGLD: On ViT networks, the training time is increased by 1.7 times if we replace SGLD by VI; the GPU memory footprint is increased by 2.1 times.
>
>   - iii) As far as we know, no one has used SGLD in sparse deep learning at these large network scales. Although [1] (Masked BNN) used MCMC, it adopted Metropolis-Hastings, which is less sample efficient due to the rejection probability.
>
> >**2. I would suggest the author use formula to describe the modifications to SGLD described from line 212 to line 223.**
>
> As the reviewer suggested, and also for the purpose of better clarification, we will re-write Eq.5 to reflect the modification as described in Line212-223.
>
> >**3. What optimizers are used for BayesTune? Does the baseline approach use same optimizer?**
>
> The Adam optimizer is used for all competing methods on Tab 1, following the common practice in [Lee19, Jiant20, Xu21]. For the vision tasks (VTAB), all methods use the AdamW optimizer following standard practice in the VTAB benchmark suite.
>
>   - [Lee19] Mixout: Effective regularization to finetune large-scale pretrained language models, C. Lee et al., ICLR 2019.
>   - [Jiant20] Jiant2.0: A software toolkit for research on general-purpose text understanding models, J. Phang et al., http://jiant.info/, 2020.
>   - [Xu21] Raise a child in large language model: Towards effective and generalizable finetuning, R. Xu et al., EMNLP 2021.
>
> >**4. What happens if LoRA have the same the number of parameter used by BayesTune?**
>
> For NLP tasks, we have the same sparsity level ($p=0.005$) for all competing methods, so it is already a completely controlled comparison. For vision (VTAB) tasks, we think that the reported LoRA with dim=8 was the optimal hyperparameter choice. However, as the reviewer suggested, we can match the sparsity levels of LoRA and our BayesTune for a more controlled comparison. Please recall that LoRA-dim8 amounts to updating 0.29M parameters while BayesTune updated 0.38M parameters.
>
> Instead of increasing the dimension of LoRA, which might require corresponding re-tuning of other LoRA learner hyperparameters, thus potentially being unfair to LoRA, we instead decrease the sparsity level for our  BayesTune so that we have the same 0.29M parameters updated. The results are:
>
> Model (# params updated) | Average Rank | # times Rank=1
> :------: | :---: | :----:
> LoRA (0.29M)       |  2.68        | 4
> BayesTune (0.38M)        |  2.37            | 7
> BayesTune (0.29M)          |  2.58            | 6
>
> So, even when the number of parameters are made equal, BayesTune outperforms LoRA.
> (Note here that we used the latest version Table 3 in Appendix, instead of Table 2).
> Please also note that the other existing competitors in the VTAB comparison used to compute ranks above are not parameter count controlled like this, as parameter count controlling was not standard practice in prior work.
>
> >**5. What's the advantage of posterior inference over MAP estimation in this setting, i.e. remove the last term in Eq.5?**
>
> MAP vs. SGLD: MAP aims to find a mode of the posterior distribution, which might be more sensitive to data noise than the mean of the posterior.
>
> The following is some empirical comparison between MAP and SGLD on NLP tasks, showing that this distinction does lead to empirical benefit:
>
> Avg 10 runs | CoLA | STS-B | MRPC | RTE | CB | COPA | WSC | **AVG**
> :--: | :--: | :--: |  :--: |  :--: |  :--: |  :--: | :--: | :--:
> MAP   | 58.55 | 90.13 | 90.54 | 76.72 | 86.71 | 71.09 | 60.34 | 76.30
> SGLD  | 60.85 | 90.40 | 90.61 | 77.87 | 91.25 | 75.00 | 60.87 | 78.12
>
> Another benefit of SGLD is that we can also exploit the variance of the $\lambda$ posterior in parameter selection (e.g., for two parameters with similar posterior mean $\lambda$ values, we prefer to select the one with smaller posterior variance).   In future algorithms, we can also exploit this idea of variance-based weight pruning (but we didn’t do this yet).
>
> >**6. (Minor) Some figures are not in vector format and the fonts are too small.
> It would be interesting to see the performance of the proposed model on Large language model.**
>
> We apologize for this. After the review/rebuttal phase, we will replace them with vector format figures. We plan to do it on LLMs even larger than RoBERTa, eg, LLAMA, in our future work.

---

> > ### Comment · Reviewer_F3ir · 2023-08-11
> > **Post rebuttal comment**
> >
> > I would like to thank the author for the detailed response, however I cannot agree with the authors' argument regarding MCMC and hierarchical Bayes methods
> >
> > - The author should consider adding more discussions on the many prior arts for hierarchical Bayes prior's application in Bayesian deep learning. At this point, I did not see any citations for these literatures.
> >
> > - It is NOT accurate to say using Metropolis Hastings (MH) adjustment for SGLD is  **inefficient**. If I understand correctly, the author of [1] uses MH to ensure the unbiasedness of posterior approximation, which is not discussed in this submission. From my opinion, it is OK and a common practice to drop MH adjustment in Bayesian deep learning, but it is not reasonable consider this as an advantage. In fact, I believe the SGLD used in the submission is just the most standard SGLD without any modifications, with that said, I do agree that **it is a novel application of SGLD** but the authors do not tailor on the inference algorithm for foundation model fine-tuning setting.
> >
> > - I apologize for the confusion in my question regarding the choice of optimizer, the major question I have is: When implementing SGLD, does the author strictly follow Eq.5 or uses "Adam + gradient noise", which is a commonly seen incorrectly implemented SGLD. In fact, if the author needs preconditioning and adaptive step size together with SGLD, the author should consider the version of SGLD provided by the paper "Bayesian Neural Network Priors Revisited".
> >
> >  - I would like to thank the author for the additional experiments on LoRA rank, it resolves my concern.
> >
> > - MAP v.s. SGLD: I would like to thank the author for providing the additional experiments. It would be good if the author can provide more comparision on the value of \lambda acquired and details on how the MAP is acquired in later revisions.
> >
> > I decided to raise my score however I still believe the paper's details in the Bayes part should be discussed more clearly, as SGLD is an algorithm that involves many details, e.g. step size scheduling, temperature setting, model/parameter ensembling, etc. Presenting the details (and potentially the sensitivity to those hyper parameters) more clearly can allow better reproducibility.

---

> > > ### Author Response · Authors · 2023-08-15
> > > **Thank you for the post rebuttal comments!**
> > >
> > > We thank the reviewer again for valuable post rebuttal comments. Our responses to the follow-up questions and comments are as follows:
> > >
> > > > **1. The author should consider adding more discussions on the many prior arts for hierarchical Bayes prior's application in Bayesian deep learning. At this point, I did not see any citations for these literatures.**
> > >
> > > We promise that we will add those papers on hierarchical Bayesian methods with applications to deep learning. We will do extensive investigation on the most relevant and recent prior works, as well as including the followings as suggested by the reviewer,
> > >
> > >   - [1] “Masked Bayesian Neural Networks : Theoretical Guarantee and its Posterior Inference”, Insung Kong, Dongyoon Yang, Jongjin Lee, Ilsang Ohn, Gyuseung Baek, Yongdai Kim, ICML 2023
> > >
> > >   - [2] “Dropout as a structured shrinkage prior”, Eric Nalisnick, José Miguel Hernández-Lobato, Padhraic Smyth, ICML 2019
> > >
> > >   - [3] “Posterior concentration for sparse deep learning”, Nicholas Polson, Veronika Rockova, NeurIPS 2018
> > >
> > >   - [4] “Bayesian compression for deep learning”, Christos Louizos, Karen Ullrich, Max Welling, NeurIPS 2017
> > >
> > >
> > > > **2. It is NOT accurate to say using Metropolis Hastings (MH) adjustment for SGLD is inefficient. If I understand correctly, the author of [1] uses MH to ensure the unbiasedness of posterior approximation, ... I do agree that it is a novel application of SGLD but the authors do not tailor on the inference algorithm for foundation model fine-tuning setting.**
> > >
> > > We agree with the reviewer’s concerns and we will remove the parts we said that MH is computationally inefficient. Our statement in the response was over-exaggeration.
> > >
> > > Yes, we used basic settings for SGLD. Investigating further optimisation for SGLD as the reviewer suggested, will be valuable and promising, and we will pursue this in the future work. Thank you for your insightful comments!
> > >
> > >
> > > > **3. I apologize for the confusion in my question regarding the choice of optimizer, the major question I have is: When implementing SGLD, does the author strictly follow Eq.5 or uses "Adam + gradient noise" ...**
> > >
> > > We used the Adam optimiser for updating the model parameters, thus we believe that there was some sort of (internal) gradient adaptation and momentum effect under the hood. To be honest, we were not aware of the implementation detail that the reviewer mentioned although we found some previously proposed strategies that considered adaptive drift and momentum in SGLD (eg, https://arxiv.org/pdf/2009.09535.pdf). In this regard, we thank the reviewer for the in-depth comments. Even though we doubt that our SGLD update scheme with Adam could lead to a significantly different solution compared to the original SGLD formulation, we will consider (and cite) those references the reviewer pointed out in our revised paper.
> > >
> > >
> > > > **4. MAP v.s. SGLD: I would like to thank the author for providing the additional experiments. It would be good if the author can provide more comparison on the value of $\lambda$ acquired and details on how the MAP is acquired in later revisions.**
> > >
> > > *1) The learned $\lambda$ values comparison between MAP and SGLD*:
> > > In response to reviewer's request, we have prepared figures comparing the sparsity patterns of the learned $\lambda$s for our SGLD and the MAP solution, on the NLP benchmarks (similar to Figure 6 and the rest in our submitted appendix). Due to the difficulty of sharing figures at this discussion stage, we only visualise some small snapshot as a table at the bottom of this thread. This is for the COPA task and module-wise sparsity patterns of the learned SGLD and MAP. Visually we find that they exhibit quite different sparsity patterns, indicating that the impact of the noise/drift term in our SGLD is significant. We will add the full figures of SGLD and MAP that we prepared in our revised paper.
> > >
> > > *2) How the MAP is acquired*:
> > > For the MAP we dropped the last noise term in Eq.(5). The rest steps are the same as SGLD.
> > >
> > > | Module# | Module name | SGLD (% updated) | MAP (% updated) |
> > > | :--- | :--- | :---: | :---: |
> > > | 150 | encoder.layer.9.attention.self.query.bias | 0.91 | 0.00 |
> > > | 45 | encoder.layer.2.attention.output.LayerNorm.weight | 0.65 | 3.26 |
> > > | 182 | encoder.layer.11.attention.self.query.bias | 0.78 | 0.00 |
> > > | 93 | encoder.layer.5.attention.output.LayerNorm.weight | 0.52 | 3.12 |
> > > | 178 | encoder.layer.10.output.dense.bias | 1.17 | 0.39 |
> > > | 13 | encoder.layer.0.attention.output.LayerNorm.weight | 0.65 | 3.12 |
> > > | 166 | encoder.layer.10.attention.self.query.bias | 0.91 | 0.13 |
> > > | 4 | embeddings.LayerNorm.bias | 0.39 | 2.86 |
> > > | 134 | encoder.layer.8.attention.self.query.bias | 1.04 | 0.26 |
> > > | 99 | encoder.layer.5.output.LayerNorm.weight | 0.39 | 2.34 |
> > > | 181 | encoder.layer.11.attention.self.query.weight | 0.73 | 0.05 |
> > > | 115 | encoder.layer.6.output.LayerNorm.weight | 0.39 | 2.34 |
> > >
> > > We will keep the reviewer’s points in mind when we prepare a revised version. Thank you very much.

---

### Official Review · Reviewer_2Yw3 · 2023-07-13

**Soundness:** 4 excellent
**Presentation:** 4 excellent
**Contribution:** 3 good
**Rating:** 8
**Confidence:** 4

**Summary:**

The authors propose an automated sparse fine-tuning method for foundations model, that bypasses the need for human intuition-based heuristics. The neurons to update are revealed during the posterior inference of the sparse scale parameters of a Laplace prior, by thresholding the scale parameters.
The method is experimentally validated in both vision  and NLP tasks.

**Strengths:**

The proposed method is principled, and relies on a hierarchical Bayesian model.

Posterior approximation is done with Langevin MCMC, which does not introduce a significant computational overhead.

The experimental results show that the proposed method convincingly improves upon already existing heuristics.

**Weaknesses:**

I wonder what's the rationale behind using the elbow rule (figure 1) to select the proportion p of parameters to update.

**Questions:**

- Why was the Gamma distribution chosen as a hyperprior ?
- If the step after SGLD is to evaluate the mean of the scale parameters, aren't there cheaper methods that evaluate the mean, without stochastically approximating the posterior ?

---

> ### Author Rebuttal · Authors · 2023-08-07
>
> >**1. I wonder what's the rationale behind using the elbow rule (figure 1) to select the proportion** $p$ **of parameters to update.**
>
> This is just an illustration of one potential heuristic method to select the sparsity level $p$. Other criteria could also be used.
>
> >**2. Why was the Gamma distribution chosen as a hyperprior ?
> If the step after SGLD is to evaluate the mean of the scale parameters, aren't there cheaper methods that evaluate the mean, without stochastically approximating the posterior ?**
>
> We adopt Gamma because with $\alpha<1$ it has mode at $0$ and it is decreasing, thus allowing us to express our preference of small $\lambda$ a priori. But any other distribution with this property could also be used as a hyperprior.
>
> *Regarding SGLD posterior mean*: Yes, one can use other methods to evaluate the posterior mean or any surrogate for it, eg, SWAG or EMA could potentially be used to estimate the posterior mean. However SGLD is also as efficient as these other alternative methods, and benefits from the fact that SGLD’s stochastic dynamics guarantee (in theory) to lead to the posterior mean exactly.

---

> > ### Comment · Reviewer_2Yw3 · 2023-08-14
> >
> > Thank you for your answers !

---

> > > ### Author Response · Authors · 2023-08-15
> > > **Thank you very much!**
> > >
> > > Thank you very much!

---

### Decision · Program_Chairs · 2023-09-21

**Decision:**

Accept (poster)

**Comment:**

This paper introduces an interesting algorithm for fine-tuning foundation models, namely a sparse Bayesian fine-tuning method employing hierarchical Laplace priors. The proposed method has received positive feedback from most reviewers, who acknowledge its novelty and potential practical impact. However, a notable concern revolves around the implementation details of the SGLD procedure. Specifically, the authors have implemented SGLD as Adam with noise, which raises questions about whether this truly corresponds to a valid adaptive version of SGLD. This aspect, in turn, prompts doubts about the paper's claim that the proposed algorithm constitutes a "Bayes" Tuning method. Reviewers encourage the authors to address this issue in the final version of the manuscript.